



# Can canopy interception and biomass be inferred from cosmic-ray neutron intensity? Results from neutron transport modeling.

Mie Andreasen[1], Karsten H. Jensen[1], Darin Desilets[2], Marek Zreda[3], Heye Bogena[4] and Majken C. Looms[1]

[1] Department of Geosciences and Natural Resource Management, University of Copenhagen, Denmark

[2] Hydroinnova LLC, Albuquerque, New Mexico

[3] Department of Hydrology and Water Resources, University of Arizona, Arizona

[4] Agrosphere IBG-3, Forschungszentrum Juelich GmbH, Germany

*Correspondence to*: Mie Andreasen (mie.andreasen@ign.ku.dk)

**Keywords**

1. Cosmic-ray neutron intensity method
2. Neutron transport modeling
3. Canopy interception
4. Forest biomass

**Abstract**

Cosmic-ray neutron intensity is inversely correlated to all hydrogen present in the upper decimeters of the subsurface and the first few hectometers of the atmosphere above the ground surface. This method has been used for measuring soil moisture and snow water equivalent, but it may also be used to identify and quantify canopy interception and biomass. We use a neutron transport model with various representations of the forest and different parameters describing the subsurface to match measured profiles and time series of thermal and epithermal neutron intensities at a field site in Denmark. A sensitivity analysis is performed to quantify the effect of forest canopy representation, soil moisture, complexity of soil matrix chemistry, forest litter, soil bulk density, canopy interception and forest biomass on neutron intensity. The results show that forest biomass has a significant influence on the neutron intensity profiles at the examined field site, altering both the shape of the profiles and the ground level thermal-to-epithermal neutron ratio. The ground level thermal-to-epithermal neutron ratio increases significantly with increasing amounts of biomass and minor with canopy interception. Satisfactory agreement is found between measurements and model results at the forest site as well as two nearby sites representing agricultural and heathland ecosystems. The measured ground level thermal-to-epithermal neutron ratios of the three site range from around 0.56 to 0.82. The significantly smaller effect of canopy interception on the ground level thermal-to-epithermal neutron ratio was modeled to range from 0.804 to 0.836 for a forest with a dry and a very wet canopy (4 mm of canopy interception), respectively. At the examined field site the signal of the canopy interception is lower than the measurement uncertainty.





### 1. Introduction

The cosmic-ray neutron intensity (eV range) at the ground surface is a product of the elemental composition and density of the immediate air and soil matrix. Hydrogen is, because of its physical properties and often relative high concentration, a significant element controlling neutron transport. As a result, neutron intensity is inversely correlated with the hydrogen
content of the surrounding hectometers of air and top decimeters of the ground (Zreda et al., 2008). Neutron intensity measurements were found to be suitable for the detection of soil moisture since it often forms the major dynamic pool of hydrogen within the footprint of the detector. Soil moisture plays an important role in water and energy exchanges at the ground-atmosphere interface, but is difficult and expensive to measure at the intermediate scale. The cosmic-ray method has been developed to circumvent the shortcomings of existing measurement procedures for soil moisture detection at the multi
hectare scale (e.g. Zreda et al. (2008) and Franz et al. (2012)).

Cosmic-ray neutron intensity detection also has potential for estimating other pools of hydrogen present within the footprint of the neutron detector (Zreda et al., 2008; Desilets et al., 2010). Hydrogen is stored statically, quasi-statically or dynamically in soil water, atmospheric water vapor, water in soil minerals, soil organic matter, snow, buildings/roads, above and below ground biomass, and canopy intercepted precipitation (see Table 1). The signal of some of these hydrogen pools
has already been investigated with the aim of correcting cosmic-ray neutron soil moisture measurements. A scheme to correct the fast neutron signal for time-varying atmospheric water vapor was developed by Rosolem et al. (2013) using neutron transport modeling; and a correction function was determined to account for above-ground biomass based on field measurements (Baatz et al., 2015). Until now independent measurements of neutrons impacted by hydrogen pools other than soil moisture have received little attention.

*Table 1 is inserted here*

The ability to separate signals of canopy interception and biomass from the cosmic-ray neutron intensity would be valuable as they form essential hydrological and ecological variables. Both are difficult and expensive to measure continuously at larger scales. Although the unwanted effect of biomass growth on cosmic-ray estimated soil moisture (Hornbuckle et al., 2012) can potentially be accounted for using independent methods (thereby improving soil moisture determinations), there
is currently no established method for independently constraining biomass based on neutron data alone.

Canopy interception is for some climatic and environmental settings an important variable to include in water balance studies, as well as in hydrological and climatological modeling. For the forest site studied here the canopy interception loss was found to be 31-34% of the gross precipitation, making it a vital variable to consider (Ringgaard et al., 2014). A common method to estimate canopy interception is by subtracting the precipitation measured at ground level below canopy
(throughfall) from precipitation measured above the forest canopy (gross precipitation) using standard precipitation gauges. The spatial scale of measurement is small and is not representative of larger areas as the canopy interception is highly heterogeneous. In order to obtain a representative measure of canopy interception multiple throughfall stations must be installed. This is labor intensive and measurement uncertainties are significant. Precipitation underestimation due to wind





turbulence, wetting loss, and forest debris plugging the measurement gauge at the forest floor are sources of significant uncertainty (Dunkerley, 2000).

The forest biomass represents an important resource for timber industry and renewable energy. Furthermore, forest modifies the weather through the mechanisms and feedbacks related to evapotranspiration, surface albedo and roughness. Overall, the

forest ecosystems have a cooling impact on global climate as significant amounts of carbon are accumulated through photosynthesis. Carbon sequestration by afforestation and an effective forest management is a widely used method to decrease the concentration of carbon dioxide in the atmosphere and thereby attenuate the greenhouse effect (Lal, 2008). The carbon sequestration in vegetation can be quantified by monitoring the growth of biomass over time. The most conventional and accurate method to estimate forest biomass is the use of allometric models describing the relationship between the

biomass of a specific tree species and easily measurable tree parameters, such as tree height and tree diameter at breast height (Jenkins et al., 2003). However, this approach is time consuming and labor intensive because numerous trees have to be surveyed to obtain accurate and representative results (Popescu, 2007). Remote sensing technology offers alternative methods to estimate biomass as high correlations are found between spectral bands and vegetation parameters. One method providing high resolution maps is airborne *Light Detection And Ranging* (LiDAR) technology (Boudreau et al., 2008). The

LiDAR system is installed in small aircrafts and digitizes the first and last return of near-infrared laser recordings. The canopy height at a decimeter grid-size scale can be obtained and the biomass can be estimated from regression models. Instruments and aircraft-surveys are expensive, and measurements of tree growth will often be at a coarse temporal resolution.

Here, the potential of detecting intermediate scale canopy interception and biomass from cosmic-ray neutron intensities is

investigated. The analysis is based on thermal and epithermal neutron intensity profiles of a forest boundary layer using measurements from proportional detectors and modeling. Thermal and epithermal neutron intensity measurements are obtained from measurements using bare and moderated detectors constrained with correction factor models (Andreasen et al., 2016). Modeling is based on the recognized and widely used neutron transport model MCNP6 (Pelowitz, 2013).

Neutron transport modeling of specific sites is limited and has only been performed for non-vegetated field sites (Franz et

al., 2013; Andreasen et al., 2016). In this context, forest sites are especially complex to conceptualize as the number of free parameters is very high (e.g. biomass, litter, soil chemistry, interception and the structure of the forest). In this study, we focus on a sensitivity analysis of various forest canopy conceptualization model setups, forest parameters and variables to identify and quantify their effect on modelled thermal and epithermal neutron intensity at forest sites.

The effects are identified and quantified in relation to a reference model of the forest field site. This model is developed

from measured soil and vegetation parameters at the specific locality. The modeled neutron intensity profiles are evaluated against profile measurements on two different dates separated by five months, and also against time-series of neutron intensity measurements at two heights. Specifically, we test the possibility to isolate and quantify the signals from canopy interception and biomass. In addition, measurements at an agricultural field site with no biomass and at a heather field site with a smaller amount of biomass are used to underpin the assessment. To our knowledge this is the first study which





provides a quantitative analysis of the potential of using the cosmic ray technique for estimation of interception and biomass.

### 2. Field locations

Three field sites are used in this study; the primary site is Gludsted Plantation, and two secondary sites are Voulund

Farmland and Harrild Heathland. The sites are located within Skjern River Catchment in the Western part of Denmark (Figure 1) and are all part of the Danish hydrological observatory (HOBE) (Jensen and Illangasekare, 2011). The sites are situated at an elevation of approximately 50 - 60 m above sea level on an outwash plain from the last glaciation composed of nutrient depleted sandy stratified soils. Harrild Heathland is located 1 km south of Voulund Farmland, both approximately 10 km west of Gludsted Plantation.

*Figure 1 is inserted here*

Gludsted Plantation forest field site (56°04'24"N 9°20'06"E) is situated within a coniferous forest plantation covering an area of around 3500 ha. The trees of the plantation are densely planted in rows and are in general composed of Norway spruce with small patches of Sitka spruce, Larch and Douglas fir. Within the field site area (38 ha) the trees were estimated to be up to 25 m high and the dry above-ground biomass to be around 100±46 t/ha (one standard deviation) using LiDAR

images from 2006 and 2007 (Nord-Larsen and Schumacher, 2012). The dry below-ground biomass was calculated to be 25 t/ha using a root-to-shoot ratio for Norway spruce of 0.25 (Levy et al., 2004). Information on the vegetation at the forest field site is acquired from a register managed by The Danish Nature Agency (representative of the 2012 conditions); see Table 2.

*Table 2 is inserted here*

In Scandinavian forests around 79% of the total above-ground biomass of Norway spruce is stored within the tree trunks. The remaining 21% is found in the branches and needles (termed *foliage*). A typical density of the tree trunk is 0.83 g/cm$^3$ (Serup et al., 2002). The major component of the tree biomass is cellulose ($C_6H_{10}O_5$) and represents around 55% of the total mass, while the remaining 45% is vegetation water (Serup et al., 2002). Based on these approximations, the wet above- and below-ground biomass at the field site area are estimated to be 182 t/ha and 45 t/ha, respectively. With a leaf area index

(LAI) of 4.5 and a canopy interception capacity coefficient of 0.5 mm/LAI (Andreasen et al., 2013) the maximum storage of canopy intercepted rain is estimated to be 2.25 mm.

Soil samples were collected within the footprint of the cosmic-ray neutron detector on August 26 – 27, 2013 following the procedure of Franz et al. (2012). Based on these samples the organic rich litter layer is found to be 5 - 10 cm thick. The dry bulk density of the litter and mineral layer are calculated by oven drying the soil samples (Table 2), and the soil organic

matter content of the mineral soil is determined from the loss-on-ignition method (16.9% in 10 - 20 cm depth and 7.6% in 20 - 30 cm depth). A time series of soil moisture is calculated from cosmic-ray neutron intensity, starting in spring, 2013,



using the $N_0$-method as presented in Desilets et al. (2010). Lastly, the chemical composition of the soil matrix is estimated for two random soil samples collected at 20-25 cm depth using the *X-ray fluorescence* (XRF) analysis (Table 3).

*Table 3 is inserted here*

The element Gadolinium (Gd) can have a significant impact on thermal neutron intensity even at low concentrations due to its very high absorption cross-section of 49000 barns (1 barn $= 10^{-24}$ cm$^2$). The detection limit of the XRF in this study is 50 ppm for gadolinium (Gd). The two soil samples from Gludsted Plantation both have Gd concentration below the detection limit of the XRF. Inductively coupled plasma mass spectrometry (ICP-MS) detects metals and several non-metals at very small concentrations and was used to characterize the soil chemistry of a nearby field site with similar soil conditions (Salminen et al., 2005). A Gd concentration of 0.51 ppm was found at that site and we assume this value to be representative of the conditions at Gludsted Plantation.

Gludsted Plantation is a heavily equipped research field site with a 38-m high tower for measurements at multiple heights within the forest canopy. The tower is instrumented with an eddy-covariance system, humidity and air temperature sensors in addition to other sensors not used in this study (Ringgaard et al., 2011). Precipitation is measured using a tipping bucket mounted on top of an instrument container. Additionally, throughfall is measured using tipping buckets at three locations within the cosmic-ray neutron detector footprint. Four 6-m long rain gutters were placed in four directions from the tipping bucket, providing a surface area of 2.5 m$^2$.

Voulund Farmland (56°02'14"N 9°09'38"E) is an agricultural field site. In 2015, the fields were cropped with spring barley. After harvest in the late summer until ploughing in spring 2016 (prior to sowing) the fields were covered with stubble (around 10 cm high). A 25 cm layer of relatively organic rich soil (4.45% soil organic matter) is found at the top of the soil column and is a result of the cultivation practices. More information about the field site can be found in Andreasen et al. (2016).

Harrild Heathland (56°01'33"N 9°09'29"E) is a shrub land field site dominated by grasses and heather. The heathland is maintained by controlled burning, yet, the field site area has not recently been burnt. An organic rich litter layer of around 10 cm thickness is present at the top of the mineral soil and is observed visually during soil sampling field campaigns at the field site. Podsolization has resulted in a low permeability hardpan-layer at a depth of around 25-30 cm hindering percolation.



## 3. Method

### 3.1. Terminology

The energy of a neutron determines the probability of the neutron interacting with other elements and the type of interaction (i.e. absorbing or scattering). Overall, an important threshold for the behavior of low energy neutrons is present at energies

somewhere below 0.5 eV (1 eV = $1.6*10^{-19}$ J). The specific energy ranges of thermal, epithermal and fast neutrons are ambiguous. The following terminology for neutron energies is used for the purpose of this paper:

- Thermal: Energy range 0 – 0.5 eV.
- Epithermal: Energies above 0.5 eV.
- Fast: Energy range 10 - 1000 eV.

When modeling neutron transport for hydrological applications it is common to consider fast energy ranges (10 – 100 eV or 10 – 1000 ev) (Desilets et al., 2010; 2013; Rosolem et al., 2013; Franz et al., 2013; Köhli et al., 2015), while measurements using standard soil moisture neutron detectors will at best represent the entire epithermal energy range (Andreasen et al., 2016). Despite this fact, we will use the term epithermal for both measured and modeled energy ranges.

### 3.2. Cosmic-ray neutron detection

#### 3.2.1. Equipment

Cosmic-ray neutron intensity was measured using the CR1000/B system from Hydroinnova LLC, Albuquerque, New Mexico. The system has two detectors that consist of tubes filled with boron-10 (enriched to 96%) trifluoride ($^{10}BF_3$) proportional gas. The neutron detection relies on the $^{10}B(n,\alpha)^7Li$ reaction for converting thermal neutrons into charged particles ($\alpha$) and then into an electronic signal. One detector is unshielded (bare detector), while the other is shielded by 25

mm of high-density polyethylene (moderated detector). These different configurations give the bare and moderated tubes different energy sensitivities.

The thermal neutron absorption cross-section of $^{10}B$ is very high (3835 barns) (Sears et al., 1992). This absorption cross-section decreases rapidly with increasing neutron energy following a $1/E_n^{0.5}$ law (where $E_n$ is neutron energy) (Knoll 2010). Therefore, the energies measured by the bare tube comprise a continuous distribution which is heavily weighted toward

thermal neutrons (<0.5 eV), with a small proportion of epithermal neutrons also being detected (<10%) (Andreasen et al., 2016).

The moderated detector is more sensitive to higher neutron energies (> 0.5 eV). The purpose of the polyethylene is to slow (moderate) epithermal neutrons through interactions with hydrogen in order to increase the probability of them being captured by $^{10}B$ in the detector. At the same time the polyethylene attenuates the thermal neutron flux through neutron

capture by hydrogen. Nonetheless, a large proportion (approximately 40% of the thermal neutrons detected by the bare detector) originates from below 0.5 eV (Andreasen et al., 2016).



Obeying Poissonian statistics (Knoll 2010) the measurement uncertainty of a given neutron intensity, N, decreases with increasing neutron intensity and the standard deviation equals $N^{0.5}$.

The measured neutron intensities are corrected for variations in barometric pressure, atmospheric water vapor and incoming cosmic-ray intensity following procedures of Zreda et al. (2012) and Rosolem et al. (2013). Unfortunately, the water vapor
correction of Rosolem et al. (2013) is only valid for fast and epithermal neutron measurements. Since the development of correction methods is beyond the scope of this study, we refrained from using a vapor correction for the measured thermal neutron intensities. We believe that this missing correction will only have a minor effect on our results. Nevertheless, we suggest that future studies should investigate the effect of water vapor on thermal neutron intensities and to develop appropriate correction methods.

### 3.2.2.   Pure thermal and epithermal neutron detection
In order to limit the epithermal and thermal neutron contribution to the bare and the moderated detectors, respectively, we use the cadmium-difference method (Knoll, 2010; Glasstone and Edlund, 1952). The thermal absorption cross-section of cadmium is very high (approximately 3500 barns) for neutron energies below 0.5 eV. The cross-section drops to approximately 6.5 barns at neutron energy 0.5 eV and remains low with increasing neutron energies. Thus, a cadmium
shielded neutron detector only measures neutrons of energies higher than 0.5 eV. The epithermal neutron intensity was measured from a cadmium shielded moderated detector, while the thermal neutron intensity was calculated by subtracting the neutron intensity measured by the cadmium-shielded bare detector from the neutron intensity measured by the bare detector (unshielded). The cadmium-difference method is described in Andreasen et al. (2016) in detail.

Appropriate correction factor models were applied in order to obtain pure thermal and pure epithermal neutron intensity
measurements for the time periods when the cadmium-difference method was not applied (Andreasen et al., 2016). The correction factors were obtained from field campaigns applying the cadmium-difference method on bare and moderated detectors at various locations (height levels and land covers). The determination of the correction models was based on the relationships of measurements from unshielded and shielded neutron detectors (Andreasen et al., 2016).

### 3.2.3.   Footprint
The footprint of the two detectors is not expected to be the same as the properties of thermal and epithermal neutrons are very different. The footprint of the bare detector is unexplained, while the footprint of the moderated detector was determined from modeling by Desilets and Zreda (2013) and Köhli et al. (2015). However the findings of these two studies were inconsistent. Desilets and Zreda (2013) used the neutron transport code Monte Carlo N-Particle eXtended (MCNPx) and found the footprint to be nearly 600 m in diameter in dry air, while Köhli et al. (2015) using the Ultra Rapid Adaptable
Neutron-Only Simulation (URANOS) estimated the footprint to be 260 – 480 m in diameter depending on the air humidity, soil moisture and vegetation. The potential mismatch in the footprint of the bare and the moderated detectors is a concern when combining the neutron intensity measurements. In this study we will as a first approximation assume that neutron intensities measured by the two different detector types can be compared as the environmental conditions at the field sites are fairly homogeneous.



### 3.2.4. Field measurements

At Gludsted Plantation, CR1000/B systems were installed at ground level (1.5 m height) and canopy level (27.5 m height) in the spring of 2013. Hourly neutron intensities have been continuously detected (Andreasen et al., 2016) except for short periods where the detectors were used for other types of measurements or during times of malfunctions. Neutron intensity

profiles extending from the ground surface to 35-m-height above the ground were measured at approximately 5 m-increments during two field campaigns on November 28 – 29, 2013 and March 12 - 14, 2014 at Gludsted Plantation. During the field campaign on March 12 -14, 2014 an epithermal neutron intensity profile (with no thermal contribution) was measured using a cadmium-shielded moderated detector (Andreasen et al., 2016). For the profile measurements neutron intensities were recorded at a 10-minute time resolution. As the thermal neutron intensity decreases significantly with height

we choose to extend the time of measurement with the height level increments to maintain a low and consistent measurement uncertainty. The volumetric soil moisture content measured using the cosmic-ray neutron method (Zreda et al., 2008) was 0.18 during both field campaigns.

Ground level neutron intensities were measured on September 22 and 23, 2015 at Voulund Farmland (Andreasen et al., 2016). The measurements were conducted using the bare and the moderated neutron detectors normally installed at

Gludsted Plantation and data were logged every 10 minutes. In the period from October 27 to November 16, 2015 the ground level thermal and epithermal neutron intensity was measured directly at Harrild Heathland using the cadmium-difference method (Knoll, 2010). The cadmium-difference method was applied using two bare and one moderated detector normally installed at Gludsted Plantation. The neutron intensity was integrated and recorded on an hourly basis. The measurements at Voulund Farmland and Harrild Heathland will be used in the discussion of the effect of biomass and litter

on thermal and epithermal neutron intensity.

### 3.3. Neutron transport modeling

The three-dimensional Monte Carlo N-Particle transport code version 6 (MCNP6) (Pelowitz, 2013) simulating thermal and epithermal neutrons is used to model the forest site. The code holds libraries of measured absorption and scattering cross-sections used to compute the probability of interactions between earth elements and neutrons. The MCNP6 combines Monte

Carlo N-Particle Transport code version 5 (MCNP5) and Monte Carlo N-Particle Extended Radiation Transport code (MCNPX). MCNPX has been used for most neutron transport modeling within the field of hydrology (Desilets et al., 2013; Rosolem et al., 2013; Zweck et al., 2013). However, the improved and more advanced MCNP6 has recently been introduced and provided more realistic neutron intensity profiles for Voulund Farmland field site (Andreasen et al., 2016).

The number of particle histories released at the center of the upper boundary of the model domain is specified to obtain an

uncertainty below 1%. The released particles represent a distribution of high-energy particles typical for the spectrum of incoming cosmic-rays traveling through the atmosphere. The modeled neutron intensities are normalized per unit source particle providing relative values (Zweck et al., 2013). In order to obtain values comparable to measurements conversion factors are used (Andreasen et al., 2016). The conversion factors $3.739\times10^{12}$ and $1.601\times10^{13}$ are multiplied by the modeled thermal neutron fluences in the energy range of 0 – 0.5 eV and epithermal neutron fluences in the energy range 10 – 1000





eV, respectively. We stress that, the conversion factors are detector-specific as well as dependent on the horizontal area of the model setup in MCNP6. The dependence of the environmental settings is at this point in time unclear and should be addressed in future studies.

### 3.3.1. The reference model

The model domain of MCNP6 is defined by cells of varying geometry, and each cell is assigned a specific chemical composition and density. The lowest 4 m of the Gludsted Plantation reference model consists of subsurface layers. The chemical composition of the mineral soil is prescribed according to the chemical composition from XRF measurements; assumed Gd concentration of 0.51 ppm, wet below-ground biomass (cellulose) of 45 t/ha, dry bulk density of 1.09 g/cm$^3$ and soil moisture content of 0.18. The litter layer is defined according to the chemical composition of cellulose, dry bulk

density of 0.34 g/cm$^3$ and moisture content similar to that of mineral soil (see also Table 3). The same soil moisture was used for the whole soil column, as the soil moisture profile was unknown for the days of neutron profile measurements, and furthermore we wanted to test the signal of soil moisture. The atmosphere is composed of 79% nitrogen and 21% oxygen by volume and extends from the forest canopy surface to the upper boundary of the model domain at 2 km height. Here, an incoming spectrum adapted to the specific level of the atmosphere is specified (Hughes and Marsden, 1966). The density of

air is assumed to be 0.001165 g/cm$^3$. Multiple sublayers of varying vertical discretization cover the vertical extent of the model in order to record neutron intensities at multiple heights and depths from the ground surface. The resolution of the layers increases with proximity to the ground surface ranging in thickness from 0.025 m to 0.20 m for the subsurface layers and from 1 m to 164 m for the layers above the ground surface. 1 m layers are used from the ground to 28 m height to enable neutron intensity to be modeled at the measured heights. The neutron intensity detectors are layers of 1 m height and

extent the full lateral model domain (400 m x 400 m). Reflecting surfaces constrain the model domain. Thus, the particles reaching a model boundary will be reflected specularly back into the model domain. Wet above-ground biomass of 182 t/ha is distributed within the forest canopy layers extending from the ground surface to 25 m above the ground (Table 4). In order to allow for a forest canopy layer to be composed of multiple materials (cellulose and air) and densities (massive tree trunks and less dense foliage and air), the horizontal discretization of the forest canopy layers is reduced to smaller cells of

4.72 m by 4.72 m (Figure 2E).

*Table 4 and Figure 2 are inserted here*

The bole of each tree is represented by a cylinder with a diameter of 0.14 m, a composition of cellulose, and a density of 0.83 g/cm$^3$. A tree is placed at the center of each cell and extends from the ground surface to the top of the forest canopy layer and foliage is specified as a 1.7 m thick band around the tree cylinder. The foliage material is a composite of air and

cellulose and the density is the sum of the two (0.00151 g/cm$^3$). The remaining volume of the cells is composed of air. A total of 7182 trees are evenly spaced within the model domain. As previously described, the share of biomass stored in the tree trunk and the foliage is 79% and 21%, respectively, typical of Norway spruce.





### 3.3.2. Sensitivity to environmental conditions

The sensitivity of neutron intensity to forest representation is investigated by comparing the results of the reference model (Figure 2E) with the results of three alternative representations of the forest canopy (Figures 2B-D). In the first representation (Model *Foliage*; Figure 2B) the forest canopy layers is not reduced to smaller cells as a homogeneous layer

with a relatively low density material composed of cellulose and air was used to describe the forest. Here, the total density of cellulose and air is 0.00189 g/cm$^3$. In the second representation a smaller horizontal discretization of the forest canopy layers is implemented (Model *Tree trunk, Air*; Figure 2C). The setup is similar to the reference model except for the cell size and the materials included describing the forest canopy layers. Here, the cells are 4.20 m by 4.20 m and the remaining volume beyond the bole of the tree is made of air alone (density 0.001165 g/cm$^3$). For this model all biomass is stored in the

bole of the trees and the cell size is adjusted to obtain a wet above-ground biomass of 182 t/ha resulting in 9070 trees within the model domain. In the third representation the setup is equal to the reference model except that air is not included in the description of the forest canopy layers (Model *Tree trunk, Foliage*; Figure 2D). Here, the cell is divided between the bole of the tree and foliage. The foliage material is composed of cellulose and air, and the total density of the material is 0.001318 g/cm$^3$. The density of the foliage in Model *Tree trunk, Foliage* is smaller than for the reference model as the volume of the

foliage is larger and the density is reduced to obtain the same above-ground biomass as for the other models.

The reference model is used to test the sensitivity of the modeled thermal and epithermal neutron intensities to soil moisture. The soil moisture in the reference model is specified to 0.18 and both drier and wetter soils are modeled to test the sensitivity, i.e. 0.05, 0.10, 0.25, 0.35 and 0.45. The same soil moisture range is modeled using the Model *Foliage* conceptualization.

In addition to hydrogen the thermal and epithermal neutron intensity is also a product of the elemental composition and density of the soil matrix. The reference model (Model *Tree trunk, Foliage, Air)* is used to test the sensitivity of the modeled thermal and epithermal neutron intensities to soil chemistry. The reference model holds the most complex soil chemistry (fourth order complexity) with multiple subsurface layers composed of measured concentrations of major elements determined by XRF, soil organic matter, gadolinium and roots (Table 3). One component is excluded at a time to

test the effect of simplifying the soil chemistry: 1) third order complexity; soil organic matter is excluded, 2) second order complexity; soil organic matter and roots are excluded, 3) first order complexity; soil organic matter, roots and gadolinium are excluded, and 4) pure SiO$_2$; all other components are excluded.

The sensitivity of the modeled thermal and epithermal neutron intensities to the presence of the organic litter layer is investigated using the reference model, in which the thickness of the litter layer is set to be 10.0 cm. Sensitivity simulations

are carried out for the following thicknesses of the litter layer: 0.0 cm, 2.5 cm, 5.0 cm and 7.5 cm. For all litter layer models, the total thickness of the subsurface is kept constant at 4 m.

The materials of forest floor litter and mineral soil differ distinctly in terms of chemical composition and dry bulk density. The determination of dry bulk density of the two materials is characterized by measurement uncertainty, especially for the litter as sampling and drying is very challenging for materials including large amounts of soil organic matter (O'Kelly,





2004). Given that the elemental composition and density of the soil matrix is relevant for the neutron intensity the sensitivity of dry bulk density on thermal and epithermal neutron intensity is examined. The dry bulk density of the reference model is set to 0.34 g/cm$^3$ for the litter layer and 1.09 g/cm$^3$ for the mineral soil. The reference model is used to test the sensitivity applying four scenarios: 1) higher dry bulk density of the litter layer (0.50 g/cm$^3$), 2) higher dry bulk

density of the mineral soil (1.60 g/cm$^3$), 3) lower dry bulk density of the litter layer (0.20 g/cm$^3$), and 4) lower dry bulk density of the mineral soil (0.60 g/cm$^3$). All values with the exception of higher dry bulk density of 1.60 g/cm$^3$ for the mineral soil (standard value for quartz; soil particle density of 2.66 g/cm$^3$ and a porosity of 0.40) are within the range of the measurements (see Table 2).

The reference model (Model *Tree trunk, Foliage, Air)* is used to test the sensitivity to canopy interception by increasing the

density and water content of the cells described by foliage material. The forest canopy of the reference model is dry (foliage material density 0.00151 g/cm$^3$). In order to test the effect, water equivalent to 1 mm (foliage material density 0.00155 g/cm$^3$), 2 mm (foliage material density 0.00159 g/cm$^3$) and 4 mm (foliage material density 0.00167 g/cm$^3$) of canopy interception is added to the foliage volume.

The sensitivity to biomass is investigated using the reference model (Model *Tree trunk, Foliage, Air)* as well as a simplified

model-setup (Model *Foliage*). The biomass of the reference model is equivalent to a dry above-ground biomass of 100 t/ha and a dry below-ground biomass of 25 t/ha, following the root-to-shoot ratio of 0.25 typical of Norway spruce. This distribution is used for both models. For the sensitivity analysis one model without vegetation (Model *0 t/ha*, Figure 2A) and three models with different amounts of biomass are used (see Table 4). The forest canopy layer extending uniformly from the ground to 25 m above the ground surface is for the model with no vegetation assigned with the material

composition and density of air. The amount of biomass modeled for the three remaining models is equivalent to a dry above-ground biomass of: 1) 50 t/ha, 2) 200 t/ha, and 3) 400 t/ha. The size of the cells in the forest layers and the density of the foliage material are adjusted in order to obtain the correct amount of biomass.

## 4. Results and discussions

### 4.1. The reference

Neutron intensity profiles modeled with the Gludsted Plantation reference model are presented in Fig. 3, along with time-series of hourly and daily ranges of thermal and epithermal neutron intensities collected at the Gludsted Plantation during the period 2013-2015, and measured/estimated thermal and epithermal neutron intensity profiles (November 2013 and March 2014). Following the Poissonian statistics the relative uncertainty decreases with increasing neutron intensity. The relative measurement uncertainty is therefore lower for the hourly time series data than for the multi-hourly (2-12 hr) and

daily measurements. All the measurements are included in all neutron profile figures, i.e. Figs. 4 – 12, to enable comparison.

*Figure 3 is inserted here*





We choose to rely mostly on the time-series measurements, as the neutron profiles are very different despite of similar soil moisture during the time of neutron profile detection. The different neutron profiles may be a result of different climate and weather conditions related to the seasons of detections (spring and fall). Furthermore, although the area average soil moisture is the same for the two field campaigns the soil moisture profiles may be different resulting in different neutron

profile slopes and thermal-to-epithermal neutron ratios. In particular, the assumption of identical soil moisture of the litter layer and the mineral soil may be inappropriate as this was not the case during two out of three soil sampling field campaigns where the results differed considerably (soil samples were collected at 18 locations within a circle of 200 m in radius and in 6 depths from 0-30 cm depth following the procedure of Franz et al. (2012)). However, both neutron profiles are within the ranges of the daily time-series measurements and we therefore still believe that they can be used in the

assessment of the modeled neutron profiles. For future studies we recommend soil sample field campaigns to be conducted on the days of neutron profile measurements.

A remarkable agreement between measured and modeled neutron intensities is seen in Fig. 3. We stress that no calibration of the governing physical properties in the forest model is performed and that the estimates are based on measured properties. The modeled thermal neutron intensity profiles are especially consistent with measurements, as both modeled

ground and canopy neutron intensity are within the daily measurement ranges. In contrast, the modeled epithermal neutron profile is slightly underestimated and the profile slope is steeper than the measured profiles. Nevertheless, the modeled epithermal neutron intensity profile is still within the ranges of the time-series of hourly measurements at both height levels. To investigate whether the slight misfit of measurements and modeling may be due to misrepresentations in litter and mineral soil layer thickness, density and composition, forest canopy conceptualization, canopy interception and soil

moisture a sensitivity analysis of these parameters and variables is conducted.

### 4.2.    Forest conceptualization

The thermal and epithermal neutron intensity profiles modeled using the Gludsted Plantation reference model (Model *Tree trunk, Foliage, Air*) and three models with other forest canopy conceptualizations are presented in Fig. 4.

*Figure 4 is inserted here*

All modeled neutron intensity profiles are within the range of hourly time-series measurements, and in particular the thermal neutron profiles are modeled satisfactorily. The ground and canopy level thermal neutron intensity of models with forest canopy conceptualization of Model *Tree trunk, Foliage* and Model *Tree trunk, Foliage, Air* are within the daily ranges of the time-series measurements. Overall, the models of the more complex forest canopy conceptualizations, including a tree trunk, provide similar thermal and epithermal neutron profiles. The neutron intensity profiles of the simpler

forest canopy conceptualization of Model *Foliage* is less steep and is the only model providing an epithermal neutron intensity profile within the daily ranges of the time-series measurements at both the ground and canopy level.

The sensitivity of forest canopy conceptualization on thermal and epithermal neutron intensities is quantified at the ground and canopy level relative to the reference model (Table 5).





*Table 5 is inserted here*

The most appropriate forest canopy conceptualization is not obvious from Fig. 4 and Table 5 as the best fit of the thermal measurements is found using a complex conceptualization, while the more simple foliage conceptualization matches the epithermal measurements better. We can, however, conclude that the neutron transport at the ground-atmosphere interface is

highly sensitive to the level of complexity of the forest canopy conceptualization. For the following analysis the most complex model was chosen for the sensitivity analysis, although some examples of modeling using the simplest forest canopy conceptualization will be provided.

The modeled thermal and epithermal neutron intensity profiles of the Gludsted Plantation *reference model* and Model *Foliage* using six different soil moistures, 0.05, 0.10, 0.18, 0.25, 0.35 and 0.45, are presented in Figs. 5 and 6, respectively.

*Figure 5 and Figure 6 are inserted here*

As expected, the thermal and epithermal neutron intensity is seen in Figs. 5 and 6 to decrease with increasing soil moisture. For both model-setups, the largest changes in neutron intensity occur at the dry end of the soil moisture range and for the epithermal neutrons (see also Table 5). For the reference model (Figure 5), only a minor decrease in the sensitivity of soil moisture on epithermal neutron intensity is observed going from ground level to canopy level (approximately 15%

reduction in intensity range corresponding to a soil moisture change of 0.40). On the other hand, the sensitivity of the thermal neutron intensity is reduced more than 50% (Table 5) most likely caused by the lower mean-free path length of the thermal neutrons compared to that of epithermal neutrons. The response to soil moisture is similar for the model with a simple forest canopy conceptualization (Figure 6). However, both thermal and epithermal neutron intensities are found to be slightly more sensitivity to soil moisture.

**4.3.    Subsurface properties**

Thermal and epithermal neutron intensity profiles modeled using the reference model (with *fourth order complexity*) and models of decreasingly complex soil chemistry are presented in Fig. 7.

*Figure 7 is inserted here*

The effect of varying the soil chemistry on thermal and epithermal neutron intensity profiles is small at Gludsted Plantation.

The exact sensitivity of the different components (soil organic matter, gadolinium, roots and major elements relative to a simple soil chemistry of $SiO_2$) on ground and canopy level thermal and epithermal neutron intensity is quantified in Table 5 (see values in parentheses). Only the removal of soil organic matter changes the neutron intensity significantly, i.e. an increase in the ground level thermal and epithermal neutron intensity of 19 cts/hr and 25 cts/hr, respectively, is observed. The sensitivity to soil chemistry on thermal and epithermal neutron intensity profiles was found to be much more substantial

at Voulund Farmland (Andreasen et al., 2016). The soil organic matter content at Voulund Farmland is smaller and the soil chemistry is, except from a few elements (added in relation to farming activities; spreading of manure and agricultural lime), similar to Gludsted Plantation. Modelling shows that the sensitivity to soil chemistry at Gludsted Plantation is





dampened by the considerable amount of hydrogen present in the forest biomass and the litter at the forest floor (not presented here).

In Fig. 8, the thermal and epithermal neutron intensity profiles modeled for a forest with litter layer of various thicknesses are presented. The Gludsted Plantation reference model with a 10.0 cm thick litter layer is used along with forest models

with litter layers of 0.0 cm, 2.5 cm, 5.0 cm and 7.5 cm thickness.

*Figure 8 is inserted here*

Neutron intensities are found to decrease with an increasing layer of litter, having the greatest impact on the epithermal neutron intensities. The considerable amount of hydrogen in litter causes the probability of scattering of neutrons travelling through the subsurface to increase with increasing amounts of litter. Thereby, the thermal-to-epithermal neutron intensity

ratio is found to be altered when changing the thickness of the litter layer. This effect is most pronounced when the model without a litter layer is compared to the model with just a thin 2.5 cm thick litter layer (see also Table 5).

Thermal and epithermal neutron intensity profiles modeled using the Gludsted Plantation reference model and models of altered bulk densities of subsurface layers are provided in Fig. 9.

*Figure 9 is inserted here*

The modified bulk densities of litter and mineral soil only provided slight changes in thermal and epithermal neutron intensities. Nevertheless, a reverse response of changed bulk densities is observed. A decrease in neutron intensity is obtained both by increasing the bulk density of the litter material and decreasing the bulk density of the mineral soil. Conversely, higher neutron intensities are computed by decreasing the bulk density of the litter material and increasing the bulk density of the mineral soil. Thus, here the mineral soil acts as a producer of thermal and epithermal neutrons, while the

litter acts as an absorber.

### 4.4.      Canopy interception

The thermal and epithermal neutron intensity profiles modeled by the Gludsted Plantation reference model with a dry forest canopy (model: *Dry canopy*) and models of 1 mm, 2 mm and 4 mm of canopy interception are presented in Fig. 10.

*Figure 10 is inserted here*

Except for a slight increase in ground level thermal neutron intensities with wetting of the forest canopy, no effect of canopy interception on neutron intensity is observed in Fig. 10. A maximum change of approximately 3% (15 cts/hr) is observed for thermal neutron intensity at ground level going from a dry canopy to 4 mm of canopy interception. At the specific field site a maximum canopy storage capacity of 2.25 mm is expected, producing a change in observed ground level thermal neutron intensity of approximately 7 cts/hr. Given an average neutron intensity of 504 cts/hr of ground level thermal neutrons with

the installed detectors, an uncertainty of 22 cts/hr is expected based solely on Poissonian statistics. In order to obtain a signal-to-noise ratio of 1, either an 11-hour-integration time or 11 detectors similar to the installed are needed. However,




longer integration times are not appropriate when considering Gludsted Plantation as the return time of canopy interception (cycling between precipitation and evaporation) often is short (half-hourly to hourly time resolution).

Although detection of canopy interception at Gludsted Plantation is unfavorable it may still be possible at more appropriate locations. Canopy interception modeling as described above is therefore also performed for soil moisture 0.05, 0.10, 0.25

and 0.40. Ground level thermal-to-epithermal neutron ratios of the 20 model combinations are plotted against ground level thermal neutron intensity, ground level epithermal neutron intensity and volumetric soil moisture (Figure 11). We choose not to include measurement in the figure as we from the calculation in the previous section found the measurement uncertainty at a relevant integration time to be greater than the signal of canopy interception.

*Figure 11 is inserted here*

Overall, ground level thermal-to-epithermal neutron ratio is found to be independent of ground level thermal neutron intensity (Figure 11A), ground level epithermal neutron intensity (Figure 11B) and volumetric soil moisture (Figure 11C). Ground level thermal-to-epithermal neutron ratio is found to increase with increasing canopy interception. The ground level thermal-to-epithermal neutron ratio for a dry canopy is on average 0.804, while the average at 4 mm of canopy interception is 0.836. Overall, the same increase in ground level thermal-to-epithermal neutron ratio is obtained per 1 mm additional

canopy interception. Although the change in the ratio with wetting/drying of the forest canopy is small the canopy interception may potentially be measured using cosmic-ray neutron intensity detectors at locations with: 1) a high neutron intensity level (lower latitude and/or higher altitude, 2) more sensitive neutron detectors, and 3) greater amounts of canopy interception with longer residence time (e.g. snow). We suggest future studies investigating the effect of canopy interception on the neutron intensity signal to be performed at locations matching one or more of these criteria.

**4.5.    Biomass**

The sensitivity to the amount of forest biomass on thermal and epithermal neutron intensity profiles using the forest canopy conceptualization of Model *Tree trunk, Foliage, Air* (reference model) and Model *Foliage* are presented in Figs. 12 and 13, respectively. The neutron intensity profiles are provided for a scenario with no vegetation, the Gludsted Plantation reference model (model: *100 t/ha*) and models with biomass equivalent to dry above-ground biomass of: 50 t/ha, 200 t/ha and 400

t/ha. In order to calculate the relative changes listed in Table 5, the model with biomass equivalent to 100 t/ha dry above-ground biomass with the same forest canopy conceptualization is used.

*Figure 12 and Figure 13 are inserted here*

Forest biomass is seen to significantly alter the thermal and epithermal neutron intensity profiles both with regards to the differences between ground and canopy level, and ground level thermal-to-epithermal neutron intensity ratios (Figures 12

and 13). The direction and magnitude of these changes are found to be rather different depending on the two forest canopy conceptualizations. For the Model *Tree trunk, Foliage, Air* the increase in biomass results in an increase in thermal neutron intensity (Figure 12A) while the epithermal neutron intensity decreases (Figure 12B). This effect is almost constant up to an




elevation of 20 m, but decreases sharply near the top of the forest canopy. Increasing the biomass in the Model *Foliage* from 0 t/ha to 50 t/ha (Figure 13) results in a considerable increase in ground level thermal neutron intensity (136 cts/hrs, Table 5) while at canopy level thermal neutron intensity is almost unaltered. A further increase in biomass (>50 t/ha) decreases both ground and canopy level thermal neutron intensities. This decrease is greatest at canopy level resulting in a less steep

profile slope for models with larger quantities of biomass (Figure 13A). The epithermal neutron intensity decreases at ground level and increase proportionally at canopy level with increasing amounts of biomass (Figure 13B).

As shown in Figs. 4, 12 and 13 the resulting thermal and epithermal neutron intensity profiles depend highly on the chosen model-setup (forest conceptualization). At this stage, we cannot determine which conceptualization is more realistic, and we therefore choose to use both conceptualizations in the further analysis.

Figure 14 presents measured and modeled difference in ground and canopy level thermal neutron intensity (Figure 14A – 14C), and ground and canopy level epithermal neutron intensity (Figure 14D –14F), respectively, for different amounts of forest biomass when the reference forest canopy conceptualization is used (Model *Tree trunk, Foliage, Air*). These differences are plotted against ground level thermal neutron intensity (Figures 14A and 14D), ground level epithermal neutron intensity (Figures 14B and 14E), and volumetric soil moisture estimated using the $N_0$-method (Desilets et al., 2010)

(Figures 14C and 14F). In the modeling we have varied both the soil moisture (six values from dry to saturation) and the biomass (five different values from 0 t/ha to 400 t/ha), resulting in a total of 30 combinations. The measurements are provided as daily averages, biweekly averages and as a total average of the two-year-period.

*Figure 14 is inserted here*

The effect of forest biomass is apparent considering the modeled differences between ground and canopy level thermal

neutron intensity against ground level epithermal neutron intensity (Figure 14B) and volumetric soil moisture (Figure 14C). Overall, the difference increases with greater amounts of biomass, with the most substantial change occurring from 0 t/ha to 50 t/ha. A similar positive correlation is observed between the difference in ground and canopy level epithermal neutron intensity and biomass. However, here the relationship exists when the differences are plotted against ground level thermal neutron intensity (Figure 14D) and volumetric soil moisture (Figure 14F).

Overall, the measured and modeled differences in ground and canopy level thermal and epithermal neutron intensities are of the same magnitude. However, the measurements do not fall onto a modeled curve representing a constant biomass value and the model underestimates the measured differences. The mean measured difference in ground and canopy level epithermal neutron intensity (two-year-period) is similar to the difference modeled for a forest of 400 t/ha dry above-ground biomass when related to ground level thermal neutron intensity (Figure 14D) and volumetric soil moisture (Figure 14F).

Considering biweekly averages the measured differences are very variable and corresponds both to models of very low (close to no vegetation) and very high amounts of biomass (> 400 t/ha dry above-ground biomass).





In Fig. 15 are the results for Model *Foliage* presented. Measured and modeled difference in ground and canopy level thermal neutron intensity (Figures 15A and 15C), and epithermal neutron intensity (Figures 15D – 15F) are shown for different amounts of forest biomass. Similar to Fig. 14, the differences are plotted against ground level thermal neutron intensity (Figures 15A and 15D), ground level epithermal neutron intensity (Figures 15B and 15E), and volumetric soil

moisture estimated using the $N_0$-method (Desilets et al., 2010) (Figures 15C and 15F).

*Figure 15 is inserted here*

Again a positive correlation is found between the differences between ground and canopy level neutron intensities and the amount of biomass. The difference in neutron intensity is increased compared to Fig. 14, and all six subplots in Fig. 15 have distinct relations for each biomass value. Compared to the results of Model *Tree trunk, Foliage, Air* (Figure 14), the

modeled differences in ground and canopy level neutron intensity are overestimated, however, only slightly for epithermal neutron intensity plots (Figures 15D – 15F). Here, the total average of the entire two year measurement period is found to agree reasonably with modeling and the range of measured differences (biweekly averages) is within the modeled differences provided by models of forest biomass equivalent to 0 t/ha to 100 t/ha dry above-ground biomass. The major change in differences between ground and canopy level thermal neutron intensity is seen to occur between the model with

no biomass and the forest with 50 t/ha dry above-ground biomass (around 150 cts/hr). Only minor changes in the differences occur with increasing biomass above 50 t/ha, i.e. the addition of 350 t/ha dry above-ground biomass (from 50 t/ha to 400 t/ha dry above-ground biomass) only increases the difference in thermal neutron intensity with around 40 cts/hr.

One can also potentially use the thermal-to-epithermal ratio at the ground level to assess biomass. The advantage is that only one station is needed - and that at a convenient location. This would also allow for surveys of biomass estimations to

be conducted from mobile cosmic-ray neutron intensity detector systems, e.g. installed in vehicles. As stated previously, we consider combined measurements of thermal and epithermal neutron intensities to be appropriate at Gludsted Plantation due to reasonably lateral homogeneity in soil, litter and vegetation prevails at the field site.

The measured and modeled ratios are again provided using both forest canopy conceptualization, i.e. Model *Tree trunk, Foliage, Air* (Figure 16) and Model *Foliage* (Figure 17). The ratios are plotted against A) ground level thermal neutron

intensity, B) ground level epithermal neutron intensity, and C) soil moisture estimated using the $N_0$-method (Desilets et al., 2010). Like before, measurements are provided as daily averages, biweekly averages and as a total average of the whole two-year-period.

*Figure 16 and Figure 17 are inserted here*

The modeled thermal-to-epithermal ratio increases with forest biomass (Figures 16 and 17). The ratios are found to be

independent of changes in the ground level thermal neutron intensity, the ground level epithermal neutron intensity and volumetric soil moisture. However, this independence is not seen in the measurements, where the ground level epithermal neutron intensity and soil moisture (Figures 16C and 17C) in particular seem to impact the ratio. A fairly proportional



increase in the ground level thermal-to-epithermal ratio with respect to greater amounts of biomass is found when using the reference conceptualization of Model *Tree trunk, Foliage, Air* (Figure 16). Contrarily, when using Model *Foliage* (Figure 17), a more uneven increase in the ratio with increasing amounts of biomass is provided. A major increase in the ground level thermal-to-epithermal neutron ratio of around 0.22 appears from no vegetation to a dry above-ground biomass of 50

t/ha. However, additional amounts of biomass only increase the ground level thermal-to-epithermal ratio slightly. With additional 350 t/ha biomass (from 50 t/ha to 400 t/ha dry above-ground biomass) the ratio increases by only 0.05 cts/hr.

A remarkably fit of measurements and modeling can be seen in Fig. 16. The two-year-average measurement is consistent with the reference model estimate of 100 t/ha dry above-ground biomass and the biweekly averages of measurements are all within the ratios modeled for biomass of 50 t/ha - 200 t/ha. For the Model *Foliage* in Fig. 17, the two-year-average of the

measured ratios corresponds to approximately the modeled value of 50 t/ha dry above-ground biomass. Moreover, the biweekly averages of the measurements exceed the lower and upper boundary of ratios provided by the models of 50 t/ha and 400 t/ha dry above-ground biomass.

Contrary to the modeling results, our measurements suggest a dependence of ground level thermal-to-epithermal neutron ratio to soil moisture changes. The discrepancy of measurements and modeling could be related to: 1) shortcomings in the

model setup, i.e. a need for an even more realistic forest conceptualization, and more detailed and up-to-date forest information, 2) discrepancy of measured and modeled energy ranges as discussed in Andreasen et al. (2016), and 3) unrepresentative biomass estimate. The 100 t/ha dry above-ground biomass was estimated using LiDAR images from 2006 and 2007 and therefore not completely representative of the 2013-2015 conditions (because of tree growth). Furthermore, the biomass estimate varied considerably within the image (standard deviation = 46 t/ha), and the image coverage did not

fully match the footprint of the cosmic-ray neutron intensity detector.

The ground level thermal-to-epithermal neutron intensity is compared with two additional field sites close to Gludsted Plantation. The three field sites have similar environmental settings (e.g. neutron intensity, soil chemistry), though different land covers (stubble pasture and heathland).

At Voulund Farmland the ground level thermal-to-epithermal ratio was measured to be 0.53 and 0.58 on September 22[nd] and

September 23[rd] 2015, respectively. Only minor amounts of organic matter were present in the stubble and residual of spring barley harvested in August 2015. Additionally, the ground level thermal-to-epithermal ratio was determined based on modeling of bare ground and site specific soil chemistry measured at Voulund Farmland (Andreasen et al., 2016). The modeled ratio was found to be 0.56 in agreement with the measured ratios. The ratio modeled based on the non-vegetated conceptualization of Gludsted Plantation was slightly higher (0.60, see Figures 16 and 17). Here, a 10 cm thick litter layer

was included in the model. The sensitivity analysis on the effect of litter layer on neutron intensity (Figure 8 and Table 5) implies that lower thermal-to-epithermal neutron intensities are found at locations with a thin or no litter layer.

The ground level thermal-to-epithermal neutron ratio at the Harrild Heathland was measured to 0.66 during the period October 27 to November 16 2015. The ratio is slightly higher than the non-vegetated model for Gludsted Plantation, yet,



both a considerable layer of litter and some amount of biomass in the form of grasses, heather plants and bushes are present at the Harrild Heathland. The slightly higher ratio at Harrild Heathland relative to the non-vegetated Gludsted Plantation may be due to this smaller amount of biomass. At Gludsted Plantation, the ratio is 0.73 for dry above-ground biomass equivalent of 50 t/ha. Accordingly, the ratio measured at Harrild Heathland is somewhere in between the ratio modeled for a

non-vegetated field site and a field site with biomass equivalent to 50 t/ha dry above-ground biomass.

The modeled decrease in ground level thermal-to-epithermal ratios with smaller amounts of biomass are in line with the measurements conducted at the three field sites of similar soil chemistry and dissimilar land covers in terms of litter and vegetation.

Detecting the ground level thermal-to-epithermal neutron ratio at locations of known biomass should be accomplished to

test the suggested relationship obtained using the forest canopy conceptualization of Model *Tree trunk, Foliage, Air*. We recommend a detection system with higher sensitivity to be used when a location of low neutron intensity rates (like Gludsted Plantation) is surveyed, unless long periods of measurements can be conducted at each measurement location. This can be accomplished by using larger sensors, an array of several sensors and/or sensors that are more efficient, as is done in roving surveys (Chrisman and Zreda, 2013; Franz et al., 2015).

**5. Conclusion**

The potential of applying the cosmic-ray neutron intensity method for other purposes than soil moisture detection was explored using profile and time-series measurements of neutron intensities combined with neutron transport modeling. The vegetation and subsurface layers of the forest model-setup were described by average measurements and estimates and a remarkable agreement was found for measured and modeled thermal and epithermal neutron intensity profiles without

adjusting parameters and variables. Following, a sensitivity analysis was performed to quantify the effect of the forests governing parameters/variables on the neutron transport profiles.

The ground level thermal-to-epithermal neutron ratio was found to increase with increasing amounts of canopy interception and to be independent of ground level thermal neutron intensity, ground level epithermal neutron intensity and soil moisture. However, the increase was minor and the measurement uncertainty exceeds the signal of canopy interception at a

timescale appropriate to detect canopy interception at Gludsted Plantation (half-hour to hourly). However, the signal of canopy interception can potentially be isolated in measurements from locations of higher neutron intensities (lower latitudes and/or higher altitudes) with canopy interception of longer residence time and larger storage capacity (e.g. snow). After soil moisture, the next most important variables affecting neutron intensity profiles were the thicknesses of the litter layer and the amount of above-ground biomass. An increased litter layer at the forest floor resulted in reduced neutron intensities,

particularly for epithermal neutrons. Increased amounts of forest biomass altered the thermal and epithermal neutron intensity profiles significantly. Both the difference between ground and canopy level thermal and epithermal neutron intensity, respectively, and the ground level thermal-to-epithermal neutron ratios were changed with additional amounts of biomass. The best agreement between measurements and modeling was obtained for the ground level thermal-to epithermal




neutron ratio using a model with a complex forest canopy conceptualization. Furthermore, the modeled ratios were found to agree well with two nearby field sites with different land covers (a bare ground agricultural field and a heathland field site).

### 6. Acknowledgements

We acknowledge The Villum Foundation (www.villumfonden.dk) for funding the HOBE project (www.hobe.dk). Lars M. Rasmussen and Anton G. Thomsen (Aarhus University) are greatly thanked for the extensive help in the field. We would like to extend our gratitude to Vivian Kvist Johannsen and Johannes Schumacher from the Section for Forest, Nature and Biomass, University of Copenhagen. Finally, we also acknowledge the NMDB database (www.nmdb.eu), founded under the European Union's FP7 programme (contract no. 213007) for providing data. Jungfraujoch neutron monitor data were kindly provided by the Cosmic Ray Group, Physikalisches Institut, University of Bern, Switzerland.

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





**Tables**

Table 1 – Dynamics of different hydrogen pools.

|  | Static | Quasi-static | Transient |
|---|---|---|---|
| Soil moisture |  |  | x |
| Tree roots |  | x |  |
| Soil organic matter |  | x |  |
| Water in soil minerals | x |  |  |
| Vegetation (cellulose, water) |  | x | x |
| Snow |  | x | x |
| Puddles |  |  | x |
| Open water (river, sea, lake) |  | x |  |
| Canopy intercepted water |  |  | x |
| Buildings/roads | x |  |  |
| Atmospheric water vapor |  |  | x |

Table 2 – Average tree height, tree diameter and dry bulk density (bd$_{dry}$) of the litter layer and the mineral soil at Gludsted Plantation field site. Tree height and diameter are representative of conditions for year 2012.

|  | Average | Standard deviation | Max. | Min. |
|---|---|---|---|---|
| Tree height* [m] | 11 | 6 | 25 | 3 |
| Tree diameter* [m] | 0.14 | 0.08 | 0.34 | 0.03 |
| Dry bulk density litter layer, [g cm$^{-3}$] | 0.34 | 0.29 | 1.09 | 0.09 |
| Dry bulk density mineral soil, [g cm$^{-3}$] | 1.09 | 0.28 | 1.53 | 0.22 |

* Data obtained from the Danish Nature Agency





Table 3 – Chemical composition of major elements at Gludsted Plantation determined using X-ray fluorescence analysis on soil samples collected in 0.20-0.25 m depth.

|  | Gludsted Plantation [%] |
| --- | --- |
| O | 52.78 |
| Si | 44.86 |
| Al | 1.54 |
| K | 0.53 |
| Ti | 0.29 |

Table 4 – Forest properties used in modeling.

5    *Specific for model with forest conceptualization of Model *Tree trunk, Foliage, Air*. **Reference model.

|  |  | Models | | | |
| --- | --- | --- | --- | --- | --- |
|  | No vegetation | 50 t ha$^{-1}$ | 100 t ha$^{-1}$** | 200 t ha$^{-1}$ | 400 t ha$^{-1}$ |
| Dry above-ground biomass [t ha$^{-1}$] | 0 | 50 | 100 | 200 | 400 |
| Wet above-ground biomass [t ha$^{-1}$] | 0 | 91 | 182 | 364 | 727 |
| Dry below-ground biomass [t ha$^{-1}$] | 0 | 12.5 | 25 | 50 | 100 |
| Wet below-ground biomass [t ha$^{-1}$] | 0 | 23 | 45 | 91 | 182 |
| Tree trunk density [g cm$^{-3}$] * | - | 0.83 | 0.83 | 0.83 | 0.83 |
| Tree trunk radius [m] * | - | 0.07 | 0.07 | 0.07 | 0.07 |
| Tree height [m] * | - | 25 | 25 | 25 | 25 |
| Foliage density [g cm$^{-3}$] * | - | 0.00134 | 0.00151 | 0.00185 | 0.00255 |
| Foliage band [m] * | - | 2.44 | 1.70 | 1.18 | 0.82 |
| Sub-cell size [m x m] * | - | 6.67 x 6.67 | 4.72 x 4.72 | 3.34 x 3.34 | 2.36 x 2.36 |





Table 5 – Sensitivity in modeled ground level (1.5 m) and canopy level (27.5 m) thermal neutron intensity and epithermal

neutron intensity due to (1) forest conceptualization, (2) soil moisture, (3) soil chemistry, (4) litter layer thickness, (5)

mineral soil and litter bulk density, (6) canopy interception and (7) biomass. The sensitivity is relative to the simulations

based on the *reference model* given in Fig. 3 and Model *Foliage* given in Fig. 4, respectively. Values provided in

5    parentheses specifies the direct effect of one-by-one excluding soil organic matter, Gd, below ground biomass and site

specific major elements soil chemistry.* Reference, in absolute values.

| | | Thermal 1.5 m | Thermal 27.5 m | Epithermal 1.5 m | Epithermal 27.5 m |
|---|---|---|---|---|---|
| Conceptualization models (Fig. 4) | Tree trunk, Air, Foliage | 504* | 257* | 623* | 717* |
| | Foliage | 70 | -50 | 58 | 113 |
| | Tree trunk, Air | -20 | 15 | -13 | -22 |
| | Tree trunk, Foliage | 32 | 4 | -4 | -1 |
| Soil moisture models (Fig. 5) | 0.18 | 504* | 257* | 623* | 717* |
| Model *Tree trunk, Air, Foliage* | 0.05 | 100 | 47 | 131 | 109 |
| | 0.10 | 45 | 20 | 58 | 50 |
| | 0.25 | -25 | -12 | -27 | -23 |
| | 0.35 | -47 | -22 | -53 | -45 |
| | 0.45 | -59 | -28 | -69 | -59 |
| Soil moisture models (Fig. 6) | 0.18 | 573* | 207* | 681* | 813* |
| Model *Foliage* | 0.05 | 119 | 40 | 142 | 115 |
| | 0.10 | 56 | 18 | 68 | 53 |
| | 0.25 | -27 | -9 | -30 | -23 |
| | 0.35 | -50 | -16 | -55 | -48 |
| | 0.45 | -64 | -21 | -74 | -61 |
| Soil chemistry models (Fig. 7) | 4$^{th}$ order complexity | 504* | 257* | 623* | 717* |
| Model *Tree trunk, Air, Foliage* | 3$^{rd}$ order complexity | 19 (+19) | 8 (+8) | 25 (+25) | 14 (+14) |
| | 2$^{nd}$ order complexity | 18 (-1) | 9 (+1) | 27 (-2) | 17 (+3) |
| | 1$^{st}$ order complexity | 22 (+4) | 10 (+1) | 26 (-1) | 18 (+1) |
| | $SiO_2$ | 27 (+5) | 11 (+1) | 23 (-3) | 19 (+1) |
| Litter layer models (Fig. 8) | 10.0 cm | 504* | 257* | 623* | 717* |
| Model *Tree trunk, Air, Foliage* | 7.5 cm | 11 | 4 | 26 | 22 |
| | 5.0 cm | 18 | 9 | 53 | 41 |
| | 2.5 cm | 24 | 12 | 85 | 71 |
| | No litter layer | 22 | 17 | 131 | 113 |
| Density models (Fig. 9) | Gludsted Plantation* | 504* | 257* | 623* | 717* |
| Model *Tree trunk, Air, Foliage* | Higher bd$_{litter\ layer}$ | -7 | -5 | -10 | -6 |





|  |  |  |  |  |  |
|---|---|---|---|---|---|
|  | Higher bd$_{mineral soil}$ | 15 | 5 | 17 | 10 |
|  | Lower bd$_{litter layer}$ | 7 | 2 | 14 | 10 |
|  | Lower bd$_{mineral soil}$ | -26 | -13 | -22 | -18 |
| Canopy interception models (Fig. 10) | Dry canopy | 504* | 257* | 623* | 717* |
| Model *Tree trunk, Air, Foliage* | 1 mm | 4 | -2 | -3 | 0 |
|  | 2 mm | 7 | -3 | -5 | 5 |
|  | 4 mm | 15 | -7 | -5 | 2 |
| Biomass models (Fig. 11) | 100 t ha$^{-1}$ | 504* | 257* | 623* | 717* |
| Model *Tree trunk, Air, Foliage* | No vegetation | -67 | -21 | 99 | 85 |
|  | 50 t ha$^{-1}$ | -16 | -8 | 45 | 33 |
|  | 200 t ha$^{-1}$ | 14 | 2 | -70 | -47 |
|  | 400 t ha$^{-1}$ | 21 | 2 | -172 | -116 |
| Biomass models (Fig. 12) | 100 t ha$^{-1}$ | 573* | 207* | 681* | 813* |
| Model *Foliage* | No vegetation | -136 | 29 | 41 | -28 |
|  | 50 t ha$^{-1}$ | 0 | 24 | 13 | -23 |
|  | 200 t ha$^{-1}$ | -9 | -32 | -26 | 22 |
|  | 400 t ha$^{-1}$ | -48 | -59 | -82 | 73 |





**Figures**

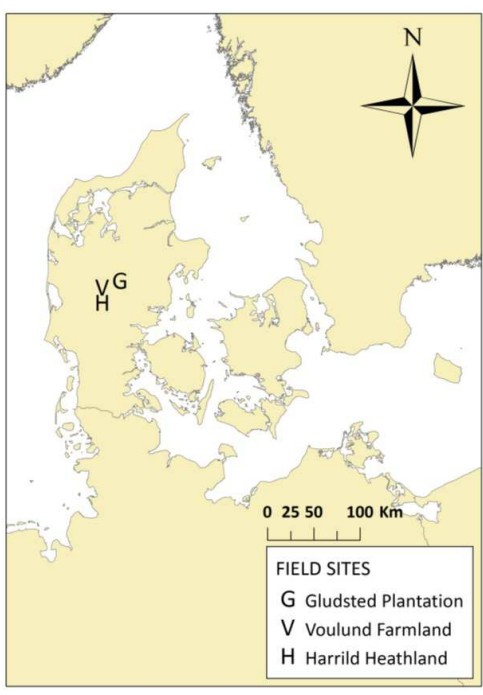

Figure 1 – Map showing the location of the three field sites; Gludsted Plantation, Voulund Farmland and Harrild Heathland.



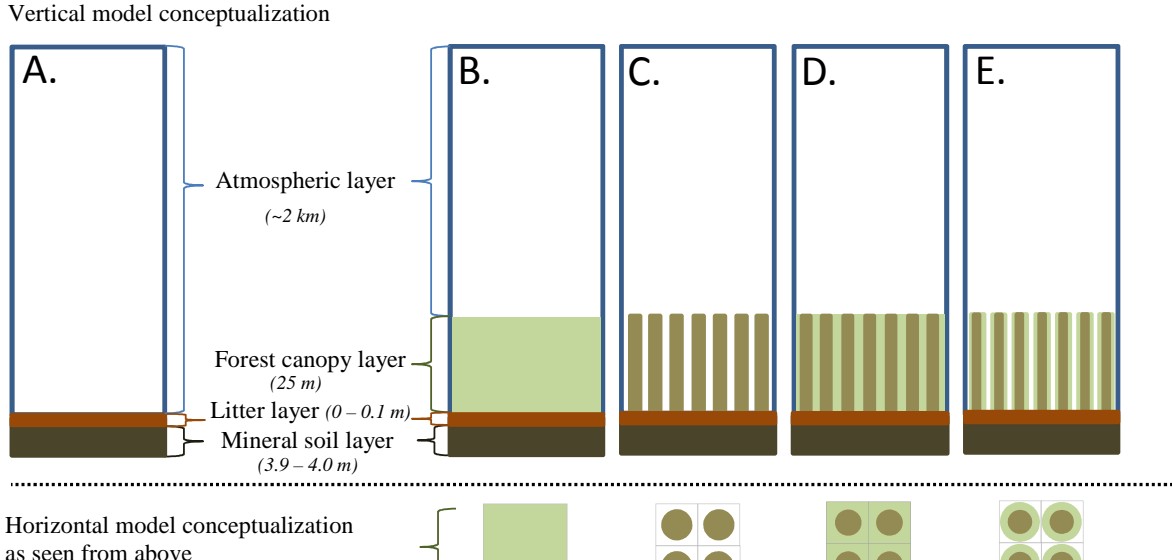

Figure 2 – Model conceptualizations of forest. A: no forest canopy layer (model name: *0 t ha$^{-1}$*); B: homogeneous foliage

layer with a uniformly distributed biomass (model name: *Foliage*); C: cylindrical tree trunks with air in between (model

name: *Tree trunks, Air*); D: cylindrical tree trunks with foliage in between (model name: *Tree Trunks, Foliage*); E:

5    cylindrical tree trunks enveloped in a foliage-cover with air in between (model name: *Tree trunks, Foliage, Air*). The bottom

four figures illustrate the forest conceptualization seen from above.



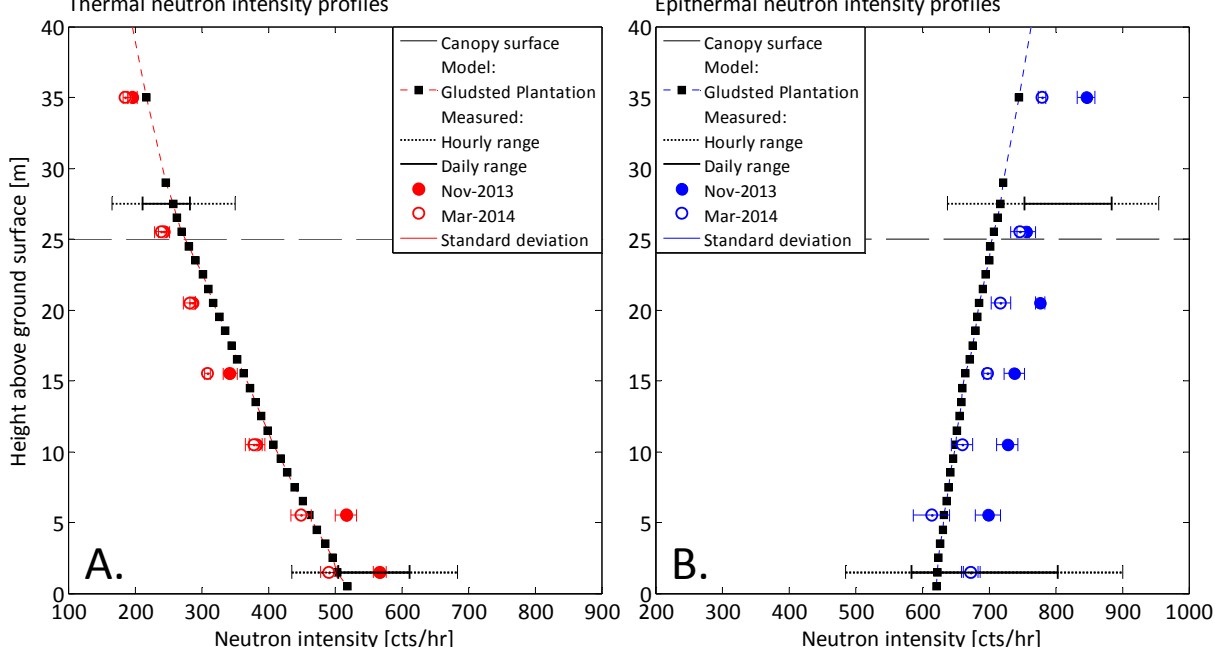

Figure 3 – Gludsted Plantation reference model. Measured and modeled (A.) thermal and (B.) epithermal neutron intensity profiles. Hourly and daily ranges of variation of thermal and epithermal neutron intensities at ground and canopy level for the period 2013–2015.




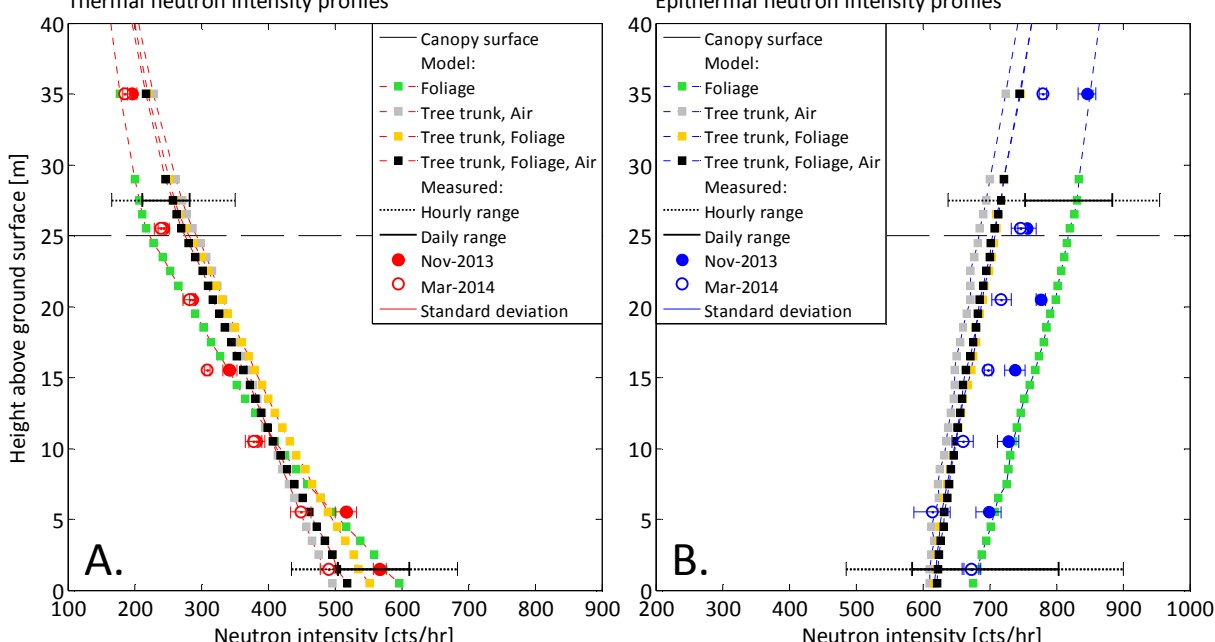

Figure 4 – Sensitivity to forest canopy conceptualization. Measured and modeled (A.) thermal and (B.) epithermal neutron intensity profiles at Gludsted Plantation. Hourly and daily ranges of variation of thermal and epithermal neutron intensities at ground and canopy level for the period 2013–2015.





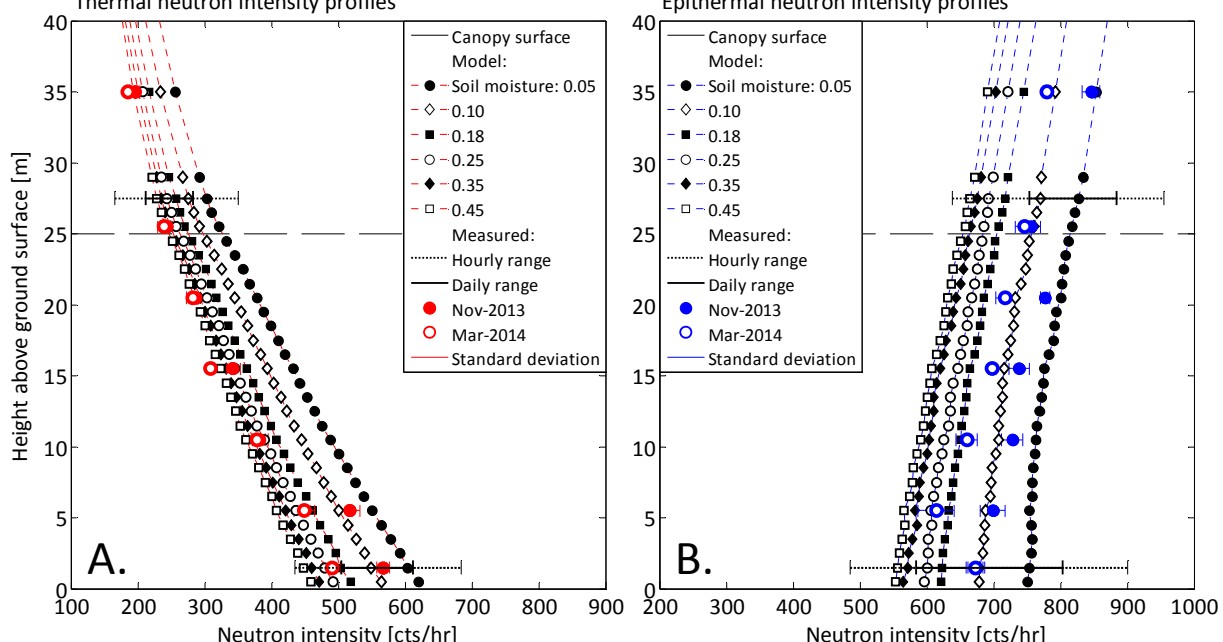

Figure 5 – Sensitivity to soil moisture (Model *Tree trunk, Foliage, Air*). Measured and modeled (A.) thermal and (B.) epithermal neutron intensity profiles at Gludsted Plantation. Hourly and daily ranges of variation of thermal and epithermal neutron intensities at ground and canopy level for the period 2013–2015.





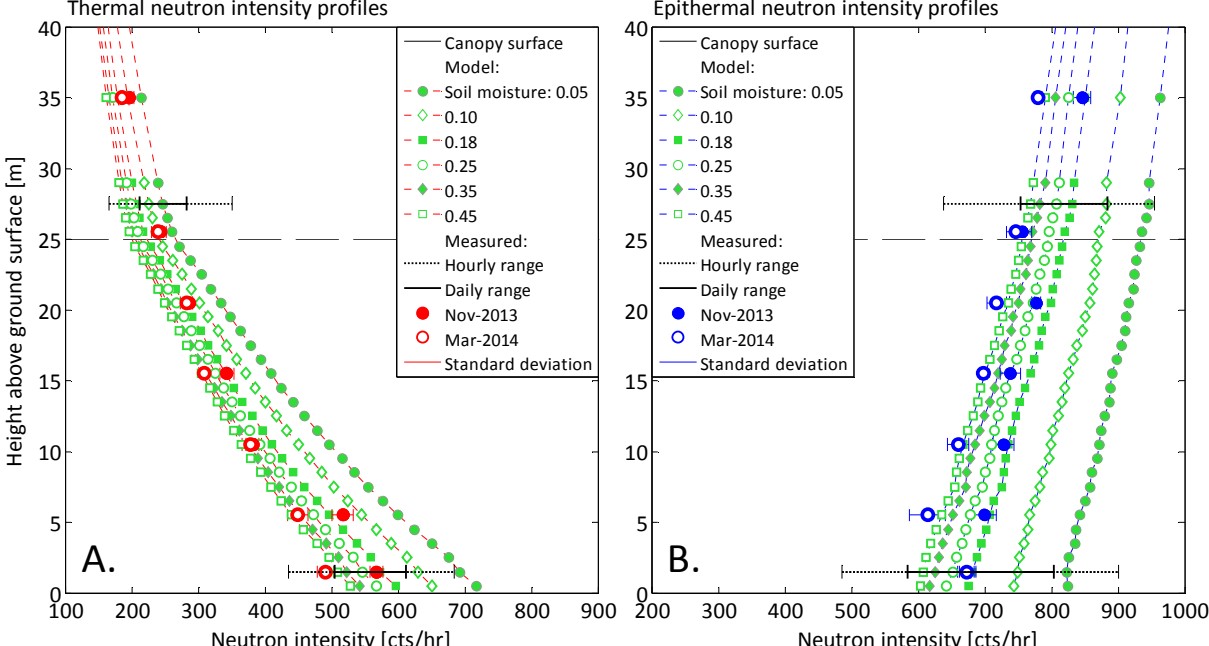

Figure 6 - Sensitivity to soil moisture (Model *Foliage*). Measured and modeled (A.) thermal and (B.) epithermal neutron intensity profiles at Gludsted Plantation. Hourly and daily ranges of variation of thermal and epithermal neutron intensities at ground and canopy level for the period 2013–2015.





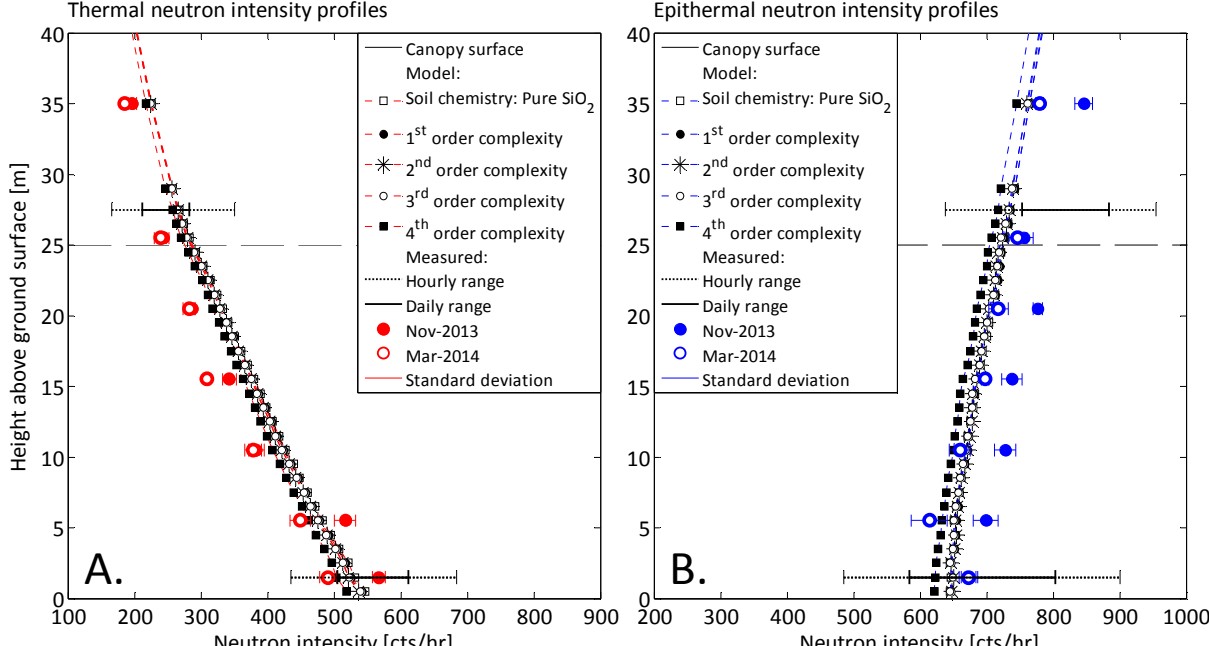

Figure 7 - Sensitivity to soil chemistry complexity (Model *Tree trunk, Foliage, Air*). Measured and modeled (A.) thermal and (B.) epithermal neutron intensity profiles at Gludsted Plantation. Five models of increasing complexity in soil chemistry are shown: 1) Pure $SiO_2$, 2) First order complexity; site specific soil chemistry of the major elements (XRF), 3) Second order complexity; XRF and a Gadolinium (Gd) concentration of 0.51 ppm, 4) Third order complexity; XRF, Gd and an assumed below ground dry biomass of 25 t ha$^{-1}$ (roots), and 5) Fourth order complexity is the *reference model* shown in Fig. 3 and includes XRF, Gd, roots and measured soil organic matter. Hourly and daily ranges of variation of thermal and epithermal neutron intensities at ground and canopy level for the period 2013–2015.





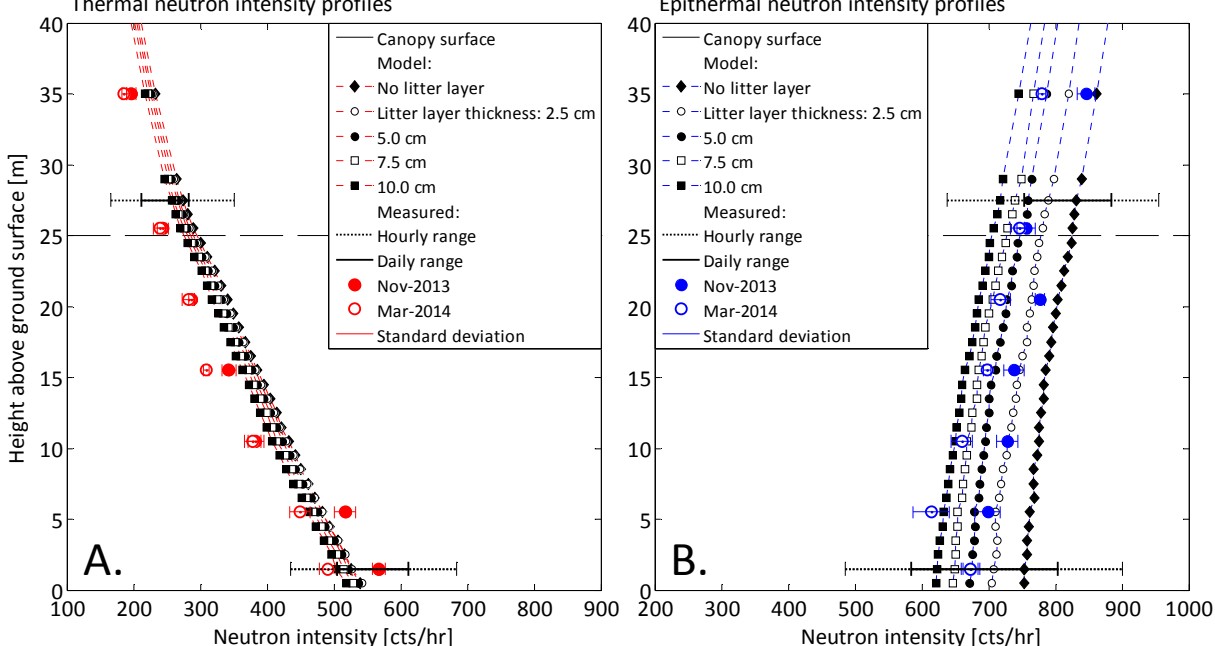

Figure 8 - Sensitivity to litter layer thickness (Model *Tree trunk, Foliage, Air*). Measured and modeled (A.) thermal and (B.) epithermal neutron intensity profiles at Gludsted Plantation. Hourly and daily ranges of variation of thermal and epithermal neutron intensities at ground and canopy level for the period 2013–2015.



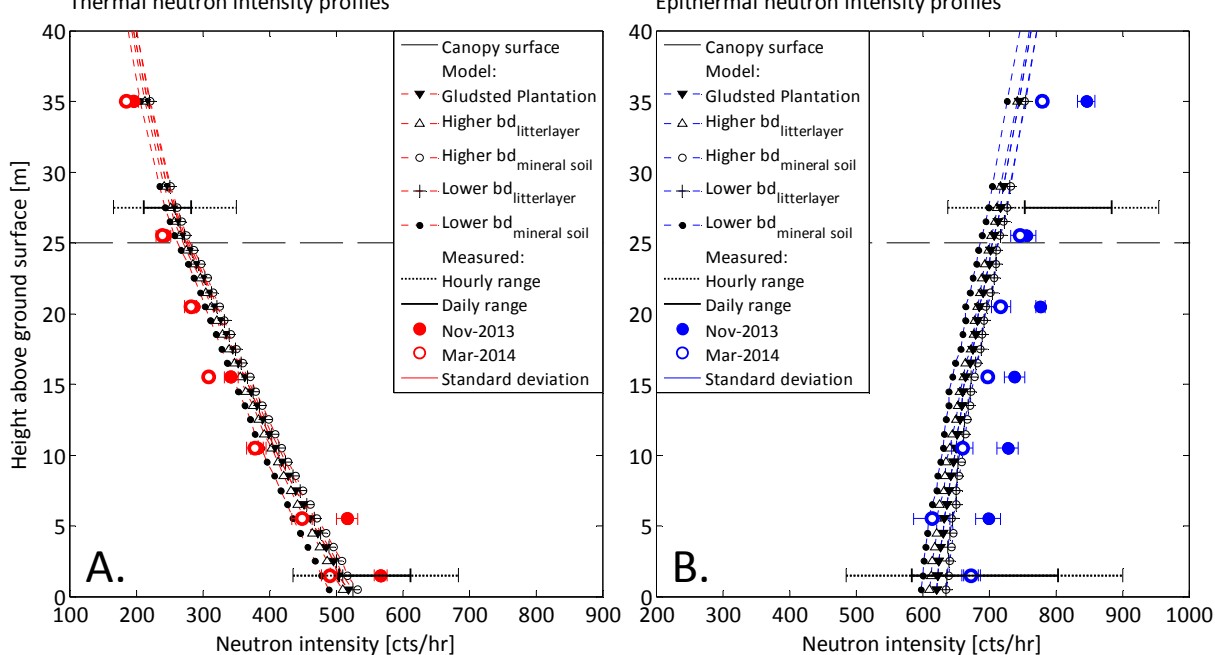

Figure 9 - Sensitivity to litter and mineral soil bulk density (bd) (Model *Tree trunk, Foliage, Air*). Measured and modeled (A.) thermal and (B.) epithermal neutron intensity profiles at Gludsted Plantation. Gludsted Plantation model is the *reference model* with $bd_{dry}$ = 0.34 g cm$^{-3}$ for the litter layer and $bd_{dry}$ = 1.09 g cm$^{-3}$ for the mineral soil (Table 2). Relative to

5  the *reference model* models of higher and lower litter and mineral soil bulk densities are shown: 1) Higher $bd_{litterlayer}$ (litter layer $bd_{dry}$ = 0.50 g cm$^{-3}$), 2) Higher $bd_{mineral\ soil}$ (mineral soil $bd_{dry}$ = 1.60 g cm$^{-3}$), 3) Lower $bd_{litterlayer}$ (litter layer $bd_{dry}$ = 0.20 g cm$^{-3}$), and 4) Lower $bd_{mineral\ soil}$ (mineral soil $bd_{dry}$ = 0.60 g cm$^{-3}$). Hourly and daily ranges of variation of thermal and epithermal neutron intensities at ground and canopy level for the period 2013–2015.





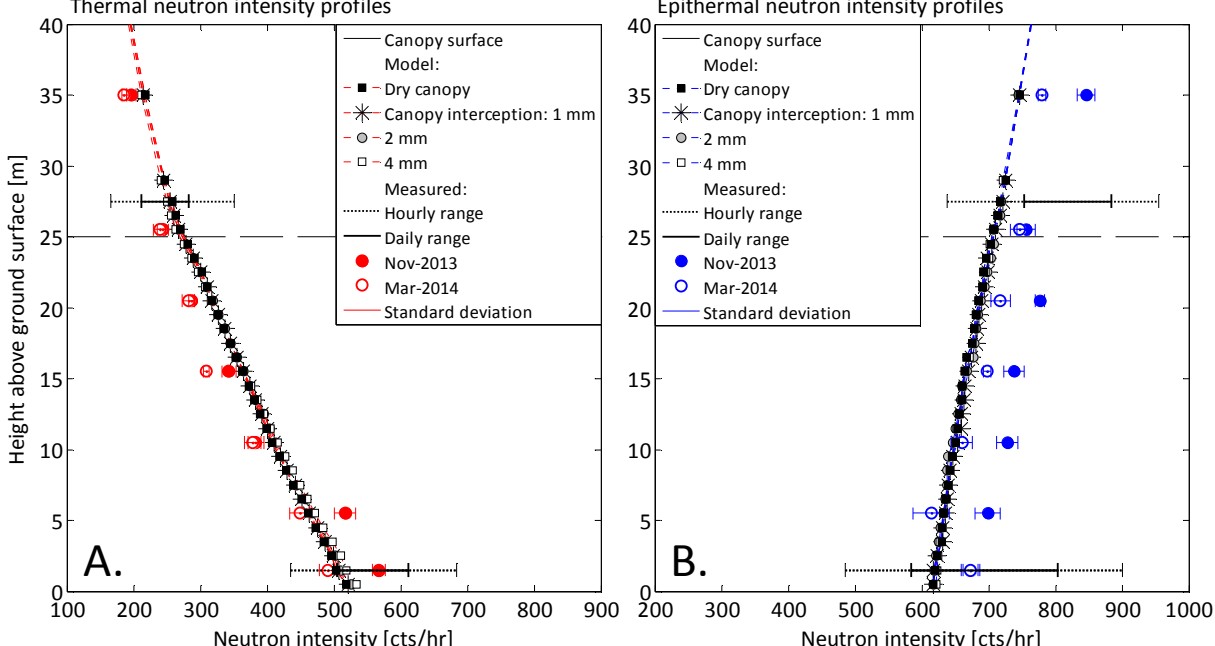

Figure 10 – Sensitivity to canopy interception (Model *Tree trunk, Foliage, Air*). Measured and modeled (A.) thermal and (B.) epithermal neutron intensity profiles at Gludsted Plantation. Hourly and daily ranges of variation of thermal and epithermal neutron intensities at ground and canopy level for the period 2013–2015.

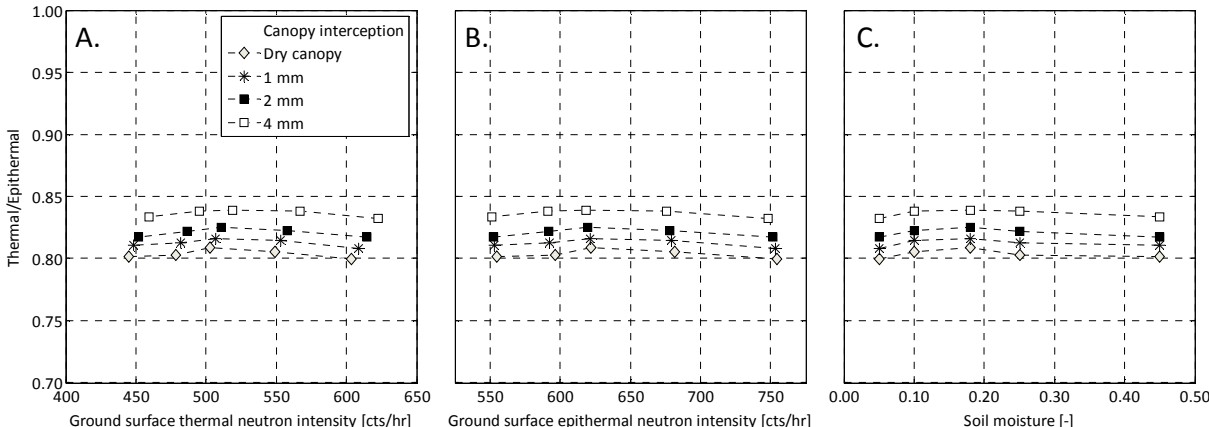

Figure 11 – Modeled ground level thermal-to-epithermal neutron intensity ratios using the Model *Tree trunk, Foliage, Air* for a dry forest canopy and canopy interception of 1 mm, 2 mm and 4 mm. plotted against modeled: A.) ground level thermal neutron intensity, B.) ground level epithermal neutron intensity, and C.) volumetric soil moisture.



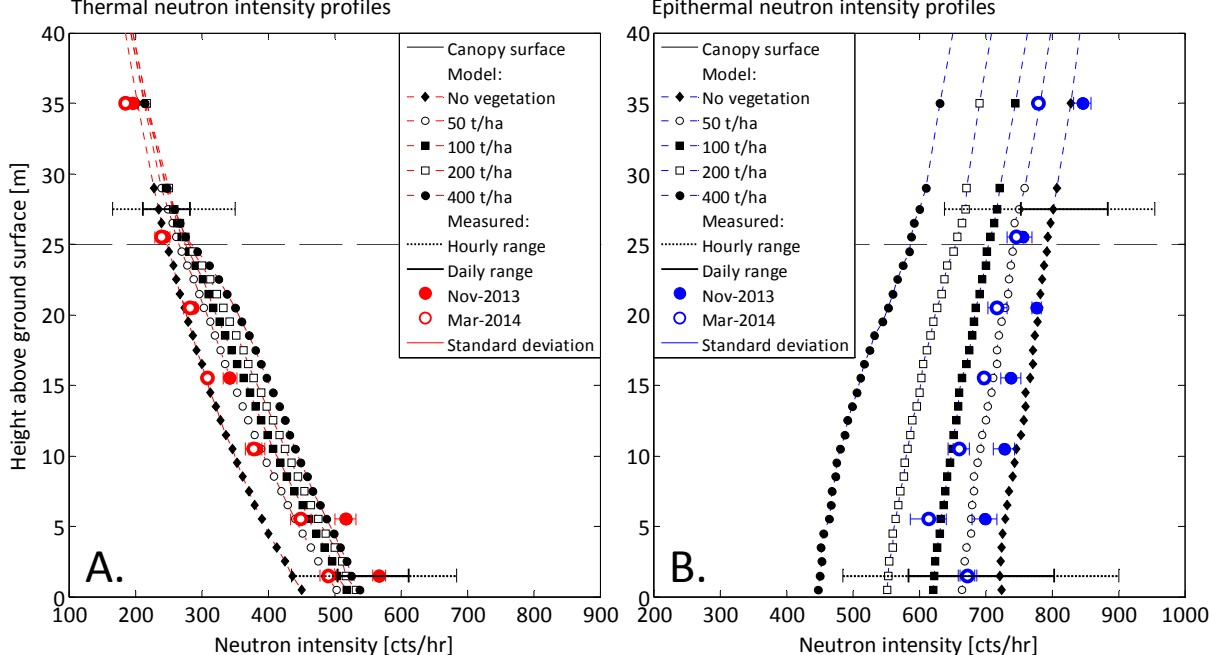

Figure 12 – Sensitivity analysis to forest biomass (Model *Tree trunk, Foliage, Air*). Measured and modeled (A.) thermal and (B.) epithermal neutron intensity profiles at Gludsted Plantation. Hourly and daily ranges of variation of thermal and epithermal neutron intensities at ground and canopy level for the period 2013–2015.



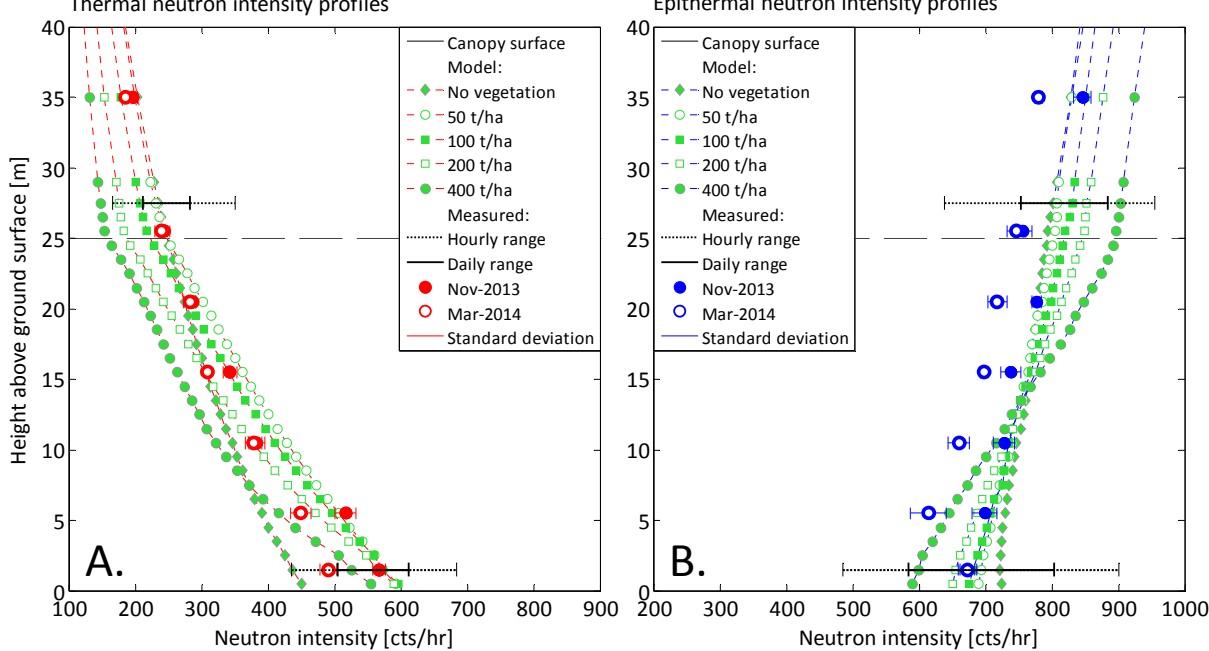

Figure 13 – Sensitivity analysis to forest biomass (Model *Foliage*). Measured and modeled (A.) thermal and (B.) epithermal neutron intensity profiles at Gludsted Plantation. Here, the forest conceptualization of model Foliage is used (see Figure 2B and Figure 7). Hourly and daily ranges of variation of thermal and epithermal neutron intensities at ground and canopy level for the period 2013–2015.





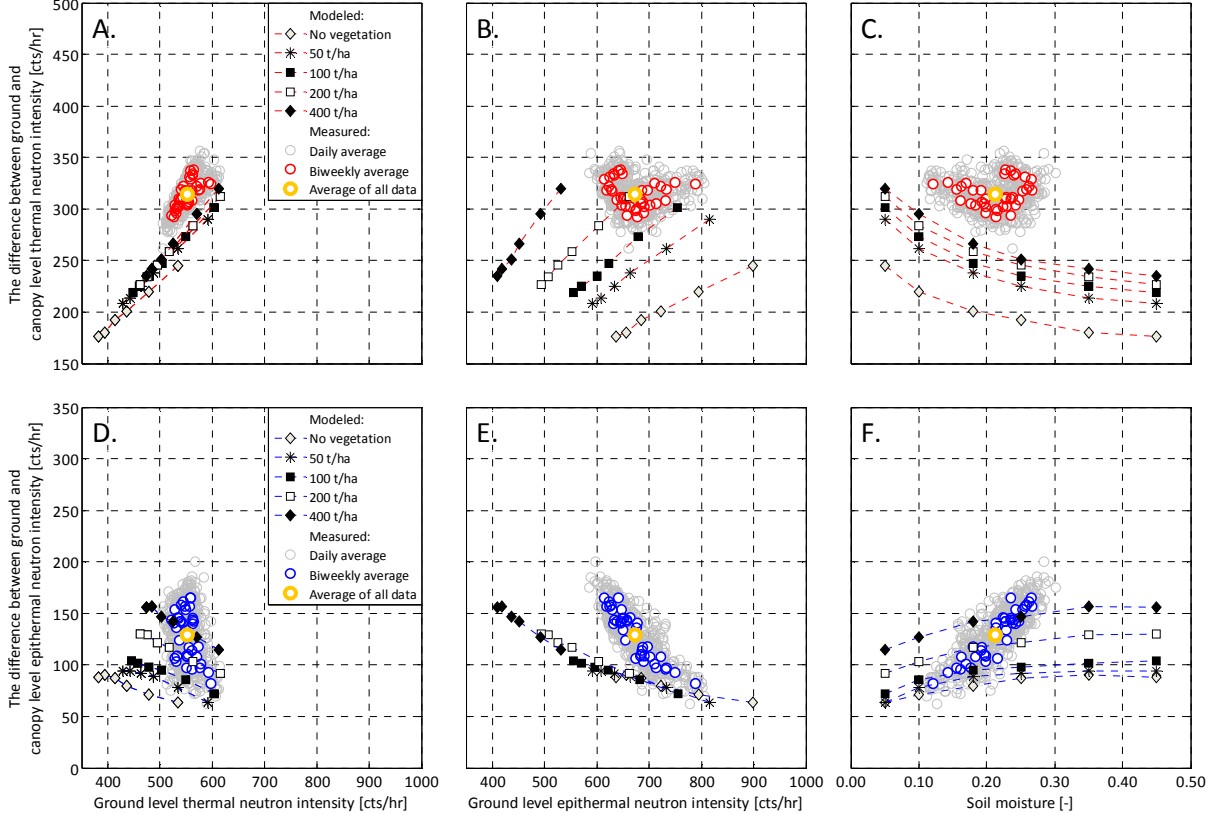

Figure 14 – Neutron intensities measured at Gludsted Plantation in the time period 2013-2015 and modeled using the Model *Tree trunk, Foliage, Air*. Difference in ground and canopy level thermal (A. – C.) and epithermal neutron intensity (D. – F.)

5   plotted against measured and modeled: A. and D.) ground level thermal neutron intensity, B. and E.) ground level epithermal neutron intensity, and C. and F.) volumetric soil moisture.



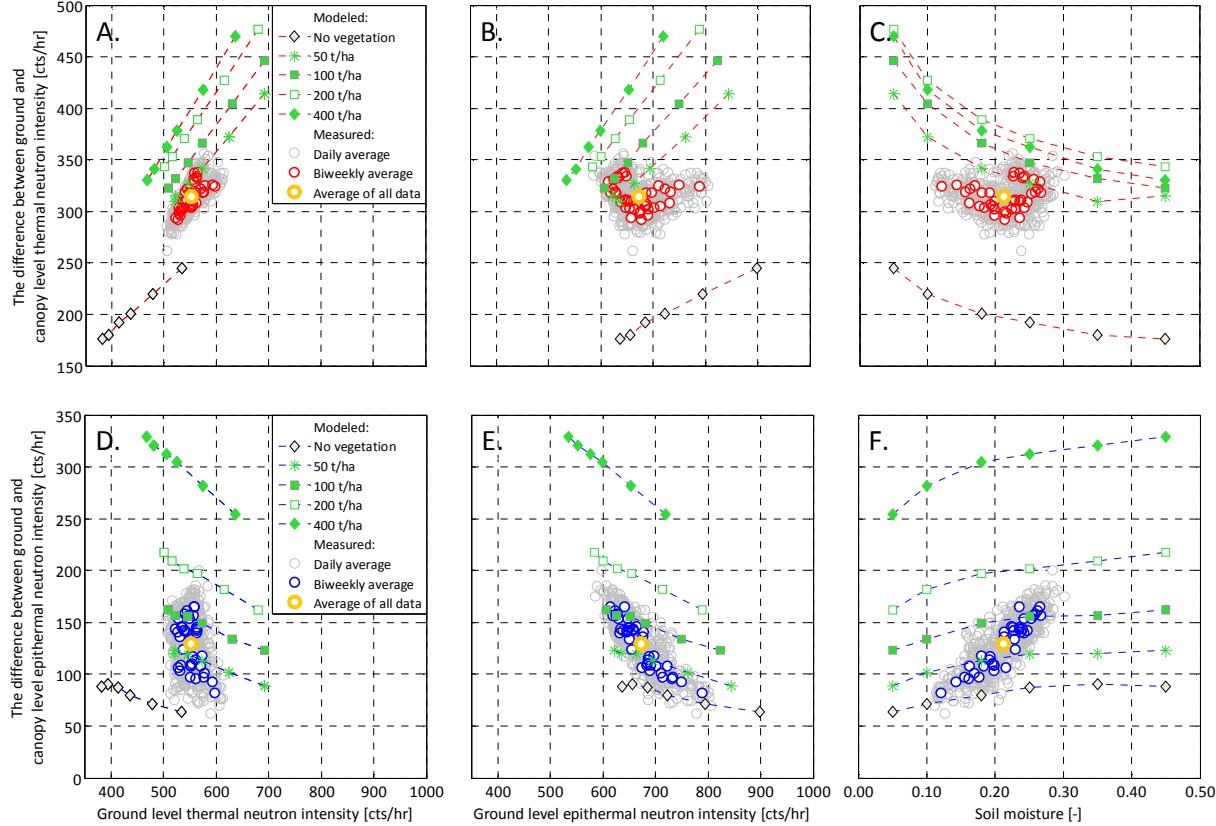

Figure 15 - Neutron intensities measured at the Gludsted Plantation in the time period 2013-2015 and modeled using the Model *Foliage*. Difference in ground and canopy level thermal (A. – C.) and epithermal neutron intensity (D. – F.) plotted against measured and modeled: A. and D.) ground level thermal neutron intensity, B. and E.) ground level epithermal neutron intensity, and C. and F.) volumetric soil moisture.





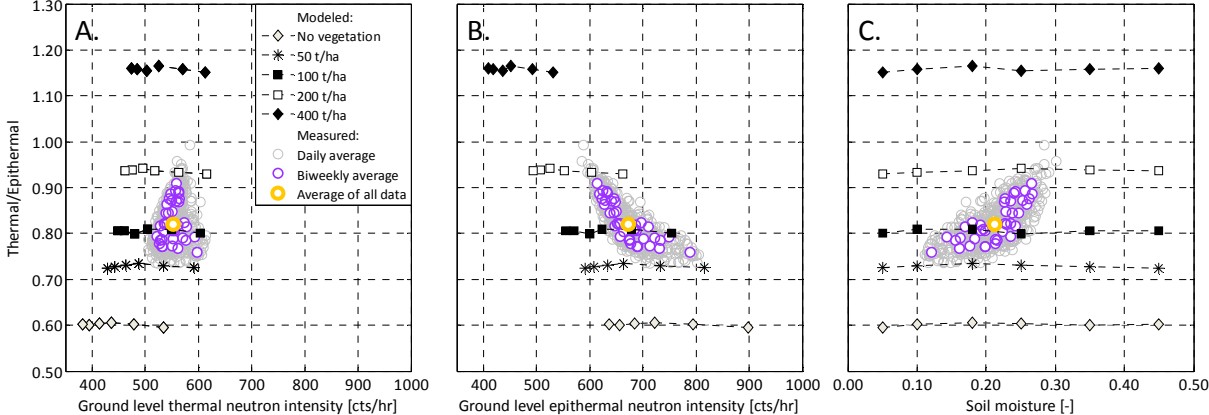

Figure 16 – Neutron intensities measured at Gludsted Plantation in the time period 2013-2015 and modeled using the Model *Tree trunk, Foliage, Air*. Ground level thermal-to-epithermal neutron intensity ratio plotted against measured and modeled: A.) ground level thermal neutron intensity, B.) ground level epithermal neutron intensity, and C.) volumetric soil moisture.

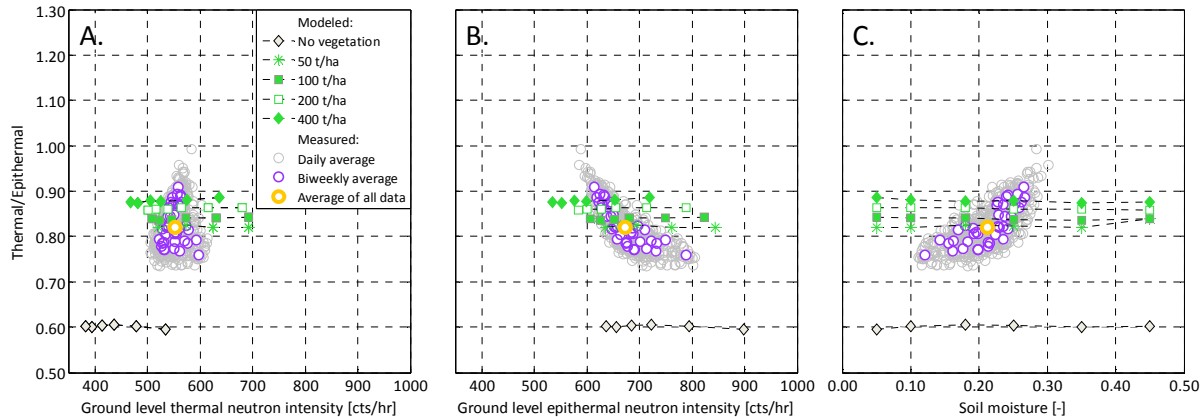

Figure 17– Neutron intensities measured at Gludsted Plantation in the time period 2013-2015 and modeled using the Model *Foliage*. Ground level thermal-to-epithermal neutron intensity ratio plotted against measured and modeled: A.) ground level thermal neutron intensity, B.) ground level epithermal neutron intensity, and C.) volumetric soil moisture.