# Peer review of "Cosmic-ray neutron transport at a forest field site: the sensitivity to various environmental conditions with focus on biomass and canopy interception"

_Hydrology and Earth System Sciences, 2016_

## Referee Comment (RC1) · Anonymous Referee #1 · 9 Jun 2016

The authors present an interesting study combining novel cosmic-ray neutron probe observations with MCNP modeling. The paper is well written and suitable for HESS. Moreover, the paper is a first attempt to better resolve the discrepancies between observed moderated and bare neutron counts and what is modeled with neutron transport simulations. The ability for the CRNP to detect smaller pools of hydrogen in the environment remains a challenging and exciting problem in this field. I have a few suggestions to help improve the manuscript.

Comments:

The Andreasen 2016 WRR article (i.e. pg. 7 L 18 and elsewhere) is not yet available to my knowledge. I suggest the authors remove the citations or include the manuscript

for the reviewers to investigate. Hopefully the WRR paper comes out before this paper, otherwise the reference is inappropriate in its current form or without the accompanying manuscript.

Based off my own unpublished observations of biomass detection with CRNP, I am curious if plotting moderated counts (corrected for water vapor) vs. bare to moderated ratio vs. standing biomass/water equivalence reveals a linear plane. This linear plane is very evident in soybean and maize data. Perhaps plotting the data in this manner will elucidate the biomass and or canopy interception signal?

Pg 3. L6. free parameters is relatively high...

Pg 4. L11. Should coordinates be in decimal degrees in stead of minutes and seconds? Not sure of HESS guidelines...

Pg 4. L23. How dynamic is the 45% vegetation water component over the year? Were repeated bole gravimetric water measurements made? This turned out to be very important in a study in ponderosa pine in AZ. Unfortunately, tree water content is very rarely reported (i.e. Jenkins 2003).

Pg 7. L 30. Despite the CRNP detector footprint mismatch and volume changes, techniques like eddy covariance have overcame these shortcomings to be established as the gold standard in surface energy balance. This is useful to remember when getting caught up into footprint details that may never be fully resolved. No action items but more of a comment.

Pg 9. L 31. Any idea about the effect of clustering or aggregation of trees in space? Probably beyond the scope of this paper but would be interesting to extend this sensitivity analysis to where the detector is located vs. the local aggregation of tree clustering.

Pg 11. L 29. The relative uncertainty for hourly time series is lower than 2-12 hr and daily? Is that true?

Pg 12. L1. Very different despite of similar soil? Sentence doesn't make sense, please

revise.

Pg 15. L9. As we from the calculation? Sentence doesn't make sense, please revise.

Pg. 16 L 30. Are highly variable...

Pg 18. L 7. A remarkable fit...

---

## Referee Comment (RC2) · G. Baroni (Referee) · 22 Jun 2016

Summary

The Authors present the results of a neutron model used to explore the effect of different hydrogen pools on the signal of the Cosmic-Ray neutron sensors (CRNS). The neutron model was set-up to mimic a specific forest site in Denmark. Based on that, a sensitivity analysis (SA) to several environmental conditions (7 factors) was provided. The effect on thermal neutrons, epithermal neutrons and sensors placed at different heights are discussed. The study is relevant since the CRNS is a method that was applied in several conditions for soil moisture measurements but the role of other hydrogen pools has to be further investigated. Overall, the manuscript (MS) could be an

interesting publication suitable for HESS. However, it needs improvement in different directions. The story line is not always consistent, the introduction part is limited and the presentation of the results should be better organized. Finally, I think the MS could be extended with a discussion section. For these reasons I think the Authors should put some more effort to improve the manuscript before publication.

General comments

[1] The story line is built on the use of CRNS for biomass and canopy interception while a SA is conducted to explore the role of several other hydrogen pools. Moreover, in my opinion, the manuscript is relevant also because the neutron modeling explores in details the use of thermal neutrons and, for the first time, the use of sensors placed at different heights. However, these two novel aspects are completely missed in the introduction and they are taken for granted in the discussion of the results. For these reasons I think the story line is not consistent with the actual analysis reported and introduction and conclusions does not provide a clear roadmap and summary of what this study accomplishes. Overall the manuscript should be reshaped along a clearer story line more consistent with the analyses reported where the Readers should be introduced to the actual state of the CRNS applications (e.g., only moderated counter and just above ground measurements). Novelties of the study and concluding remarks about potentiality and limitations should be better clarified in the final conclusions (i.e., the use of the bare counter and ratio between bare and moderated; the use of sensors placed at different heights; the effect of several environmental conditions to the signal). Specific comments/suggestions are reported below.

[2] Despite I understand the goal of the Authors to strengthen the need of such a study, I found in the introduction several statements that are misleading (e.g., P2L19-25). Contrary to what is stated by the Authors, in my knowledge several important contributions were published to address the (wanted or unwanted) effect of additional hydrogen pools. Moreover, most of these studies focused on the effect of biomass e.g., in addition to the references reported in the MS, preliminary evaluation of biomass were

presented in (Rivera Villarreyes et al., 2011); (Franz et al., 2013) presented an approach to isolate any hydrogen pools but soil moisture and showed the estimation of the crop biomass; (Baatz et al., 2015; Hawdon et al., 2014) introduced an empirical correction to account for biomass. In comparison to biomass, the effect of snow on CRNS signal has received much less attentions. Even if the first concepts were already introduced by (Desilets et al., 2010), a preliminary analysis was just presented by (Rivera Villarreyes et al., 2011) and only recently a study with longer time series of snow was published (Sigouin and Si, 2016). Other hydrogen pools were also addressed: e.g., the analysis of the role of little layer was discussed in detail by (Bogena et al., 2013) and in (Baroni and Oswald, 2015) we presented the first measurements for quantifying also the canopy interception. Overall I believe that all these experimental studies called for additional attentions on hydrogen pools than soil moisture. In this context, the present MS is the first modeling study where complex forest is simulated and the effect of several environmental factors are explored. For these reasons I think the MS could represent a good answer to those calls and the introduction of the MS should be rephrased accordingly.

[3] I found the presentation of the results obtained with the reference model and the forest conceptualizations not clear (P11L24- P13L7). The Authors first stated about a remarkable agreement of the reference model (P12L12). Later they compared different forest conceptualizations and they found the best fit not to be unique (P13L2-4). Similarly they stated that they cannot determine which conceptualization is more realistic (P16l7-9). For this reason they conducted the SA using two conceptualizations. Overall, I believe that the mismatch should be clearly acknowledged from the beginning. Assuming that two forest conceptualizations are selected, the results of the SA could be then presented.

[4] The discussion of the results of the SA is not always clear and together with the 17 images I think the Readers are lost on the major findings of the study. In addition most of the discussion reported is a qualitative description of the figures. I would suggest searching for a way to sum up the results section (i.e., reducing the number of figures) where first the results of thermal and epithermal neutrons are discussed providing a quantitative comparison of the different effect of the environmental conditions explored. Secondly the ratio between thermal and epithermal is introduced and results are discussed for the factors that showed different response in thermal and epithermal neutrons.

[5] It would be interesting to extend the MS with a discussion section where the overall results of the SA are summarized e.g., the advantages of using sensors at different heights, the advantages of using thermal and epithermal neutrons, the misfits of model and measurements and indication for further improvements. Concluding remarks could stress the potential use of CRNS for other applications but it would be interesting to extend the discussion also on the role of the spatial sensitivity of the sensor i.e., any estimation by CRNS is a spatial weighted value of the actual target (e.g., biomass). Finally, it is stated that for a good matching between measurements and simulations it was important the correcting factor (Page 7, L10-23). Since all the probes installed so far around the world does not account for that, it would be important to know what the implications are e.g., could we aspect the same sensitivity to environmental conditions when comparing bare and moderated counter instead of thermal and epithermal neutrons?

Specific comments

Page 1, L1-2: the title's focus on biomass and canopy interception is not entirely representative of the sensitivity analysis presented in the MS, which is broader. It should be rephrased accordingly.

Page 1, L18: in my knowledge the effect of snow has received much less attention than other hydrogen pools. In addition, the analysis reported in the MS does not focus on biomass and interception but several other factors are discussed. For this reason I would rephrase the sentence in "...soil moisture but several other hydrogen pools

affect the signal".

Page 1, L22-31: in my opinion the presentation of the main results should be extended to honor also the other analyses provided in the MS (i.e., the role of the other factors).

Page 2, L12: the terminology used (static, quasi-static and dynamic) is too arbitrary. For a clearer discussion I would suggest presenting the hydrogen compartments in term of temporal scales (e.g., hours/days, season, years).

Page 2, L15: for consistency I would mention here that the signal of hydrogen pools with low temporal dynamic (e.g., lattice water, SOC etc) is usually subtracted.

Page 2, L18-25: see general comment #2 and the additional references reported to reshape the paragraph.

Page 2, L26 – Page 3, L18: the sensitivity analysis focuses on several environmental conditions. In the light of reshaping the MS to honor this, I would say that these paragraphs are not relevant and could be omitted.

Page 3, L19 – L34: summary of the aims of the paper and the methods should be rephrased to honor the actual analysis i.e., sensitivity analysis to environmental conditions to understand the role of different hydrogen pools.

Page 4, L1-2: the sentence is misleading: as reported in general comment #2 several publications were presented to estimate biomass. In (Baroni and Oswald, 2015) we have also presented the first measurements of canopy interception. Even in the case the Authors have any concerns about these studies, I think it would be part of the constructive advanced of the research field to integrate these opinions in the MS.

Page 4, L3: this section 2 could be moved and integrated in the section 3.2.4 Field measurements.

Page 7, L10-23: for a clearer description of the results, the Authors could start the section making the list of the factors analyzed and referring here also to table 5. In

addition the values presented in Table 5 could be plotted for easier comparison (e.g., bar plot).

Page 12, L4: I'm surprised: do you really think that the soil moisture profiles could explain such a difference? But in case it is relevant, why did you not evaluate this in the SA? Overall understanding the role of the different factors (i.e., environmental conditions) is the goal of the SA and of this paper.

Page 12, L12: I think the term "remarkable agreement" should be rephrased in the light of the overall discussion reported about the discrepancies and the inability to define which conceptualization is more realistic (e.g., P16L8). In addition I noticed that the thermal measurements show a regular decreasing from the ground to the canopy level. On the contrary the epithermal measurements show an inversion: the measurements decrease from ground to 5 meters and then start to increase regularly when moving to the canopy level. If I'm not wrong none of the models conceptualizations and the different environmental settings is able to reproduce this behavior. For this reason I think it could be an important result to discuss. Unfortunately this behavior is detected only for the profile measured on Mar-2014 while the measurements conducted on Nov-2013 does not have these measurements in the plot: is this a mistake in plotting or really do you not have these measurements?

Page 12, L26: if it is a SA the results should be discussed in term of sensitive or not sensitive. The term "satisfactorily" suggests that here you are still looking for a forest conceptualization that fits the profile measurements. See also general comment #3.

Page 13, L8-19: this paragraph could be titled as a new section e.g., effect of soil moisture. Possibly, the analysis could be extended to explore the effect of soil moisture profiles (see also comment Page 12, L4).

Page 14, L25 - Page 15, L8: the presentation of the results jumps from the description of the thermal neutrons to the ratio i.e., epithermal neutrons are not described. For a clearer presentation I would suggest first to discuss both thermal and epithermal

neutrons. Secondly, to introduce the use of the ratio explaining the reasons for doing that e.g., what do you expect to see with the use of the ratio instead of the single signal?

Page 18, L21 - Page 19, L5: the discussion about the results obtained with the field locations of Voulund and Harrild is very limited and it refers to analysis presented in the submitted (and not available) paper of Anderson et al. (2016). Moreover I think that at the current status, these results do not provide any new insights on the present study. Either the Authors integrate better the description, the analyses and the results obtained in these locations, or in my opinion the results obtained based on these two sites could be completely omitted in this MS.

Page 18, L19: as discussed also in previous comments. I think the term "remarkable agreement" is misleading. On the contrary I think the misfits are interesting results to highlights providing the base for further studies.

Page 18, L22: before starting speaking about canopy interception, I would introduce also a summary of the role of the other hydrogen pools explored. This would better honor the SA reported in the MS.

Technical corrections

Page 4, L11: eV

Page 13, L26: the definition (4th order, 3rd order) of the chemical complexity are not self-explained. I would suggest instead the use in the table of other definitions e.g., (SOM+Gd+Root+??+SiO2) for the more complex and so on.

Page 13, L28: what is cts? To be defined.

Page 15, L4: conditions instead of locations.

Figure 1: the domain represented in the figure is too extended and not well informative. I would suggest using this as a general overview but adding also a panel where the

positions of the experimental sites are visualized with higher resolutions.

References

Baatz, R., Bogena, H.R., Hendricks Franssen, H.-J., Huisman, J.A., Montzka, C., Vereecken, H., 2015. An empirical vegetation correction for soil water content quantification using cosmic ray probes. Water Resour. Res. n/a-n/a. doi:10.1002/2014WR016443

Baroni, G., Oswald, S.E., 2015. A scaling approach for the assessment of biomass changes and rainfall interception using cosmic-ray neutron sensing. J. Hydrol. 525, 264–276. doi:10.1016/j.jhydrol.2015.03.053

Bogena, H.R., Huisman, J.A., Baatz, R., Hendriks-Franssen, H.-J., Vereecken, H., 2013. Accuracy of the cosmic-ray soil water content probe in humid forest ecosystems: The worst case scenario. Water Resour. Res. n/a-n/a. doi:10.1002/wrcr.20463

Desilets, D., Zreda, M., Ferré, T.P.A., 2010. Nature's neutron probe: Land surface hydrology at an elusive scale with cosmic rays. Water Resour. Res. 46. doi:10.1029/2009WR008726

Franz, T.E., Zreda, M., Rosolem, R., Hornbuckle, B.K., Irvin, S.L., Adams, H., Kolb, T.E., Zweck, C., Shuttleworth, W.J., 2013. Ecosystem-scale measurements of biomass water using cosmic ray neutrons. Geophys. Res. Lett. n/a-n/a. doi:10.1002/grl.50791

Hawdon, A., McJannet, D., Wallace, J., 2014. Calibration and correction procedures for cosmic-ray neutron soil moisture probes located across Australia. Water Resour. Res. n/a-n/a. doi:10.1002/2013WR015138

Rivera Villarreyes, C.A., Baroni, G., Oswald, S.E., 2011. Integral quantification of seasonal soil moisture changes in farmland by cosmic-ray neutrons. Hydrol. Earth Syst. Sci. 15, 3843–3859. doi:10.5194/hess-15-3843-2011

Sigouin, M.J.P., Si, B.C., 2016. Calibration of a non-invasive cosmic-ray probe for

wide area snow water equivalent measurement. The Cryosphere 10, 1181–1190. doi:10.5194/tc-10-1181-2016

---

## Referee Comment (RC3) · Anonymous Referee #3 · 29 Jun 2016

General Comments:

In this paper the authors use a model to investigate how different model conceptualizations and different model parameters influence both thermal and epithermal neutron intensities from the ground surface up to a height of 35 m. They want to find out whether it is possible to use combined measurements of thermal and epithermal neutrons at ground level to determine both aboveground vegetation biomass (quasi-statically) and canopy interception (dynamically). In order to do that they need to assess whether there are factors other than aboveground biomass and interception that alter the ratio of thermal to epithermal neutron intensity.

I like the approach. It is novel to measure neutron intensities at different heights in

a forest and it is novel to try to use the ratio of thermal to epithermal neutrons for biomass determination. Therefore the topic is interesting and the paper is well-suited for publication in HESS.

Still, there is room for improvement. In the end as a reader I felt a little lost on what are the actual outcomes from the study. It seems as if equifinality is a very big problem. Many of the investigated model setups and parameters seem to influence the detected neutron intensity profiles and therefore it is unclear which setup represents reality best. Unfortunately, the discussion section often lacks more detailed interpretations of the comparison of model results and measurements. Therefore the full potential of the study is not yet explored.

So my main point is that a refocus of the discussion section (away from just describing towards interpreting) would definitely improve the manuscript and the value for the reader.

Specific Comments:

p. 1, l. 1: Title: Since canopy interception only plays a minor role in the paper, I would suggest removing it from the title. You are investigating so many more things, like forest canopy representation, complexity of soil matrix chemistry, litter, soil bulk density. A more obvious choice for the title might be going along the line of forest canopy representation (since this part appears most prominently and novel when reading the manuscript). Also, posing a question in the title is not ideal, especially when you answer one part of it with no and the other part with yes.

p. 1, l. 31: It would be good to insert an explicit concluding statement into the abstract that answers the question you were posing in the title. ('Therefore we conclude that while there is potential to infer biomass from cosmic... canopy interception cannot be inferred.')

p. 2, l. 2: '..relativeLY high concentration CLOSE TO THE LAND SURFACE,...'

p. 2, l. 2-10: I would reorder this paragraph. Start with the role of soil moisture and the difficulties of its detection. Then introduce cosmic-ray neutrons and the detector before mentioning its footprint in line 7.

p. 2, l. 13: In Table 1 you use the word 'transient', here you say 'dynamically'.

p. 2, l. 13-14: Try to categorize this list. 'Hydrogen is stored statically in water in soil minerals and buildings/roads, quasi-statically in. . .'

p. 2, l. 31: 'HOWEVER, the spatial scale of measurement. . .'

p. 3, l. 27: '. . .we PERFORM a sensitivity analysis. . .'.

p. 3, l. 28: Only to look at their effect on MODELED thermal and epithermal neutron intensity? Or also to make statements about their effect on ACTUAL thermal and epithermal neutron intensity?

p. 4, l. 16: Could you shortly introduce what this 'root-to-shoot ratio' is?

p. 4, l. 16: Information? Be more specific.

p. 5, l. 2: Why random soil samples? A composite sample representing mean soil properties would have been much more representative of the soils within the footprint of the sensor given small-scale variability.

p. 5, l. 24: What do you mean by: '. . .is observed VISUALLY. . .'?

p. 5, l. 26: Do you mean that the hardpan-layer hinders percolation to deeper depths?

p. 6, l. 12: represent might not be the right word here. Maybe 'detect' or 'be sensitive to'?

p. 6, l. 13: What do you want to express when you write: 'Despite this fact. . .'?

p. 7, l. 5: The term 'epithermal' includes 'fast', no? So you don't need to say '. . .fast and epithermal. . .'.

p. 7, l. 7: Why do you believe in this minor effect on your results?

p. 7, l. 19-20: What are these correction factor models, when exactly where they applied and how did the output of these models look compared to the cadmium-difference model?

p. 7, l. 32-34: The fact that the environmental conditions at the field sites are fairly homogeneous is no explanation for your assumption that the neutron intensities measured by the two different detectors can be compared. Please elaborate.

p. 9, l. 5-32: What about the sub-canopy structure of real forests? With a lot of the leaves and branch biomass a couple of meters above the ground and only the trunks with a lot of air in between near the ground surface. Would you expect the same outcome? How could this impact your results? It would be good to discuss this somewhere.

p. 10, l. 20-21: Rephrase. Maybe something like: 'The thermal and epithermal neutron intensity is both a product of hydrogen abundance as well as elemental composition. . .'.

p. 12, l. 30: From here on I will ask the question 'Why?' whenever I would like to see a more detailed discussion of one of your results/observations. Throughout the discussion section there are instances where you observe and describe your results without giving a proper (attempt of) interpretation. For example here you state that '. . .the neutron intensity profiles of the simpler forest canopy conceptualization . . . is less steep and is the only model providing an epithermal neutron intensity profile within the daily ranges of the time-series measurements. . .'. Still there is no explanation on why this could be the case.

p. 13, l. 19: Why?

p. 14, l. 17: Why?

p. 14, l. 19: How can the mineral soil act as a producer of epithermal neutrons? Thermal neutrons would have to be accelerated to become epithermal. How does this

happen?

p. 15, l. 7: Move '...from the calculation in the previous section...' to the beginning or the end of the sentence.

p. 15, l. 31-32: Why?

p. 16, l. 5-6: Why?

p. 16, l. 26-27: Why?

p. 17, l. 7-17: So would you say that this model representation is better than the more complex one? It certainly fits better to your observed data. What does it mean that the average conditions (without separate trunk, foliage, air) perform better? It should be the other way around, no?

p. 17, l. 22: Do you maybe mean '...prevailING at the field site.'

p. 17, l. 31-32: Why?

p. 18, l. 7-12: Is that an indication that this more complex model is a more realistic representation of the forest environment? How is this observation compatible with the previous observation that shows the better fit of the less complex model when comparing the differences between ground and canopy level thermal and epithermal neutron intensity?

p. 18, l. 13-20: How would each of these 3 factors influence the modeled ratios?

p. 19, l. 3: Should the amount of biomass not be slightly larger for the Heathland site compared to the non-vegetated Gludsted plantation?

p. 19, l. 6: It would be helpful to introduce an abbreviation for the term 'thermal-to-epithermal ratio' somewhere at the beginning (Rt/e) and use it throughout the manuscript.

Figures & Tables:

Figure 1: Provide a map that zooms in onto your study area with a little more detail and move the current overview map of Denmark into one of the corners of the new map.

Figure 3-10, 12-13: Remove the line in the legend in front of 'Canopy surface model'. I was looking for it but it is not in the actual figure, is it? Maybe just call it 'Modeled' in comparison to 'Measured'.

Technical Corrections:

p. 1, l. 25: 'minor' is no adverb. Maybe use 'insignificantly'.

p. 1, l. 27: siteS

p. 4, l. 5: '. . .within THE Skjern River. . .'.

References:

A couple of references are listed but not referenced in the text:

Bogena et al. 2013

Heidbüchel et al. 2016

Rivera Villareyes et al. 2013

―――――――――――――――――――

---

## Author Comment (AC1) · 3 Oct 2016

**Replies to review comments**

**MS No.: hess-2016-226**

**by Andreasen et al.**

**Referee #1:**

The authors present an interesting study combining novel cosmic-ray neutron probe observations with MCNP modeling. The paper is well written and suitable for HESS. Moreover, the paper is a first attempt to better resolve the discrepancies between observed moderated and bare neutron counts and what is modeled with neutron transport simulations. The ability for the CRNP to detect smaller pools of hydrogen in the environment remains a challenging and exciting problem in this field. I have a few suggestions to help improve the manuscript.

Comments:

The Andreasen 2016 WRR article (i.e. pg. 7 L 18 and elsewhere) is not yet available to my knowledge. I suggest the authors remove the citations or include the manuscript for the reviewers to investigate. Hopefully the WRR paper comes out before this paper, otherwise the reference is inappropriate in its current form or without the accompanying manuscript.

AC1 (Author comment # 1): The paper was accepted on July 29 2016. The reference provided in the manuscript has been updated.

Andreasen, M., K. H. Jensen, M. Zreda, D. Desilets, H. Bogena, and M. C. Looms (2016), Modeling cosmic ray neutron field measurements, Water Resour. Res., 52, doi: 10.1002/2015WR018236.

Based off my own unpublished observations of biomass detection with CRNP, I am curious if plotting moderated counts (corrected for water vapor) vs. bare to moderated ratio vs. standing biomass/water equivalence reveals a linear plane. This linear plane is very evident in soybean and maize data. Perhaps plotting the data in this manner will elucidate the biomass and or canopy interception signal?

AC2: We did the plot as suggested by Referee #1. We found a plane, yet, it was not very evident, and we have therefore chosen not to include it in the manuscript.

Pg 3. L6. free parameters is relatively high. . .

AC3: The suggested change has been added to the manuscript.

Pg 4. L11. Should coordinates be in decimal degrees instead of minutes and seconds? Not sure of HESS guidelines. . .

AC4: HESS manuscript preparation guideline for authors has few details on coordinate systems. I have looked through a few papers published by HESS, and here they used the same coordinate system as we do (e.g. Hengl, T., Heuvelink, G. B. M. and van Loon, E. E. (2010): On the uncertainty of stream networks derived from elevation data: the error propagation approach, Hydrol. Earth Syst. Sci., 14, 1153–1165, 2010, doi: 10.5194/hess-14-1153-2010).

Pg 4. L23. How dynamic is the 45% vegetation water component over the year? Were repeated bole gravimetric water measurements made? This turned out to be very important in a study in ponderosa pine in AZ. Unfortunately, tree water content is very rarely reported (i.e. Jenkins 2003).

AC5: Unfortunately, no bole gravimetric measurements were conducted. The same water content is assumed for the whole bole although the water content in the outer rim of a spruce holds more water than in the core of the bole (www.trae.dk - Danish reference). We found this assumption to be appropriate as a first attempt to model the neutron transport for a forest field site, however, we may for future studies include a more detailed description of the trees and the forest canopy.

Pg 7. L 30. Despite the CRNP detector footprint mismatch and volume changes, techniques like eddy covariance have overcame these shortcomings to be established as the gold standard in surface energy balance. This is useful to remember when getting caught up into footprint details that may never be fully resolved. No action items but more of a comment.

AC6: We agree with referee #1 and have changed the wording in Section "Footprint" a bit. A line has been erased: "The potential mismatch in the footprint of the bare and the moderated detectors is a concern when combining the neutron intensity measurements."

The last part of the section is now as follow:

"…The potential mismatch in the footprint of the bare and the moderated detectors is a concern when combining the neutron intensity measurements. Nevertheless, the environmental conditions at the field sites are fairly homogeneous and although the footprint might be different as a first approximation we assume the neutron intensity measured using the bare and the moderated detector are comparable."

Pg 9. L 31. Any idea about the effect of clustering or aggregation of trees in space? Probably beyond the scope of this paper but would be interesting to extend this sensitivity analysis to where the detector is located vs. the local aggregation of tree clustering.

AC7: We have not yet tested the impact of clustering/aggregating the trees, but we would like to in the near future. We have included this in the discussion in section 3.1.:

"Improved comparability to measurements may be obtained by advancing the forest canopy conceptualization. Currently, one tree is defined and repeated throughout the model domain. The trees are placed in even rows and the same settings are applied from the ground surface to 25 m height. In order to advance the forest canopy conceptualization, trees of different heights and diameters could be included, and the placement of the

trees could be more according to the actual placement of trees at the forest field site. Additionally, variability in tree trunk diameter, foliage density and volume with height above the ground surface could be implemented."

Pg 11. L 29. The relative uncertainty for hourly time series is lower than 2-12 hr and daily? Is that true?

AC8: Thanks for pointing this out. "Lower" has been changed to "higher".

Pg 12. L1. Very different despite of similar soil? Sentence doesn't make sense, please revise.

AC9: The paragraph has been changed:

"Still, some differences are observed between the neutron height profiles measured in November 2013 and March 2014. The soil moisture was similar during the time of neutron profile detection and we expected the differences to be …"

Pg 15. L9. As we from the calculation? Sentence doesn't make sense, please revise.

AC10: The sentence has been changed: "We choose not to include measurements in the figure because the measurement uncertainty at a relevant integration time is greater than the signal of canopy interception."

Pg. 16 L 30. Are highly variable. . .

AC11: The suggested change has been added to the manuscript.

Pg 18. L 7. A remarkable fit. . .

AC12: The suggested change has been added to the manuscript.

**Referee #2:**

The Authors present the results of a neutron model used to explore the effect of different hydrogen pools on the signal of the Cosmic-Ray neutron sensors (CRNS). The neutron model was set-up to mimic a specific forest site in Denmark. Based on that, a sensitivity analysis (SA) to several environmental conditions (7 factors) was provided. The effect on thermal neutrons, epithermal neutrons and sensors placed at different heights are discussed. The study is relevant since the CRNS is a method that was applied in several conditions for soil moisture measurements but the role of other hydrogen pools has to be further investigated. Overall, the manuscript (MS) could be an interesting publication suitable for HESS. However, it needs improvement in different directions. The story line is not always consistent, the introduction part is limited and the presentation of the results should be better organized. Finally, I think the MS could be extended with a discussion section. For these reasons I think the Authors should put some more effort to improve the manuscript before publication.

AC13: We agree that the story line is a bit week and that the paper is a bit hard to read. We find your comments and suggestions very helpful. The introduction and results section has been changed considerably, and the conclusion has been updated and improved.

Details on our edits and changes are provided in the sections below.

General comments

[1] The story line is built on the use of CRNS for biomass and canopy interception while a SA is conducted to explore the role of several other hydrogen pools. Moreover, in my opinion, the manuscript is relevant also because the neutron modeling explores in details the use of thermal neutrons and, for the first time, the use of sensors placed at different heights. However, these two novel aspects are completely missed in the introduction and they are taken for granted in the discussion of the results. For these reasons I think the story line is not consistent with the actual analysis reported and introduction and conclusions does not provide a clear roadmap and summary of what this study accomplishes. Overall the manuscript should be reshaped along a clearer story line more consistent with the analyses reported where the Readers should be introduced to the actual state of the CRNS applications (e.g., only moderated counter and just above ground measurements). Novelties of the study and concluding remarks about potentiality and limitations should be better clarified in the final conclusions (i.e., the use of the bare counter and ratio between bare and moderated; the use of sensors placed at different heights; the effect of several environmental conditions to the signal). Specific comments/suggestions are reported below.

AC14: As suggested by the reviewer we have reworked the manuscript to obtain a clearer story line.

The introduction has been reworked:

[revised manuscript text omitted]

"The response to altered amounts of biomass on thermal and epithermal neutron intensity is non-unique for the simple and complex forest conceptualization and further advancement of the forest representation is therefore necessary."

[2] Despite I understand the goal of the Authors to strengthen the need of such a study, I found in the introduction several statements that are misleading (e.g., P2 L19-25). Contrary to what is stated by the Authors, in my knowledge several important contributions were published to address the (wanted or unwanted) effect of additional hydrogen pools. Moreover, most of these studies focused on the effect of

biomass e.g., in addition to the references reported in the MS, preliminary evaluation of biomass were presented in (Rivera Villarreyes et al., 2011); (Franz et al., 2013) presented an approach to isolate any hydrogen pools but soil moisture and showed the estimation of the crop biomass; (Baatz et al., 2015; Hawdon et al., 2014) introduced an empirical correction to account for biomass. In comparison to biomass, the effect of snow on CRNS signal has received much less attentions. Even if the first concepts were already introduced by (Desilets et al., 2010), a preliminary analysis was just presented by (Rivera Villarreyes et al., 2011) and only recently a study with longer time series of snow was published (Sigouin and Si, 2016). Other hydrogen pools were also addressed: e.g., the analysis of the role of little layer was discussed in detail by (Bogena et al., 2013) and in (Baroni and Oswald, 2015) we presented the first measurements for quantifying also the canopy interception. Overall I believe that all these experimental studies called for additional attentions on hydrogen pools than soil moisture. In this context, the present MS is the first modeling study where complex forest is simulated and the effect of several environmental factors are explored. For these reasons I think the MS could represent a good answer to those calls and the introduction of the MS should be rephrased accordingly.

AC15: We agree with the reviewer and we have change the wording as well as included a more detailed description of previous studies examining the effect of other pools of hydrogen than soil moisture on the neutron signal (Section 1.):

"To date, studies have primarily aimed to advance the cosmic-ray neutron soil moisture estimation method by determining correction models to remove the effect of other influencing pools of hydrogen.

Rosolem et al. (2013) examined the effect of atmospheric water vapor on the neutron intensity (10-100 eV; 1 eV = $1.6*10^{-19}$ J) using neutron transport modeling and determined a scheme to rescale the measured neutron intensity to reference conditions. For the preparation of cosmic-ray neutron data correction for changes in atmospheric water vapor is along with corrections for temporal variations in barometric pressure and incoming cosmic radiation a standard procedure (Zreda et al., 2012).

Most studies have focused on improving the $N_0$ calibration parameter used for soil moisture estimation at forest field sites but also at high-yielding crop field sites like maize. Bogena et al. (2013) demonstrated the importance of including the litter layer in the calibration for cosmic-ray neutron soil moisture estimation at field locations with a significant litter layer. The $N_0$ calibration parameter obtained from field measurements was found to decrease with increasing biomass (Rivera Villarreyes et al., 2013; Hornbuckle et al., 2012; Hawdon et al., 2014; Baatz et al., 2015). In order to account for this effect Baatz et al. (2015a) defined a correction model to remove the effect of biomass on the neutron intensity signal. A different approach was presented by Franz et al. (2013b). Here a universal calibration function was proposed where separate estimates of the various hydrogen pools are included for cosmic-ray neutron soil moisture estimation.

Few studies have explored the potential of using the cosmic-ray neutron method for additional applications. Desilets et al. (2010) distinguished snow and rain events using measurements of two neutron energy bands, and Sigouin and Si (2016) reported an inverse relationship between snow water equivalent and the neutron intensity measured using the moderated detector. Franz et al. (2013a) demonstrated an approach to isolate the effect of vegetation on the neutron intensity signal and estimate area average biomass water equivalent in

agreement with independent measurements. Finally, the signals of biomass and canopy interception on neutron intensity, measured using the moderated detector, have also been investigated by Baroni and Oswald (2015). They account the higher soil moisture estimated using the cosmic-ray neutron method compared to the up-scaled soil moisture measured at point-scale to be the impact of canopy interception and biomass. The two pools of hydrogen were then separated in accordance to their dynamics."

[3] I found the presentation of the results obtained with the reference model and the forest conceptualizations not clear (P11L24- P13L7). The Authors first stated about a remarkable agreement of the reference model (P12L12). Later they compared different forest conceptualizations and they found the best fit not to be unique (P13L2-4). Similarly they stated that they cannot determine which conceptualization is more realistic (P16l7-9). For this reason they conducted the SA using two conceptualizations. Overall, I believe that the mismatch should be clearly acknowledged from the beginning. Assuming that two forest conceptualizations are selected, the results of the SA could be then presented.

AC16: We agree. The first representation of the results obtained from modeling has been deleted as the same modeling results also are provided in the figure on forest canopy conceptualization. The results are now presented a little differently, the title "The reference" has been changed to "Gludsted Plantation" and the first part of section 3.1. has overall been reshaped. See Section 3.1.

 [4] The discussion of the results of the SA is not always clear and together with the 17 images I think the Readers are lost on the major findings of the study. In addition most of the discussion reported is a qualitative description of the figures. I would suggest searching for a way to sum up the results section (i.e., reducing the number of figures) where first the results of thermal and epithermal neutrons are discussed providing a quantitative comparison of the different effect of the environmental conditions explored. Secondly the ratio between thermal and epithermal is introduced and results are discussed for the factors that showed different response in thermal and epithermal neutrons.

AC17: We agree with referee #2. The number of figures has been reduced considerable (from 17 figures to 10 figures). The figures have either been changed and grouped together or erased. Additionally, the description of the measurements and the discussion of the results have been extended.

Changed and erased figures:

Figure 1: The figure have been change and now hold more information.

Figure 3: The results provided in Figure 3 are also provided in Figure 4. Therefore, Figure 3 has been erased.

Figure 7: The figure has been erased, but the modeling results are still provided in Table 5 (now Table 6) and discussed in the manuscript.

Figures 8, 10, 12 and 13: The figures have been lumped into one figure, and the plots have been changed. Now the ground and canopy level thermal and epithermal neutron intensity is provided instead of thermal and epithermal neutron height profiles.

Figures 14 and 15: The figures have been erased because the results were not very promising and the description of the results was tedious. The manuscript now holds a very short description on the difference between ground and canopy level thermal and epithermal neutron intensity. Here, references to Figures 6C (new fig.) and 6D (new fig.), and Table 6 (new table) are included.

New figure: A new figure has been included. The figure sum up the modeling results provided in Figure 16 and 17 and illustrates the relationship between biomass and ground level thermal-to-epithermal neutron ratio using Model Tree trunk, Foliage, Air and Model Foliage, respectively.

Description of measurements (Section 3.1.): "Overall, time-series and profile measurements provide similar results in agreement with theory. The thermal neutron intensity decreases considerable with height above ground surface and is at canopy level reduced by around 50% compared to at the ground level. The epithermal neutron intensity increases slightly with height and is around 10-15% higher at the canopy level compared to the ground level. Still, some differences are observed between the neutron height profiles measured in November 2013 and March 2014. The soil moisture was similar during the time of neutron profile detection and we expected the differences to be a result of different climate and weather conditions related to the seasons of detections (spring and fall)."

Discussion of the results:

Litter layer results (Section 3.3., Figure 6A (new fig.)): "The production rate of low-energy neutrons (<1 MeV) per incident high-energy neutron is higher for interactions with elements of higher atomic mass ($A^{2/3}$, where A is the atomic mass) (Zreda et al., 2012). Heavier elements are in particular found in mineral soil and an increase in the dry bulk density entails a higher production rate and therefore higher neutron intensity. The concentration of hydrogen is increased with an increased dry bulk density of litter material resulting in a greater moderation and absorption of neutrons, and as a consequence lower neutron intensities."

Biomass results (Section 3.5., Figures 6C and 6D (new figs.)): "The neutron intensity depends on how many neutrons are produced, down-scattered to lower energies and absorbed. Including biomass to a system increases the concentration of hydrogen and leads to reduced neutron intensity as the moderation and absorption is intensified. Despite this, increased thermal neutron intensity is provided with greater amounts of forest biomass. We hypothesize that forest biomass enhances the rate of moderation more than the rate of absorption. Thus higher thermal neutron intensity is obtained as the number of thermal neutrons generated by the moderation of epithermal neutrons exceeds the number of thermal neutrons absorbed. This behavior may be due to the large volume of air within the forest canopy. The probability of thermal neutrons to interact with elements within this space is low as the density of air is low."… "The epithermal neutrons produced in the ground escape to the air and are moderated by the biomass, resulting in reduced epithermal neutron intensity with greater amounts of biomass. All models provide in accordance to theory increasing epithermal neutron intensity with height, yet, the reduced steepness of the neutron height profiles with added biomass is unexplained. Oppositely to Model *Tree trunk, Foliage, Air*, the ground level thermal neutron intensity decreases with added biomass. This may be due to the elemental concentration. Here, no space is occupied by a material of very low elemental density and may lead to an increased absorption of thermal neutrons."

[5] It would be interesting to extend the MS with a discussion section where the overall results of the SA are summarized e.g., the advantages of using sensors at different heights, the advantages of using thermal and epithermal neutrons, the misfits of model and measurements and indication for further improvements. Concluding remarks could stress the potential use of CRNS for other applications but it would be interesting to extend the discussion also on the role of the spatial sensitivity of the sensor i.e., any estimation by CRNS is a spatial weighted value of the actual target (e.g., biomass).

AC18: Since the manuscript has been shortened considerably (both in terms of text and number of figures) we have chosen to keep a combined section of results and discussion. The discussion of the results has been extended (see AC17), and we believe that we have addressed the potential of using the difference between ground and canopy level neutron intensity and t/e ratio. Furthermore, suggestions on how to improve the comparability of measurements and modeling are also provided in this section. We hope that the section has been structured more adequately, and that the outcome of the study is clearer for the reader.

Finally, it is stated that for a good matching between measurements and simulations it was important the correcting factor (Page 7, L10-23). Since all the probes installed so far around the world does not account for that, it would be important to know what the implications are e.g., could we aspect the same sensitivity to environmental conditions when comparing bare and moderated counter instead of thermal and epithermal neutrons?

AC19: Thermal and epithermal neutrons are both sensitive to hydrogen but are also characterized by very different physical properties. We expect unique responses to environmental settings, and pure thermal and epithermal neutron signals are therefore important examining the effect of environmental impact on neutron transport. This is already stated in the introduction, yet, to emphasize the importance of pure thermal and epithermal neutron signals an additional line has been added to Section 2.2.2.:

"We expect thermal and epithermal neutrons to have unique responses to environmental properties and settings. Therefore, it is important to consider pure signals of thermal and epithermal neutrons, and not simply the raw neutron intensity signal measured by the bare and moderated detectors."

Specific comments

Page 1, L1-2: the title's focus on biomass and canopy interception is not entirely representative of the sensitivity analysis presented in the MS, which is broader. It should be rephrased accordingly.

AC20: The title has been changed:

"Cosmic-ray neutron transport at a forest field site: identifying the signature of biomass and canopy interception."

Page 1, L18: in my knowledge the effect of snow has received much less attention than other hydrogen pools. In addition, the analysis reported in the MS does not focus on biomass and interception but several other

factors are discussed. For this reason I would rephrase the sentence in ". . .soil moisture but several other hydrogen pools affect the signal".

AC21: The suggested change has been added to the manuscript.

Page 1, L22-31: in my opinion the presentation of the main results should be extended to honor also the other analyses provided in the MS (i.e., the role of the other factors).

AC22: More results are included in the abstract (underlined text=newly added text):

"A sensitivity analysis is performed to quantify the effect of soil moisture, complexity of soil matrix chemistry, forest litter, soil bulk density, canopy interception and forest biomass on thermal and epithermal neutron intensities at multiple height levels above the ground surface. Overall, modeled thermal and epithermal neutron intensities are in satisfactory agreement with measurements, yet, the forest canopy conceptualization is found to be significant for the modeling results. The results show that the effect of canopy interception, soil chemistry and dry bulk density of litter and mineral soil on neutron intensity is small, while the sensitivity to litter layer thickness and biomass in addition to soil moisture is found to be significant. The neutron intensity decreases with added litter layer thickness, especially for epithermal neutron energies. Forest biomass has a significant influence on the neutron intensity height profiles at the examined field site, altering both the shape of the profiles and the ground level thermal-to-epithermal neutron ratio."

Page 2, L12: the terminology used (static, quasi-static and dynamic) is too arbitrary. For a clearer discussion I would suggest presenting the hydrogen compartments in term of temporal scales (e.g., hours/days, season, years).

AC23: We choose this terminology because this is used for papers within the same field of research (see Franz et al., 2013 and Bogena et al., 2013).

Page 2, L15: for consistency I would mention here that the signal of hydrogen pools with low temporal dynamic (e.g., lattice water, SOC etc) is usually subtracted.

AC24: The work done by Franz et al., (2013) has been included in the introduction. They determined a universal calibration function for soil moisture estimation. The effect of other pools of hydrogen on the neutron intensity is included and the practice of subtracting the effect of lattice water and soil organic carbon origins from this work. We are not dealing with soil moisture estimation and we are for that reason not stating the approach directly as suggested by the reviewer.

Page 2, L18-25: see general comment #2 and the additional references reported to reshape the paragraph.

AC25: See AC15

Page 2, L26 – Page 3, L18: the sensitivity analysis focuses on several environmental conditions. In the light of reshaping the MS to honor this, I would say that these paragraphs are not relevant and could be omitted.

AC26: The focus on the environmental signature on the neutron transport has been amplified, yet, we chose to hold on to a special focus on canopy interception and biomass because these in particular are interesting as they form essential hydrological and ecological variables. Thus, we have not omitted the paragraphs.

Page 3, L19 – L34: summary of the aims of the paper and the methods should be rephrased to honor the actual analysis i.e., sensitivity analysis to environmental conditions to understand the role of different hydrogen pools.

AC27: The summary of the aims of the paper has been reshaped:

"Previous studies examining the effect of hydrogen on cosmic-ray neutron intensity has for most cases considered a single neutron energy range (neutron intensity measured using the moderated neutron detector) at a single height level (typically 1.5 m above the ground). Thermal and epithermal neutrons are both sensitive to hydrogen, but are characterized by very different physical properties resulting in unique responses to environmental settings and conditions at the immediate ground-atmosphere interface. For this reason, thermal and epithermal neutron intensity at multiple height levels above the ground surface are considered in this study.

The study is conducted at a forest field site using thermal and epithermal neutron measurements from bare and moderated detectors constrained with correction factor models (Andreasen et al., 2016) and modeling using the recognized and widely used Monte Carlo N-Particle transport code (MCNP) (Pelowitz, 2013). Neutron transport modeling of specific sites is limited and has only been performed for non-vegetated field sites (Franz et al., 2013b; Andreasen et al., 2016). In this context, forest sites are especially complex to conceptualize as the number of free parameters is relatively high (e.g. biomass, litter, soil chemistry, interception and the structure of the forest). Here, we first focus on modeling a forest field site. The model is developed from measured soil and vegetation parameters at the specific locality. The modeled neutron intensity profiles are evaluated against profile measurements on two different dates separated by five months, and also against time-series of neutron intensity measurements at two heights. Following, the forests environmental impact on thermal and epithermal neutron intensities are identified and quantified by applying a sensitivity analysis based on the model representative of the forest field site. In addition to improving the understanding of the environmental effect on neutron transport the focus is also on examining the potential of detecting intermediate scale canopy interception and biomass from cosmic-ray neutrons. Measurements at an agricultural field site with no biomass and at a heather field site with a smaller amount of biomass are used to underpin the influence of certain environmental variables (e.g., biomass, litter layer). To our knowledge this is the first study which provides a quantitative analysis of the potential of using the cosmic ray technique for estimation of interception and biomass."

Page 4, L1-2: the sentence is misleading: as reported in general comment #2 several publications were presented to estimate biomass. In (Baroni and Oswald, 2015) we have also presented the first measurements of canopy interception. Even in the case the Authors have any concerns about these studies, I think it would be part of the constructive advanced of the research field to integrate these opinions in the MS.

AC28: The introduction has been reworked and extended. See AC15.

Page 4, L3: this section 2 could be moved and integrated in the section 3.2.4 Field measurements.

AC29: Thank you for the suggestion. We have integrated section 2 in section 3.2.4.

Page 7, L10-23: for a clearer description of the results, the Authors could start the section making the list of the factors analyzed and referring here also to table 5. In addition the values presented in Table 5 could be plotted for easier comparison (e.g., bar plot).

AC30: Page 7, L10-23 is the section about pure thermal and epithermal neutron detection. Did you mean Page 11 L23?

We list and descried the factors analyzed in section 2.3.2. – the section just before Section 3 ("Results and discussion").

We considered the suggestion on presenting the values of Table 5 (now Table 6) in a different way. However, neither bar plots or other figure plots improved the presentation of the results. In addition, the table values are for most cases a supplement to figure presentations of the modeling results. Thus, we chose not to change the way we presented the values given in Table 5 (now Table 6).

Page 12, L4: I'm surprised: do you really think that the soil moisture profiles could explain such a difference? But in case it is relevant, why did you not evaluate this in the SA? Overall understanding the role of the different factors (i.e., environmental conditions) is the goal of the SA and of this paper.

AC31: We would like to test many more properties and settings, yet, this would make the manuscript tedious and overwhelming. We expect the differences in the neutron intensity profiles measured in November 2013 and March 2014 to be a result of different soil moisture profiles, different climate and weather conditions related to the seasons of detections (spring and fall) and measurement uncertainty. A sentence describing this is included in section 3.1.

Page 12, L12: I think the term "remarkable agreement" should be rephrased in the light of the overall discussion reported about the discrepancies and the inability to define which conceptualization is more realistic (e.g., P16L8). In addition I noticed that the thermal measurements show a regular decreasing from the ground to the canopy level. On the contrary the epithermal measurements show an inversion: the measurements decrease from ground to 5 meters and then start to increase regularly when moving to the canopy level. If I'm not wrong none of the models conceptualizations and the different environmental settings is able to reproduce this behavior. For this reason I think it could be an important result to discuss. Unfortunately this behavior is detected only for the profile measured on Mar-2014 while the measurements conducted on Nov- 2013 does not have these measurements in the plot: is this a mistake in plotting or really do you not have these measurements?

AC32: The decreasing epithermal neutron intensity from ground level to 5 m above the ground surface followed by increasing neutron intensities is expected to be a result of measurement uncertainties. In the beginning of Section 3.1 we describe how we rely more on time-series measurements and we discuss why the two neutron height profiles (November, 2013 and March, 2014) are different despite of similar soil moisture. We have added a few lines (Section 3.1.) on the overall behavior of the measured time-series and profiles of thermal and epithermal neutrons:

"Overall, time-series and profile measurements provide similar results in agreement with theory. The thermal neutron intensity decreases considerable with height above ground surface and is at canopy level reduced by around 50% compared to at the ground level. The epithermal neutron intensity increases slightly with height and is around 10-15% higher at the canopy level compared to the ground level."

Page 12, L26: if it is a SA the results should be discussed in term of sensitive or not sensitive. The term "satisfactorily" suggests that here you are still looking for a forest conceptualization that fits the profile measurements. See also general comment #3.

AC33: "modeled satisfactorily" has been changed to "in agreement with measurements".

Page 13, L8-19: this paragraph could be titled as a new section e.g., effect of soil moisture. Possibly, the analysis could be extended to explore the effect of soil moisture profiles (see also comment Page 12, L4).

AC34: Thank you for the suggestion. Section "Soil moisture" has been added.

Page 14, L25 - Page 15, L8: the presentation of the results jumps from the description of the thermal neutrons to the ratio i.e., epithermal neutrons are not described. For a clearer presentation I would suggest first to discuss both thermal and epithermal neutrons. Secondly, to introduce the use of the ratio explaining the reasons for doing that e.g., what do you expect to see with the use of the ratio instead of the single signal?

AC35: We have extended the description in Section "Canopy interception" a little (underlined text=newly added text):

"Except for a slight increase in ground level thermal neutron intensities with wetting of the forest canopy, no effect of canopy interception on ground and canopy level thermal and epithermal neutron intensity is observed."

Page 18, L21 - Page 19, L5: the discussion about the results obtained with the field locations of Voulund and Harrild is very limited and it refers to analysis presented in the submitted (and not available) paper of Anderson et al. (2016). Moreover I think that at the current status, these results do not provide any new insights on the present study. Either the Authors integrate better the description, the analyses and the results obtained in these locations, or in my opinion the results obtained based on these two sites could be completely omitted in this MS.

AC36: Andreasen et al. (2016) has since the submission of this paper to HESS been accepted (the reference is given in AC1). In our opinion the measurements of Voulund Farmland and Harrild Heathland is valuable for the manuscript. They confirm that litter and biomass increases the ground-level thermal-to-epithermal neutron ratio, and that the modeled values agree with measurements.

Page 18, L19: as discussed also in previous comments. I think the term "remarkable agreement" is misleading. On the contrary I think the misfits are interesting results to highlights providing the base for further studies.

AC37 (Page 19, L19): We have reworked the conclusion and changed the wording ("remarkable agreement" is out). See AC14.

Page 18, L22: before starting speaking about canopy interception, I would introduce also a summary of the role of the other hydrogen pools explored. This would better honor the SA reported in the MS.

AC38: We have added the results of the sensitivity analysis to the conclusion. See AC14.

Technical corrections

Page 4, L11: eV

AC39: The typo has been corrected (Section "Terminology").

Page 13, L26: the definition (4th order, 3rd order) of the chemical complexity are not self-explained. I would suggest instead the use in the table of other definitions e.g., (SOM+Gd+Root+??+SiO2) for the more complex and so on.

AC40: We choose to keep the current terminology ($4^{th}$ order, $3^{rd}$ order…), however, we have both edited the wording in Section 3.3. and the table caption.

Section 3.3.: "Soil organic matter, below-ground biomass, Gd and the chemical composition from XRF measurements are excluded one at the time (from *third* to *first order complexity*) and the final model includes a simple silica soil ($SiO_2$). The exact sensitivity of excluding the different components on ground and canopy level thermal and epithermal neutron intensity is quantified in Table 6 (see values in parentheses). Only the removal of soil organic matter (*third order complexity*) changes the neutron intensity significantly at Gludsted Plantation …"

Table 6 caption: "… Values provided in parentheses specifies the direct effect of one-by-one excluding soil organic matter (*third order complexity*), Gd (*second order complexity*), below ground biomass (*first order complexity*) and site specific major elements soil chemistry ($SiO_2$)."

Page 13, L28: what is cts? To be defined.

AC41: Cts is counts. This has been specified in text (Section 3.3.).

Page 15, L4: conditions instead of locations.

AC42: The suggested change has been added to the manuscript.

Figure 1: the domain represented in the figure is too extended and not well informative. I would suggest using this as a general overview but adding also a panel where the positions of the experimental sites are visualized with higher resolutions.

AC43: Figure 1 has been change, and now includes more information.

**Referee #3:**

General Comments:

In this paper the authors use a model to investigate how different model conceptualizations and different model parameters influence both thermal and epithermal neutron intensities from the ground surface up to a height of 35 m. They want to find out whether it is possible to use combined measurements of thermal and epithermal neutrons at ground level to determine both aboveground vegetation biomass (quasi-statically) and

canopy interception (dynamically). In order to do that they need to assess whether there are factors other than aboveground biomass and interception that alter the ratio of thermal to epithermal neutron intensity. I like the approach. It is novel to measure neutron intensities at different heights in a forest and it is novel to try to use the ratio of thermal to epithermal neutrons for biomass determination. Therefore the topic is interesting and the paper is well-suited for publication in HESS.

Still, there is room for improvement. In the end as a reader I felt a little lost on what are the actual outcomes from the study. It seems as if equifinality is a very big problem. Many of the investigated model setups and parameters seem to influence the detected neutron intensity profiles and therefore it is unclear which setup represents reality best. Unfortunately, the discussion section often lacks more detailed interpretations of the comparison of model results and measurements. Therefore the full potential of the study is not yet explored.

So my main point is that a refocus of the discussion section (away from just describing towards interpreting) would definitely improve the manuscript and the value for the reader.

AC44: Thank you for your comments and suggestions. We agree that the manuscript is a bit tedious and that the discussion of the results could be improved. We have revised the manuscript in order to ease the readability and clarify the focus.

Specific Comments:

p. 1, l. 1: Title: Since canopy interception only plays a minor role in the paper, I would suggest removing it from the title. You are investigating so many more things, like forest canopy representation, complexity of soil matrix chemistry, litter, soil bulk density. A more obvious choice for the title might be going along the line of forest canopy representation (since this part appears most prominently and novel when reading the manuscript). Also, posing a question in the title is not ideal, especially when you answer one part of it with no and the other part with yes.

AC45: We have changed the title to:

"Cosmic-ray neutron transport at a forest field site: identifying the signature of biomass and canopy interception."

p. 1, l. 31: It would be good to insert an explicit concluding statement into the abstract that answers the question you were posing in the title. ('Therefore we conclude that while there is potential to infer biomass from cosmic…  canopy interception cannot be inferred.')

AC46: The title has been changed and is no more containing a question. Still, concluding statement on the potential of quantifying canopy interception and biomass using cosmic ray neutron detection is relevant and a few lines has therefore been added to the abstract (underlined text=newly added text) (see also AC22):

"A sensitivity analysis is performed to quantify the effect of soil moisture, complexity of soil matrix chemistry, forest litter, soil bulk density, canopy interception and forest biomass on thermal and epithermal neutron

intensities at multiple height levels above the ground surface. Overall, modeled thermal and epithermal neutron intensities are in satisfactory agreement with measurements, yet, the forest canopy conceptualization is found to be significant for the modeling results. The results show that the effect of canopy interception, soil chemistry and dry bulk density of litter and mineral soil on neutron intensity is small, while the sensitivity to litter layer thickness and biomass in addition to soil moisture is found to be significant. The neutron intensity decreases with added litter layer thickness, especially for epithermal neutron energies. Forest biomass has a significant influence on the neutron intensity height profiles at the examined field site, altering both the shape of the profiles and the ground level thermal-to-epithermal neutron ratio."

p. 2, l. 2: '..relativeLY high concentration CLOSE TO THE LAND SURFACE,…'

AC47: The suggested change has been added to the manuscript.

p. 2, l. 2-10: I would reorder this paragraph. Start with the role of soil moisture and the difficulties of its detection. Then introduce cosmic-ray neutrons and the detector before mentioning its footprint in line 7.

AC48: We have reordered the section following the suggestions of Referee #3 (see Section 1.).

p. 2, l. 13: In Table 1 you use the word 'transient', here you say 'dynamically'.

AC49: We have change the word "transient" to the word "dynamic" in Table 1.

p. 2, l. 13-14: Try to categorize this list. 'Hydrogen is stored statically in water in soil minerals and buildings/roads, quasi-statically in…'

AC50: The suggested change has been added to the manuscript.

p. 2, l. 31: 'HOWEVER, the spatial scale of measurement…'

AC51: The suggested change has been added to the manuscript.

p. 3, l. 27: '…we PERFORM a sensitivity analysis…'.

AC52: The paragraph has been changed:

"The study is conducted at a forest field site using thermal and epithermal neutron measurements from bare and moderated detectors constrained with correction factor models (Andreasen et al., 2016) and modeling using the recognized and widely used Monte Carlo N-Particle transport code (MCNP) (Pelowitz, 2013). Neutron transport modeling of specific sites is limited and has only been performed for non-vegetated field sites (Franz et al., 2013b; Andreasen et al., 2016). In this context, forest sites are especially complex to conceptualize as the number of free parameters is relatively high (e.g. biomass, litter, soil chemistry, interception and the structure of the forest). Here, we first focus on modeling a forest field site. The model is developed from measured soil and vegetation parameters at the specific locality. The modeled neutron intensity profiles are evaluated against profile measurements on two different dates separated by five months, and also against time-series of neutron

intensity measurements at two heights. Following, the forests environmental impact on thermal and epithermal neutron intensities are identified and quantified by applying a sensitivity analysis based on the model representative of the forest field site. In addition to improving the understanding of the environmental effect on neutron transport the focus is also on examining the potential of detecting intermediate scale canopy interception and biomass from cosmic-ray neutrons. Measurements at an agricultural field site with no biomass and at a heather field site with a smaller amount of biomass are used to underpin the influence of certain environmental variables (e.g., biomass, litter layer). To our knowledge this is the first study which provides a quantitative analysis of the potential of using the cosmic ray technique for estimation of interception and biomass."

p. 3, l. 28: Only to look at their effect on MODELED thermal and epithermal neutron intensity? Or also to make statements about their effect on ACTUAL thermal and epithermal neutron intensity?

AC53: The paragraph has been changed. See AC52.

p. 4, l. 16: Could you shortly introduce what this 'root-to-shoot ratio' is?

AC54: We have included an explanation to the sentence (the added text is underlined):

"The dry below-ground biomass was calculated to be 25 t/ha using a root-to-shoot ratio (the weight of the roots to the weight of the aerial part of the plant) for Norway spruce of 0.25 (Levy et al., 2004)."

p. 4, l. 16: Information? Be more specific.

AC55: Some examples of the sort of information provided by The Danish Nature Agency have been included in the sentence (the added text is underlined):

"Information on the vegetation at the forest field site (e.g. tree species, ages, heights and trunk diameters) is acquired from a register managed by The Danish Nature Agency (representative of the 2012 conditions); see Table 2."

p. 5, l. 2: Why random soil samples? A composite sample representing mean soil properties would have been much more representative of the soils within the footprint of the sensor given small-scale variability.

AC56: The field sites are located on an outwash plain from the last glaciation composed of sandy soil. We agree that a composite soil sample representing the mean conditions would have been more appropriate, however, the soil is very homogeneous and we are quite confident that two random soil samples are sufficient. The homogeneity of the soil is evident comparing the results of the XRF analysis on two random soil samples collected in 20-25 cm depth at Harrild Heathland and Gludsted Plantation. The two field sites are separated by approximately 10 km and have very similar soil chemistry (see below). The soil chemistry of Voulund Farmland is not included as it due to farming practices contains a wider range of elements.

| | Gldusted Plantation [%] | Harrild Heathland [%] |
|---|---|---|
| O | 52.78 | 52.76 |

| | | |
|---|---|---|
| Si | 44.86 | 44.71 |
| Al | 1.54 | 1.74 |
| K | 0.53 | 0.56 |
| Ti | 0.29 | 0.23 |

p. 5, l. 24: What do you mean by: '…is observed VISUALLY…'?

AC57: The line has been reworked (Section 2.2.4.):

"The organic rich litter layer is found to be around 10 cm thick during soil sampling field campaigns at the field site."

p. 5, l. 26: Do you mean that the hardpan-layer hinders percolation to deeper depths?

AC58: Yes. The sentence has been reworked (Section 2.2.4.):

"Due to podsolization a low permeable hardpan-layer hindering percolation to deeper depths is present at around 25-30 cm depth."

p. 6, l. 12: represent might not be the right word here. Maybe 'detect' or 'be sensitive to'?

AC59: "Represent" has been replaced with "is sensitive to" (Section 2.1.).

p. 6, l. 13: What do you want to express when you write: 'Despite this fact…'?

AC60: The sentence has been reworked:

"Here, the term epithermal neutrons will be used for both measured neutrons of energies above 0.5 eV and modeled neutrons of energies 10 – 1000 eV."

p. 7, l. 5: The term 'epithermal' includes 'fast', no? So you don't need to say '…fast and epithermal…'.

AC61: That is true. "Fast" has been erased.

p. 7, l. 7: Why do you believe in this minor effect on your results?

AC62: Yes this should be addressed, and was addressed by Andreasen et al. (2016). Therefore, the reference has been added at the end of the sentence.

Andreasen et al. (2016): "Preliminary modeling results by the authors and R. Rosolem (personal communication, 2015) suggest that water vapor only has a minor effect on the thermal neutron intensity measured near the land surface. This is in agreement with earlier studies of Bethe et al. [1940] and Lockwood and Yingst [1956]. However, water vapor corrections might be required for thermal neutron intensities collected high above the ground surface, and future work should address this issue."

p. 7, l. 19-20: What are these correction factor models, when exactly where they applied and how did the output of these models look compared to the cadmium-difference model?

AC63: The neutron energy correction models are described in Andreasen et al. (2016) (see AC1) and in Section 2.2.2. Additionally, a few sentences have been added to Section 2.2.4 specifying when the neutron energy correction models where applied:

"In order to obtain comparability between measurements and modeling pure thermal and epithermal neutron signals were estimated using neutron energy correction models on measurements from bare and moderated detectors, respectively. The neutron energy correction models were both used on time-series and neutron height profile measurements."

"In order to obtain pure thermal and epithermal neutron height profiles the neutron energy correction models were applied."

p. 7, l. 32-34: The fact that the environmental conditions at the field sites are fairly homogeneous is no explanation for your assumption that the neutron intensities measured by the two different detectors can be compared. Please elaborate.

AC64: The paragraph has been reworked:

"The potential mismatch in the footprint of the bare and the moderated detectors is a concern when combining the neutron intensity measurements. Nevertheless, the environmental conditions at the field sites are fairly homogeneous and although the footprint might be different as a first approximation we assume the neutron intensity measured using the bare and the moderated detector are comparable."

p. 9, l. 5-32: What about the sub-canopy structure of real forests? With a lot of the leaves and branch biomass a couple of meters above the ground and only the trunks with a lot of air in between near the ground surface. Would you expect the same outcome? How could this impact your results? It would be good to discuss this somewhere.

AC65: We have not modeled a forest with vertical variation in material and density, yet, it would be interesting to examine in the future. Here, the neutron transport was found to be sensitive to the conceptualization of the forest and we therefore expect that further advancement will have an effect too. However, we have a hard time predicting the effect on neutron transport specifying the sub-canopy structure of the forest. We have included a few suggestions in Section 3.1 on how to advance the forest conceptualization (see also AC5):

"Improved comparability to measurements may be obtained by advancing the forest canopy conceptualization. Currently, one tree is defined and repeated throughout the model domain. The trees are placed in even rows and the same settings are applied from the ground surface to 25 m height. In order to advance the forest canopy conceptualization, trees of different heights and diameters could be included, and the placement of the trees could be more according to the actual placement of trees at the forest field site. Additionally, variability

in tree trunk diameter, foliage density and volume with height above the ground surface could be implemented."

p. 10, l. 20-21: Rephrase. Maybe something like: 'The thermal and epithermal neutron intensity is both a product of hydrogen abundance as well as elemental composition…'.

AC66: The suggested change has been added to the manuscript.

p. 12, l. 30: From here on I will ask the question 'Why?' whenever I would like to see a more detailed discussion of one of your results/observations. Throughout the discussion section there are instances where you observe and describe your results without giving a proper (attempt of) interpretation. For example here you state that '…the neutron intensity profiles of the simpler forest canopy conceptualization…   is less steep and is the only model providing an epithermal neutron intensity profile within the daily ranges of the time-series measurements…'. Still there is no explanation on why this could be the case.

AC67: This is a very valid point, and we have sought to explain the effect of alterations in the environmental settings on thermal and epithermal neutron intensity. We do not always have an answer, but have in those cases provided some suggestions/thoughts on the measurements and modeling results.

p. 13, l. 19: Why?

AC68: Unfortunately, we have no explanation to why this is. The different results of the two model-setups could from measurements potentially have clarified which of the two model-setup are more appropriate, however, this was unfortunately not the case as both models provide neutron intensities in fairly good agreement with measurements. We have including this consideration in Section 3.2.:

"Neutron intensity at dry and wet soil conditions is represented by the range of time-series neutron intensity measurements. Overall, the modeled neutron intensities are within the measurement range and the more appropriate model-setup for Gludsted Plantation is not obvious from the modeling results."

p. 14, l. 17: Why?

AC69: An explanation has been added to Section 3.3.:

"The production rate of low-energy neutrons (<1 MeV) per incident high-energy neutron is higher for interactions with elements of higher atomic mass ($A^{2/3}$, where A is the atomic mass) (Zreda et al., 2012). Heavier elements are in particular found in mineral soil and an increase in the dry bulk density entails a higher production rate and therefore higher neutron intensity. The concentration of hydrogen is increased with an increased dry bulk density of litter material resulting in a greater moderation and absorption of neutrons, and as a consequence lower neutron intensities."

p. 14, l. 19: How can the mineral soil act as a producer of epithermal neutrons? Thermal neutrons would have to be accelerated to become epithermal. How does this happen?

AC70: An explanation has been added to Section 3.3. See AC69.

p. 15, l. 7: Move '…from the calculation in the previous section…' to the beginning or the end of the sentence.

AC71: The sentences have been rephrased (Section 3.4.):

"We choose not to include measurements in the figure because the measurement uncertainty at a relevant integration time is greater than the signal of canopy interception."

p. 15, l. 31-32: Why?

AC72: We have added a few sentences to explain the response of thermal and epithermal neutrons to increasing amounts of biomass using Model *Tree trunk, Foliage, Air*:

"The neutron intensity depends on how many neutrons are produced, down-scattered to lower energies and absorbed. Including biomass to a system increases the concentration of hydrogen and leads to reduced neutron intensity as the moderation and absorption is intensified. Despite this, increased thermal neutron intensity is provided with greater amounts of forest biomass. We hypothesize that forest biomass enhances the rate of moderation more than the rate of absorption. Thus higher thermal neutron intensity is obtained as the number of thermal neutrons generated by the moderation of epithermal neutrons exceeds the number of thermal neutrons absorbed. This behavior may be due to the large volume of air within the forest canopy. The probability of thermal neutrons to interact with elements within this space is low as the density of air is low."

p. 16, l. 5-6: Why?

AC73: We have added a few sentences to explain the response of thermal and epithermal neutrons to increasing amounts of biomass using Model *Foliage*:

 "The epithermal neutrons produced in the ground escape to the air and are moderated by the biomass, resulting in reduced epithermal neutron intensity with greater amounts of biomass. All models provide in accordance to theory increasing epithermal neutron intensity with height, yet, the reduced steepness of the neutron height profiles with added biomass is unexplained. Oppositely to Model Tree trunk, Foliage, Air, the ground level thermal neutron intensity decreases with added biomass. This may be due to the elemental concentration. Here, no space is occupied by a material of very low elemental density and may lead to an increased absorption of thermal neutrons."

p. 16, l. 26-27: Why?

AC74: In order to focus the paper and improve the readability the section on the difference between ground and canopy level neutron intensity has been reduced considerably, and the sentence on p. 16, l. 26-27 has been erased.

p. 17, l. 7-17: So would you say that this model representation is better than the more complex one? It certainly fits better to your observed data. What does it mean that the average conditions (without separate trunk, foliage, air) perform better? It should be the other way around, no?

AC75: The ground level thermal-to-epithermal neutron ratio was found to be more appropriate and convenient in terms of biomass determination. Thus, in order to focus the paper and improve the readability the attention of the difference between ground and canopy level thermal and epithermal neutron intensity, respectively, has been reduce markedly. The conditions mentioned by Referee #3 are not included in the manuscript anymore.

p. 17, l. 22: Do you maybe mean '…prevailING at the field site.'

AC76: The line has been erased (see AC6).

p. 17, l. 31-32: Why?

AC77: A line has been added to the sentence (see underlined part):

"Drying or wetting of soil change the thermal and epithermal neutron intensity proportionally and the ratios are accordingly found to be independent of changes in the ground level thermal neutron intensity, the ground level epithermal neutron intensity and volumetric soil moisture."

p. 18, l. 7-12: Is that an indication that this more complex model is a more realistic representation of the forest environment? How is this observation compatible with the previous observation that shows the better fit of the less complex model when comparing the differences between ground and canopy level thermal and epithermal neutron intensity?

AC78: The results on the difference between ground and canopy level neutron intensity were ambiguous and most discussion and figures on this has been removed to ease the readability and hopefully making the manuscript less comprehensive. Overall, the Model *Tree trunk, Foliage, Air* seems to perform better. The two-year-average of ground level t/e ratio fits the biomass of 100 t/ha estimated for Gludsted Plantation using lidar, and the range of measurements is overall in agreement with the standard deviation of the estimated biomass. Still, more work needs to be done as neither the Model *Tree trunk, Foliage, Air* and Model *Foliage* are fully in agreement with measurements. In order to do this, we have to look at the three points stated a little later in the same paragraph. We have not added any to answer the question asked by referee #3 in the manuscript as most of the section on the difference between ground and canopy level neutron intensity has been removed.

p. 18, l. 13-20: How would each of these 3 factors influence the modeled ratios?

AC79: Of these three factors only shortcomings in the model setup would affect the modeled ratios. This has been specified in the text:

"A model including a sufficient representation of the field site will provide neutron height profiles and t/e ratios more representative of the real conditions…"

p. 19, l. 3: Should the amount of biomass not be slightly larger for the Heathland site compared to the non-vegetated Gludsted plantation?

AC80: Yes. We have rephrased the paragraph:

"Both field sites have a considerable layer of litter, and the slightly higher t/e ratio relative to the non-vegetated Gludsted Plantation may be due to biomass in the form of grasses, heather plants and bushes present at Harrild Heathland."

p. 19, l. 6: It would be helpful to introduce an abbreviation for the term 'thermal-to-epithermal ratio' somewhere at the beginning (Rt/e) and use it throughout the manuscript.

AC81: Good idea. We have included the abbreviation "t/e" in the manuscript.

Figures & Tables:

Figure 1: Provide a map that zooms in onto your study area with a little more detail and move the current overview map of Denmark into one of the corners of the new map.

AC82: Figure 1 has been changed.

Figure 3-10, 12-13: Remove the line in the legend in front of 'Canopy surface model'. I was looking for it but it is not in the actual figure, is it? Maybe just call it 'Modeled' in comparison to 'Measured'.

AC83: The line in front of 'Canopy surface' was supposed to be dashed, and explains the horizontal dashed lined at 25 m height above the ground surface in the figures. I have edited the three figures providing results on neutron height profiles. Now the lines in the legend of the figures are all dashed.

Technical Corrections:

p. 1, l. 25: 'minor' is no adverb. Maybe use 'insignificantly'.

AC84: The suggested change has been added to the manuscript.

p. 1, l. 27: siteS

AC85: The suggested change has been added to the manuscript.

p. 4, l. 5: '…within THE Skjern River…'.

AC86: The suggested change has been added to the manuscript.

AC89: The reference is now included in the text (Section "Introduction").

[revised manuscript text omitted]

*Figure 3 is inserted here*

We choose to rely mostly on the Overall, time-series and profile measurements, as provide similar results in agreement with theory. The thermal neutron intensity decreases considerable with height above ground surface and is at canopy level reduced by around 50% compared to at the ground level. The epithermal neutron intensity increases slightly with height and is around 10-15% higher at the neutron canopy level compared to the ground level. Still, some differences are observed between the neutron height profiles are very different despite of similar measured in November 2013 and March 2014. The soil moisture was similar 
[revised manuscript text omitted]

---

## Author Response (AR3)

**22 February, 2017**

**Referee #2:**

Most of the comments were addressed in the revision process and the manuscript has been significantly improved accordingly. I have only minor comments that could be considered for final further improvements.

The comments refer to page (p) and line (l) of the paper without track changes or directly to the Author's comments (AC) provided in the response letter.

Comments:

p1,l1. Title has been changed but I regret to say that it is still not clear to me why the Authors want to keep the terms biomass and canopy interception there. The study explores 7 factors. Three showed to be important and 4 not. Biomass is one of the important factors while canopy (at least in this study) is not. I actually agree that biomass and canopy interception are two important hydrological and ecological variables (AC26). But then, why not using the litter layer in the title? It is also an important hydrological and ecological variable and it was even found to affect much more the neutron intensity than the canopy interception. Overall, recalling the title of section 2.3.2 sensitivity to environmental conditions, I think the best title to honor the study could be: Cosmic-Ray neutron transport at a forest field site: sensitivity analysis to different environmental conditions.

AC1 (Author comment # 1): Thank you for your comments and corrections. We very much appreciate you taking the time to make the review of our manuscript.

We see the point and have therefore changed the title to: "Cosmic-ray neutron transport at a forest field site: the sensitivity to various environmental conditions with focus on biomass and canopy interception"

p2,l8. Reword This method… in…This neutron intensity has been used for estimating soil moisture…

AC2: The first few lines of the abstract have been reworked:

"This correlation forms the base of the cosmic-ray neutron soil moisture estimation method. The method is, however, complicated by the fact that several hydrogen pools other than soil moisture affect the neutron intensity."

p2,l25. It would be nice concluding the abstract with a nice take-home message e.g., Overall, the results suggest the potentiality to use the ratio signal to discriminate different hydrogen contributions.

AC3: Thanks for the suggestion. A take-home message has been added in the end of the abstract:

"Overall, the results suggest a potential to use ground level thermal-to-epithermal neutron ratios to discriminate the effect of different hydrogen contributions on the neutron signal."

p3,l19. Rivera Villarreyes at al., 2011 (not 2013).

AC4: Thank you for me aware of this. I have corrected this in the text and in the reference list.

p5. I would start the page with the statement of the objectives (i.e., improving the understanding of the environmental effect on Cosmic-Ray neutron intensity based on neutron transport modelling).

AC5: I have added a small paragraph in order to illuminate the objectives of the study:

"This study is an initial step towards reaching the overall objective of improving the cosmic-ray neutron soil moisture estimation method, especially at field locations with several pools of hydrogen. Furthermore, we wish to investigate the potential of biomass and canopy interception estimation using the cosmic-ray neutron intensity measurements. Here, the aim is to address this goal using only cosmic-ray neutron intensity measurements and not auxiliary information (e.g., biomass measurements using allometric models and tree surveys)."

p5,l21-22. After all the experimental studies cited in the introduction dealing with biomass and interception (e.g., Baatz et al., 2015; Baroni and Oswald, 2015; Franz et al., 2013; Hawdon et al., 2014), this statement is not justified. Either you state that this is the first MODELLING study which provides a quantitative analysis or you rather remove the sentence.
I also underlines that a recent paper (Tian et al., 2016) was recently published presenting the use of the ratio (bare/moderator counts) for biomass and snow correction. Due to the relevance of the paper in relation of the present study, I think is worth to extend the introduction also by that.

AC6: The sentence has been modified to:

"To our knowledge this is the first study based on both measurements and modeling which provides a quantitative analysis of the potential of using the cosmic ray technique for estimation of interception and biomass."

Moreover, I have included a few sentences on the work done by Tian et al. (2016):

"A similar correcting approach to improve the cosmic-ray neutron soil moisture estimation method by removing the influence of biomass and snow was presented by Tian et al. (2016). However, the study distinguishes itself by considering the ratio of the neutron intensity measured by the bare detector and the moderated detector instead of the effect on the N0 parameter."

Tian, Z., Li, Z., Liu, G., Li, B., Ren, T., 2016. Soil Water Content Determination with Cosmic-ray Neutron Sensor: Correcting Aboveground Hydrogen Effects with Thermal/Fast Neutron Ratio. Journal of Hydrology.

doi:10.1016/j.jhydrol.2016.07.004

p9,l17. Please specify always that is volumetric soil moisture and add the units [m3 m-3]. This comment holds throughout the manuscript.

AC7: Thank you for reminding me of this. Volumetric soil moisture and the unit m3 m-3 have now been specified throughout the manuscript.

p13,l13. In agreement with theory. Could you elaborate more? Why is it expected that the thermal neutron intensity decreases while the epithermal increases?

AC8: I have added a paragraph in Section 2.1 explaining the different behavior of thermal and epithermal neutrons. The title of the section has been changed from "Terminology" to "Terminology and neutron energies" because of the additions to the section. A reference to the explanation in Section 2.1 has been added to Section 3.1.: "(see Section 2.1.)".

Text added to Section 2.1.:

"The probability of absorption reactions is greater for thermal neutrons, while the probability of scattering reactions is greater for neutrons of epithermal energies. For this reason thermal and epithermal neutron height profiles are very different at the ground-atmosphere interface. The epithermal neutron intensity increases with height above the ground surface as the neutrons at higher elevations have been scattered less than neutrons closer to the ground surface. The production rate of thermal neutrons is high in the soil and low in the air. This is related to the high density of the soil and the low density of air. The absorption rate of thermal neutrons is significant in both the ground and in the air. In the air, this is due to the presences of nitrogen. This results in a decreasing thermal neutron intensity with height until approximately 150 m at which point the thermal neutron intensity is unaffected by the soil. Above this point the thermal neutron intensity will increase with height following a similar curve as neutrons of higher energies."

p19,l14. Figure 16 and 17. You mean Figure 8 and 9.

AC9: Thank you. Yes I did indeed mean Figures 8 and 9.

References list

Rivera Villarreyes et al., 2013 is in HESSD and it was not published.

The following two studies are cited in the text but not listed:

Baroni, G., Oswald, S.E., 2015. A scaling approach for the assessment of biomass changes and rainfall

interception using cosmic-ray neutron sensing. Journal of Hydrology 525, 264–276. doi:10.1016/j.jhydrol.2015.03.053

Rivera Villarreyes, C.A., Baroni, G., Oswald, S.E., 2011. Integral quantification of seasonal soil moisture changes in farmland by cosmic-ray neutrons. Hydrology and Earth System Sciences 15, 3843–3859. doi:10.5194/hess-15-3843-2011.

AC10: Thank you for making me aware of this. I have added the two studies to the reference list.

Table 1: You choose (see AC23) the terminology in table 1 because this is used for papers within the same field of research. Still, I think it could be easily improved e.g., "years" instead of "static", "season" instead of "quasi static" and "hours" instead of Dynamic".

AC11: An extra row has been included in Table 1 in order to also specify "yearly", "sub-yearly" and "daily".

AC32 i.e., The decreasing epithermal neutron intensity from ground level to 5 m above the ground surface followed by increasing neutron intensities is expected to be a result of measurement uncertainties. I think this statement is worth being included in the manuscript.

AC12: We agree. Following sentences has been added to section 3.1.:

"Note that a decrease in the epithermal neutron intensity from the ground level to 5 m above the ground surface was measured in March 2014. This is in disagreement with theory (see Section 2.1.) and is expected to be a result of measurement uncertainties."

**Referee #3:**

You can tell from the authors' response and the revised manuscript that they really made an effort to take on the suggested changes from the reviews. That is very much appreciated.
The structure of the manuscript has improved considerably. It is much clearer now and the results are explained in a more comprehensible way than before. Also, the authors added more details and explanations of their findings.
Within the text there is still a frustrating amount of spelling, syntax and style errors. Given the amount of co-authors well capable of detecting and correcting these errors, this fact is surprising. It feels like they left these errors for the reviewers to point out and correct.
The story of the abstract could be improved for better readability and the use of the word 'sensitivity' should be checked carefully at each occurrence.
For these reasons I recommend this manuscript for publication pending some more revisions.

AC13: I´m sorry about the spelling errors. It was not our intention to leave the job of correcting the spelling and syntax errors to you and the other reviewers. I appreciate your work and hopefully we have been able to do a better job with this version of the manuscript.

Specific Comments:

Abstract: The abstract could tell a better story by giving some more information and connecting the individual statements better. You write for example: 'This method has been used for measuring soil moisture but several other hydrogen pools affect the signal.'

-Which method?

-The 'but' does not fit in this sentence

-Why is it a problem that other pools affect the signal?

Maybe it would be better to write something along the lines of 'Relating cosmic ray neutron counts to the presence of (more or fewer) water molecules within the footprint of the sensor is used for quantifying/measuring the hydrogen content of soil moisture. This is, however, complicated by the fact that several other hydrogen pools affect the signal.'

It would be much easier for the reader to follow.

AC14: The first part of the abstract has been reworked:

"This correlation forms the base of the cosmic-ray neutron soil moisture estimation method. The method is, however, complicated by the fact that several hydrogen pools other than soil moisture affect the neutron intensity. In order to improve the cosmic-ray neutron soil moisture estimation method and explore the potential for additional applications knowledge about the environmental effect on cosmic-ray neutron intensity is essential (e.g., the effect of vegetation, litter layer and soil type). In this study the environmental effect is examined by performing a sensitivity analysis using neutron transport modeling."

You should state your intentions early in the abstract: Why have you done this study? You tell us what you have done but with little explanation on your actual goals.

Appropriate would be something like: 'In order to shed light on the influence of several other hydrogen pools on the measured signal we performed a sensitivity analysis using a neutron transport model with various representations of…'.

Maybe add a concluding statement to the end. 'While biomass affects … considerably, we found canopy interception to be of minor importance…'.

AC15:

A line on the intention of the study has been added to the abstract:

"In order to improve the cosmic-ray neutron soil moisture estimation method and explore the potential for additional applications knowledge about the environmental effect on cosmic-ray neutron intensity is essential (e.g., the effect of vegetation, litter layer and soil type). In this study the environmental effect is examined by performing a sensitivity analysis using neutron transport modeling."

A concluding remark has also been added in the end of the section:

"Overall, the results suggest a potential to use ground level thermal-to-epithermal neutron ratios to discriminate the effect of different hydrogen contributions on the neutron signal."

Page 2, Line 28: '…at intermediate spatial scales…'

AC16: The suggested change has been added to the manuscript.

Page 3, Line 12: 'For the preparation of cosmic-ray neutron data THE correction for changes in atmospheric water vapor is A STANDARD PROCEDURE along with corrections for temporal variations in barometric pressure and incoming cosmic radiation…'
The native English speakers amongst your co-authors could have done a better job in correcting some of the language and style. Maybe they can carefully read the whole paper one more time.

AC17: The suggested change has been added to the manuscript.

Page 3, Line 20: What about studies that call for a modification of the calibration function itself (e.g. Iwema et al. 2015, Heidbüchel et al. 2016)? They argue that the function must be adapted to the local environment (more biomass vs. less biomass, higher variability in soil properties with depth vs. lower variability, etc.).

AC18: Thank you for reminding me of these two studies. A paragraph describing their findings and considerations has been added:

"Iwema et al. (2015) and Heidbüchel et al. (2016) applied the N0 calibration function and obtained improved cosmic-ray neutron soil moisture estimates by performing more than one calibration campaign per field site and defining a site-specific calibration function. Heidbüchel et al. (2016) speculate that the curve shape of the standard N0 calibration function is insufficient at the studied forest field site because of the presence of a litter layer and spatially heterogeneous soil moisture conditions within the neutron detector footprint."

Page 5, Line 5-6: Thermal and epithermal neutron intensities are considered in order to explore the possibilities they offer when observed at the same time (not because they are characterized by very different physical properties – that is the prerequisite for using them in combination).

AC19: We have changed the wording of the paragraph:

"Previous studies examining the effect of hydrogen on cosmic-ray neutron intensity has for most cases considered a single neutron energy range (neutron intensity measured using the moderated neutron detector) at a single height level (typically 1.5 m above the ground). Thermal and epithermal neutrons are both sensitive to hydrogen. However, they are characterized by very different physical properties and reaction patterns resulting in different height profiles, as well as unique responses to environmental settings at the immediate ground-atmosphere interface. For this reason, thermal and epithermal neutron intensity at multiple height levels above the ground surface are considered in this study as the combination may provide additional information."

Page 5, Line 15: What exactly is the 'environmental effect'?
AC20: This has been specified:

"The environmental impact refers to the effect of the specific properties and settings of the field site on neutron transport. This includes vegetation, litter, soil composition and layers, and canopy interception."

Page 5: Line 29-31: No dots after 'eV'.

AC21: The dots have been erased.

Page 6, Line 3: '…are sensitive…'.

AC22: The suggested change has been added to the manuscript.

Page 6, Line 22: Better write: 'Nonetheless, still a large proportion originates from below 0.5 eV (approximately…'.

AC23: The suggested change has been added to the manuscript.

Page 6, Line 30: Any clues on why you believe that?

AC24: Preliminary modeling suggests that water vapor only has a minor effect on the thermal neutron intensity. This reasoning has been added to the text:

"From preliminary modeling conducted by the authors and R. Rosolem (personal communication, 2015)…"

Page 7, Line 14: Potential problems could arise here. If you use a model to correct/infer pure thermal and pure epithermal neutron intensities, the observed differences can always be explained by this correction model and all the assumptions you made when setting it up.

AC25: You are correct. This is a general problem and should be kept in mind every time a correction model is applied. At the moment we are working on an additional method for obtaining comparable measured and modeled neutron intensity as the correction models are unique for specific neutron detectors and specific environments. However, this is beyond the scope of this paper.

Page 8, Line 11: '…the Danish…'.

AC26: The suggested change has been added to the manuscript.

Page 9, Line 17: Maybe add some units ($m^3 m^{-3}$)?

AC27: See comment AC7.

Page 11, Line 27: You should definitely add some units ($m^3 m^{-3}$).

AC28: See comment AC7.

Page 11, Line 32: You don't have to repeat the whole 'The Gludstedt Plantation reference model'. The second time, just say 'It'.

AC29: The suggested change has been added to the manuscript.

Page 13, Line 10: You '…rely mostly on the time series measurements…' for what purpose exactly?

AC30: Thank you for pointing this out. I forgot to specify that we because of the lower measurement uncertainty mostly rely on **daily averages** of time-series measurements. I have specified this in the sentence:

 "Accordingly we choose to rely mostly on the daily averages of time-series measurements"

Page 13, Line 13-14: Did you specify this theory anywhere? For example, why does the thermal neutron intensity theoretically decrease with height?

AC31: See AC8.

Page 13, Line 14: considerably

AC32: The spelling has been corrected.

Page 14, Line 1: What do you mean by 'Here'.

AC33: "Here" has been changed to "In this study".

Page 14, Line 1: performED

AC34: The spelling has been corrected.

Page 14, Line 1: What do you mean by 'occasionally'?

AC35: I have changed the wording of the paragraph:

"In this study, a sensitivity analysis is performed using the most complex model to examine the effect of soil moisture, soil dry bulk density and composition, litter and mineral soil layer thickness, canopy interception and biomass on the thermal and epithermal neutron transport at the immediate ground-atmosphere interface. Since the most appropriate forest canopy conceptualization is not obvious from Fig. 3 the simplest forest canopy conceptualization was also used to examine the effect of soil moisture and biomass on the neutron transport."

Page 14, Line 12-13: Please rearrange the sentence. This way it is confusing.

AC36: The sentence has been rearranged: "As expected, the thermal and epithermal neutron intensity decreases with increasing soil moisture (Table 6, Figs. 4 and 5)."

Page 14, Line 18-20: The two sentences contradict each other (similar vs. more sensitive).

AC37: The wording has been changed: "The model with a simple forest canopy conceptualization provides thermal and epithermal neutron intensities slightly more sensitive to soil moisture (Figure 5)."

Page 14, Line 20: sensitive

AC38: The spelling has been corrected.

Page 15, Line 5: Additionally, SINCE(?) a considerable range of (what kind of?) values is measured within the footprint of the neutron detector the sensitivity OF NEUTRON INTENSITY to litter and mineral soils dry bulk density is examined…

AC39: The sentence has been rearranged according to the suggestion provided above.

Carefully check your language again, often the sentences are in the wrong order and therefore very hard to understand.

AC40: Thanks for the comment. We have tried to improve the language.

Page 15, Line 7: What is a 'higher' litter layer?

AC41: The paragraph has been rephrased:

"Since a considerable range of dry bulk density values (see Table 2) is measured within the footprint of the neutron detector the sensitivity of neutron intensity to litter and mineral soils dry bulk density is examined using four model setups. Relative to the Gludsted Plantation reference model higher and lower values of dry bulk density are used. The first model includes a higher dry bulk density of 0.50 g cm-3 for the litter layer, while the second model holds a higher dry bulk density of 1.60 g cm-3 for the mineral soil. The third model has a low dry bulk density of 0.20 g cm-3 specified for the litter layer, and in the fourth model the mineral soil is described by a low dry bulk density of 0.60 g cm-3. The four model setups only provided slightly different thermal and epithermal neutron intensities (Table 6)."

Page 15, Line 9-12: Any ideas on why this is observed?

AC42: The results are discussed in Section 4.2.:

"The reverse effect of increased dry bulk density of litter and mineral soil on neutron intensity is a result of the different elemental composition of the two materials. The production rate of low energy neutrons (<1 MeV) per incident high-energy neutron is higher for interactions with elements of higher atomic mass (A2/3, where A is the atomic mass) (Zreda et al., 2012). Heavier elements are in particular found in mineral soil and an increase in the dry bulk density entails a higher production rate and therefore higher neutron intensity. The concentration of hydrogen is increased with an increased dry bulk density of litter material resulting in a

greater moderation and absorption of neutrons, and as a consequence lower neutron intensities. To summarize, the mineral soil acts as a producer of thermal and epithermal neutrons, while the litter acts as an absorber."

Page 15, Line 27: 'we chose'

AC43: The spelling has been corrected.

Page 16, Line 1: So is this change in t/e ratio larger than the noise/measurement uncertainty?

AC44: The signal is smaller than the measurements uncertainty at our field site. This is discussed in Section 4.3.:

"Ground level thermal neutron intensity was found to be sensitive to canopy interception, however, the signal is small and within the measurement uncertainty at Gludsted Plantation. In order to obtain a signal-to-noise ratio of 1, either an 11-hour-integration time or 11 detectors similar to the installed detector are needed. However, longer integration times are not appropriate when considering Gludsted Plantation as the return time of canopy interception (cycling between precipitation and evaporation) often is short (half-hourly to hourly time resolution). Although the change in the t/e ratio with wetting/drying of the forest canopy is small the canopy interception may potentially be measured using cosmic-ray neutron intensity detectors at locations with: 1) a high neutron intensity level (lower latitude and/or higher altitude, 2) more sensitive neutron detectors, and 3) greater amounts of canopy interception with longer residence time (e.g. snow). We suggest future studies investigating the effect of canopy interception on the neutron intensity signal to be performed at locations matching one or more of these criteria."

Page 16, Line 4-5: Maybe 'relation' is a more appropriate word than 'sensitivity'. Not only here but throughout the manuscript you should review the use of 'sensitivity' which describes in my view the responsiveness of one variable to the change of a certain parameter (mostly used in a modeling context).
Then you could write: The relation of thermal and epithermal neutron intensity to the amount of forest biomass using the forest canopy conceptualization of… is presented…

AC45: We prefer "sensitivity". Based on reviewers comment we also find it to be the appropriate word as the sensitivity analysis is performed in a modeling context. In order to explain the concept of sensitivity analysis a sentence has been added to Section 1:

"For the sensitivity analysis, one component at the time is changed in the model and the sensitivity of the component is quantified by calculating the change in the neutron intensity relative to a reference model."

Page 16, Line 29: 'provide'

AC46: The spelling has been corrected.

Page 17, Line 15: Are the bi-weekly averages of measurements really much wider? They should be the same, no?

AC47: Thank you for pointing this out. You are right, the bi-weekly averages of measurements are the exact same. The wording of the sentence has been changed and the paragraph has been rearranged a bit:

"Overall, a remarkable agreement is seen for the Model Tree trunk, Foliage, Air in Fig. 8 when comparing the two-year-average of the measured ratio with the modeled value of Gludsted Plantation (100 t/ha dry above-ground biomass, Figure 8). The biweekly averages of measurements are all within the ratios modeled for biomass of 50 t/ha - 200 t/ha. For the Model Foliage in Fig. 9 the measured ratio is in better agreement with a lower biomass (50 t/ha dry above-ground biomass). The small increase in t/e ratio with increasing amounts of biomass of Model Foliage causes the biweekly averages of the measurements to exceed both the lower and upper boundary of ratios provided by the models of 50 t/ha and 400 t/ha dry above-ground biomass."

Page 17, Line 27: The area average soil moisture derived from field sampling (and oven-drying)?

AC48: The soil moisture was estimated using the cosmic-ray neutron method. This has been specified in the sentence:

"The area average soil moisture estimated using the measured cosmic-ray neutron intensity was similar for the two field campaigns."

Page 17, Line 28: You do have that data to look at. I mean you have the soil moisture profile from the field sampling and the soil moisture content of the litter layer, no? You could mention the observed differences more explicitly.

AC49: Unfortunately no soil sampling field campaigns were performed on the days of measuring neutron intensity profiles.

Page 17, Line 31: 'AdditionalLY'

AC50: The spelling has been corrected.

Page 18, Line 12: 'The sensitivity OF neutron…'. Again, this should be picked up by native English speakers.

AC51: The title has been changed to: "The sensitivity of neutron intensity to soil chemistry and dry bulk density"

Page 18, Line 13: 'Contrary to Gludsted Plantation, the sensitivity of thermal and epithermal neutron intensity profiles to soil chemistry was…'

AC52: The sentence has been changed according to the suggested.

Page 18, Line 16-26: This is an excellent example of how to discuss and interpret your findings. Very clear and comprehensible.

AC53: Thank you.
Page 19, Line 3: See Page 18, Line 12.

AC54: The title has been changed to: "The sensitivity of biomass to neutron intensity"

Page 20, Line 26: '…using the simplest…'.

AC55: The sentence has been corrected.

Page 20, Line 27: '…the forest's…'.

AC56: The sentence has been corrected.

Page 20, Line 31: You should use only one tense in a sentence (or better, in the whole section) not both past and present mixed.

AC57: The sentence has been changed to: "However, the increase was minor and the measurement uncertainty was found to exceed the signal of canopy interception at a timescale appropriate to detect canopy interception at Gludsted Plantation (half-hourly to hourly)."

Page 20, Line 32: '…half-hourLY…'.

AC58: The sentence has been corrected.

Page 21, Line 2: '…more sensitive…'.

AC59: The sentence has been corrected.

Page 21, Line 9: '…were found…'.

AC60: The sentence has been corrected.

Figure 1: It looks as if something is wrong with this figure. You should make the background of the small map of Denmark nontransparent.

AC61: Figure 1 has been changed.

[Figure]

Figure 6: Better write the caption this way: 'Sensitivity to (A.) litter layer thickness using Model Tree trunk, Foliage, Air, (B.) canopy interception using Model Tree trunk, Foliage, Air, biomass using (C.) Model Tree trunk, Foliage, Air and (D.) Model Foliage, respectively…'

[revised manuscript text omitted]

---

## Author Response (AR4)

**7 March, 2017**

**Editor**

Dear authors,

thanks a lot for the detailed response and the modification in the paper. I think the paper looks now very good and is in general ready for publication in HESS. When studying the revised version, I found some technical corrections and changes that should be included into the final manuscript:

Dear Markus Weiler,

Thank you for your comments. We very much appreciate you taking the time to make these additional edits. Following your suggestions we believe that the readability of the tables and figures has been improved.

Best regards,

Mie Andreasen

Table 5 - please remove the left column with the content "Gludsted Plantation models (Fig. 3)". This is not relevant for the table.

AC1 (Author comment # 1): The left column in Table 5 has been removed. Furthermore, a title ("Models") has been added to the header row of the (new) left column.

Table 6 - please move the unit of the soil moisture the the header row of the table and also provide a header description for the two left columns

AC2: The unit of soil moisture has not been moved to the header row because other units are provided further down in the column (e.g., centimeters, millimeters and tons per hectare). Instead, an overall title ("Models") has been included in the header for the two left columns. This is done in order to emphasize that these two columns in total are describing the different models used in the sensitivity analysis. Other small adjustments have been done in order to improve the readability of the table. This includes the naming of the models and the overall setup of the table.

In general, most figures could be simplified by removing the symbols for all modelled data. Usually, observations, as they are not continuous, are illustrated as symbols and simulations are illustrated as lines. This would also help your figures as they are very busy with all the symbols for the simulated neutron intensities (fig 2-5, 8, 9).

AC3: Figures 3-5 and 7-9 have been changed according to the suggested guideline.

Please change the extend of the y axis in Figure 7 - all information is situated between 0.8 and 0.85 - so why showing such a wide range? The legend could me move in between the individual graphs.

AC4: The extent of the y-axis of Figure 7 has been decreased to 0.75 to 0.90. The range is a little wider than the suggested range. This is because the range of t/e ratio values goes a little below 0.80.

Best regards

Markus Weiler

[revised manuscript text omitted]

Formatted Table

| Page 28: [2] Formatted | Mie Andreasen | 3/7/2017 2:13:00 PM |
|---|---|---|

Line spacing:  single

| Page 28: [3] Deleted Cells | Mie Andreasen | 3/7/2017 2:13:00 PM |
|---|---|---|

Deleted Cells

| Page 28: [4] Formatted | Mie Andreasen | 3/7/2017 2:13:00 PM |
|---|---|---|

Line spacing:  single

| Page 28: [5] Formatted | Mie Andreasen | 3/7/2017 2:13:00 PM |
|---|---|---|

Line spacing:  single

| Page 28: [6] Formatted | Mie Andreasen | 3/7/2017 2:13:00 PM |
|---|---|---|

Line spacing:  single

| Page 28: [7] Formatted | Mie Andreasen | 3/7/2017 2:13:00 PM |
|---|---|---|

Line spacing:  single

| Page 28: [8] Formatted Table | Mie Andreasen | 3/7/2017 2:13:00 PM |
|---|---|---|

Formatted Table

| Page 28: [9] Formatted | Mie Andreasen | 3/7/2017 2:13:00 PM |
|---|---|---|

Centered, Line spacing:  single

| Page 28: [10] Formatted | Mie Andreasen | 3/7/2017 2:13:00 PM |
|---|---|---|

Line spacing:  single

| Page 28: [11] Formatted | Mie Andreasen | 3/7/2017 2:13:00 PM |
|---|---|---|

Line spacing:  single

| Page 28: [12] Formatted | Mie Andreasen | 3/7/2017 2:13:00 PM |
|---|---|---|

English (U.S.)

| Page 28: [12] Formatted | Mie Andreasen | 3/7/2017 2:13:00 PM |
|---|---|---|

English (U.S.)

| Page 28: [13] Formatted | Mie Andreasen | 3/7/2017 2:13:00 PM |
|---|---|---|

Font color: Black, Danish

| Page 28: [13] Formatted | Mie Andreasen | 3/7/2017 2:13:00 PM |
|---|---|---|

Font color: Black, Danish

| Page 28: [14] Formatted | Mie Andreasen | 3/7/2017 2:13:00 PM |
|---|---|---|

Line spacing:  single

| Page 28: [15] Formatted | Mie Andreasen | 3/7/2017 2:13:00 PM |
|---|---|---|

Font color: Black, Danish

| Page 28: [15] Formatted | Mie Andreasen | 3/7/2017 2:13:00 PM |
|---|---|---|

Font color: Black, Danish

| Page 28: [16] Formatted | Mie Andreasen | 3/7/2017 2:13:00 PM |
|---|---|---|

English (U.S.)

| Page 28: [17] Formatted | Mie Andreasen | 3/7/2017 2:13:00 PM |
|---|---|---|

English (U.S.)

| Page 28: [18] Formatted | Mie Andreasen | 3/7/2017 2:13:00 PM |
|---|---|---|

English (U.S.)

| Page 28: [19] Formatted | Mie Andreasen | 3/7/2017 2:13:00 PM |
|---|---|---|

English (U.S.)

| Page 28: [20] Formatted | Mie Andreasen | 3/7/2017 2:13:00 PM |
|---|---|---|

Line spacing:  single

| Page 28: [21] Formatted | Mie Andreasen | 3/7/2017 2:13:00 PM |
|---|---|---|

English (U.S.)

| Page 28: [22] Formatted | Mie Andreasen | 3/7/2017 2:13:00 PM |
|---|---|---|

Font color: Black, Danish

| Page 28: [22] Formatted | Mie Andreasen | 3/7/2017 2:13:00 PM |
|---|---|---|

Font color: Black, Danish

| Page 28: [23] Formatted | Mie Andreasen | 3/7/2017 2:13:00 PM |
|---|---|---|

English (U.S.)

| Page 28: [24] Formatted | Mie Andreasen | 3/7/2017 2:13:00 PM |
|---|---|---|

English (U.S.)

| Page 28: [25] Formatted | Mie Andreasen | 3/7/2017 2:13:00 PM |
|---|---|---|

English (U.S.)

| Page 28: [26] Formatted | Mie Andreasen | 3/7/2017 2:13:00 PM |
|---|---|---|

English (U.S.)

| Page 28: [27] Formatted | Mie Andreasen | 3/7/2017 2:13:00 PM |
|---|---|---|

Line spacing:  single

| Page 28: [28] Formatted | Mie Andreasen | 3/7/2017 2:13:00 PM |
|---|---|---|

English (U.S.)

| Page 28: [29] Formatted | Mie Andreasen | 3/7/2017 2:13:00 PM |
|---|---|---|

Font color: Black

| Page 28: [29] Formatted | Mie Andreasen | 3/7/2017 2:13:00 PM |
|---|---|---|

Font color: Black

| Page 28: [30] Formatted | Mie Andreasen | 3/7/2017 2:13:00 PM |
|---|---|---|

English (U.S.)

| Page 28: [31] Formatted | Mie Andreasen | 3/7/2017 2:13:00 PM |
|---|---|---|

English (U.S.)

| Page 28: [32] Formatted | Mie Andreasen | 3/7/2017 2:13:00 PM |
|---|---|---|

English (U.S.)

| Page 28: [33] Formatted | Mie Andreasen | 3/7/2017 2:13:00 PM |
|---|---|---|

English (U.S.)

| Page 28: [34] Formatted | Mie Andreasen | 3/7/2017 2:13:00 PM |
|---|---|---|

Line spacing:  single

| Page 28: [35] Formatted | Mie Andreasen | 3/7/2017 2:13:00 PM |
|---|---|---|

English (U.S.)

| Page 28: [36] Formatted | Mie Andreasen | 3/7/2017 2:13:00 PM |
|---|---|---|

Font color: Black

| Page 28: [37] Formatted | Mie Andreasen | 3/7/2017 2:13:00 PM |
|---|---|---|

Line spacing:  single

| Page 28: [38] Formatted | Mie Andreasen | 3/7/2017 2:13:00 PM |
|---|---|---|

Font color: Black

| Page 28: [39] Formatted | Mie Andreasen | 3/7/2017 2:13:00 PM |
|---|---|---|

Line spacing:  single

| Page 28: [40] Formatted | Mie Andreasen | 3/7/2017 2:13:00 PM |
|---|---|---|

Font color: Black

| Page 28: [41] Formatted | Mie Andreasen | 3/7/2017 2:13:00 PM |
|---|---|---|

English (U.S.)

| Page 28: [42] Formatted | Mie Andreasen | 3/7/2017 2:13:00 PM |
|---|---|---|

Line spacing:  single

| Page 28: [43] Formatted | Mie Andreasen | 3/7/2017 2:13:00 PM |
|---|---|---|

Font color: Black

| Page 28: [44] Formatted | Mie Andreasen | 3/7/2017 2:13:00 PM |
|---|---|---|

Line spacing:  single

| Page 28: [45] Formatted | Mie Andreasen | 3/7/2017 2:13:00 PM |
|---|---|---|

Font color: Black

| Page 28: [46] Formatted | Mie Andreasen | 3/7/2017 2:13:00 PM |
|---|---|---|

Line spacing:  single

| Page 28: [47] Formatted | Mie Andreasen | 3/7/2017 2:13:00 PM |
|---|---|---|

Font color: Black

| Page 28: [48] Formatted | Mie Andreasen | 3/7/2017 2:13:00 PM |
|---|---|---|

Line spacing:  single

| Page 28: [49] Formatted | Mie Andreasen | 3/7/2017 2:13:00 PM |
|---|---|---|

Font color: Black

| Page 28: [50] Formatted | Mie Andreasen | 3/7/2017 2:13:00 PM |
|---|---|---|

Line spacing:  single

| Page 28: [51] Formatted | Mie Andreasen | 3/7/2017 2:13:00 PM |
|---|---|---|

Font color: Black

| Page 29: [52] Formatted | Mie Andreasen | 3/7/2017 2:13:00 PM |
|---|---|---|

Line spacing:  single

| Page 29: [53] Formatted | Mie Andreasen | 3/7/2017 2:13:00 PM |
|---|---|---|

Line spacing:  single

| Page 29: [54] Formatted | Mie Andreasen | 3/7/2017 2:13:00 PM |
|---|---|---|

Line spacing:  single

| Page 29: [55] Formatted | Mie Andreasen | 3/7/2017 2:13:00 PM |
|---|---|---|

Line spacing:  single

| Page 29: [56] Formatted | Mie Andreasen | 3/7/2017 2:13:00 PM |
|---|---|---|

Line spacing:  single

| Page 29: [57] Formatted | Mie Andreasen | 3/7/2017 2:13:00 PM |
|---|---|---|

Line spacing:  single

| Page 29: [58] Formatted | Mie Andreasen | 3/7/2017 2:13:00 PM |
|---|---|---|

Line spacing:  single

| Page 29: [59] Formatted | Mie Andreasen | 3/7/2017 2:13:00 PM |
|---|---|---|

Font: Calibri, 11 pt

| Page 29: [60] Formatted | Mie Andreasen | 3/7/2017 2:13:00 PM |
|---|---|---|

English (U.S.)

| Page 29: [61] Formatted | Mie Andreasen | 3/7/2017 2:13:00 PM |
|---|---|---|

English (U.S.)

| Page 29: [62] Formatted | Mie Andreasen | 3/7/2017 2:13:00 PM |
|---|---|---|

English (U.S.)

| Page 29: [63] Formatted | Mie Andreasen | 3/7/2017 2:13:00 PM |
|---|---|---|

English (U.S.)

English (U.S.)

Line spacing:  single

English (U.S.)

English (U.S.)

English (U.S.)

English (U.S.)

English (U.S.)

English (U.S.)

Line spacing:  single

English (U.S.)

English (U.S.)

English (U.S.)

English (U.S.)

| Page 29: [77] Formatted | Mie Andreasen | 3/7/2017 2:13:00 PM |
|---|---|---|

English (U.S.)

| Page 29: [78] Formatted | Mie Andreasen | 3/7/2017 2:13:00 PM |
|---|---|---|

Danish

| Page 29: [78] Formatted | Mie Andreasen | 3/7/2017 2:13:00 PM |
|---|---|---|

Danish

| Page 29: [79] Formatted | Mie Andreasen | 3/7/2017 2:13:00 PM |
|---|---|---|

Line spacing:  single

| Page 29: [80] Formatted | Mie Andreasen | 3/7/2017 2:13:00 PM |
|---|---|---|

English (U.S.)

| Page 29: [81] Formatted | Mie Andreasen | 3/7/2017 2:13:00 PM |
|---|---|---|

English (U.S.)

| Page 29: [82] Formatted | Mie Andreasen | 3/7/2017 2:13:00 PM |
|---|---|---|

English (U.S.)

| Page 29: [83] Formatted | Mie Andreasen | 3/7/2017 2:13:00 PM |
|---|---|---|

English (U.S.)

| Page 29: [84] Formatted | Mie Andreasen | 3/7/2017 2:13:00 PM |
|---|---|---|

English (U.S.)

| Page 29: [85] Formatted | Mie Andreasen | 3/7/2017 2:13:00 PM |
|---|---|---|

Line spacing:  single

| Page 29: [86] Formatted | Mie Andreasen | 3/7/2017 2:13:00 PM |
|---|---|---|

Font: Calibri, 11 pt

| Page 29: [87] Formatted | Mie Andreasen | 3/7/2017 2:13:00 PM |
|---|---|---|

Line spacing:  single

| Page 29: [88] Formatted | Mie Andreasen | 3/7/2017 2:13:00 PM |
|---|---|---|

Line spacing:  single

| Page 29: [89] Formatted | Mie Andreasen | 3/7/2017 2:13:00 PM |
|---|---|---|

Line spacing:  single

| Page 29: [90] Formatted | Mie Andreasen | 3/7/2017 2:13:00 PM |
|---|---|---|

Line spacing:  single

| Page 29: [91] Formatted | Mie Andreasen | 3/7/2017 2:13:00 PM |
|---|---|---|

Line spacing:  single

| Page 29: [92] Formatted | Mie Andreasen | 3/7/2017 2:13:00 PM |
|---|---|---|

Line spacing:  single

| Page 29: [93] Formatted | Mie Andreasen | 3/7/2017 2:13:00 PM |
|---|---|---|

Line spacing:  single

| Page 29: [94] Formatted | Mie Andreasen | 3/7/2017 2:13:00 PM |
|---|---|---|

Line spacing:  single

| Page 29: [95] Formatted | Mie Andreasen | 3/7/2017 2:13:00 PM |
|---|---|---|

Line spacing:  single

| Page 29: [96] Formatted | Mie Andreasen | 3/7/2017 2:13:00 PM |
|---|---|---|

Font color: Black

| Page 29: [97] Formatted | Mie Andreasen | 3/7/2017 2:13:00 PM |
|---|---|---|

Line spacing:  single

| Page 29: [98] Formatted | Mie Andreasen | 3/7/2017 2:13:00 PM |
|---|---|---|

Font: Calibri, 11 pt

| Page 29: [99] Formatted | Mie Andreasen | 3/7/2017 2:13:00 PM |
|---|---|---|

Line spacing:  single

| Page 29: [100] Formatted | Mie Andreasen | 3/7/2017 2:13:00 PM |
|---|---|---|

Font color: Black

| Page 29: [101] Formatted | Mie Andreasen | 3/7/2017 2:13:00 PM |
|---|---|---|

Line spacing:  single

| Page 29: [102] Formatted | Mie Andreasen | 3/7/2017 2:13:00 PM |
|---|---|---|

Font color: Black

| Page 29: [103] Formatted | Mie Andreasen | 3/7/2017 2:13:00 PM |
|---|---|---|

Line spacing:  single

| Page 29: [104] Formatted | Mie Andreasen | 3/7/2017 2:13:00 PM |
|---|---|---|

Font color: Black

| Page 29: [105] Formatted | Mie Andreasen | 3/7/2017 2:13:00 PM |
|---|---|---|

Line spacing:  single

| Page 29: [106] Formatted | Mie Andreasen | 3/7/2017 2:13:00 PM |
|---|---|---|

Font color: Black

| Page 29: [107] Formatted | Mie Andreasen | 3/7/2017 2:13:00 PM |
|---|---|---|

Line spacing:  single

| Page 29: [108] Formatted | Mie Andreasen | 3/7/2017 2:13:00 PM |
|---|---|---|

Line spacing:  single

| Page 29: [109] Formatted | Mie Andreasen | 3/7/2017 2:13:00 PM |
|---|---|---|

Font color: Black

| Page 29: [110] Formatted | Mie Andreasen | 3/7/2017 2:13:00 PM |
|---|---|---|

Line spacing:  single

| Page 29: [111] Formatted | Mie Andreasen | 3/7/2017 2:13:00 PM |
|---|---|---|

Font color: Black

| Page 29: [112] Formatted | Mie Andreasen | 3/7/2017 2:13:00 PM |
|---|---|---|

Line spacing:  single

| Page 29: [113] Formatted | Mie Andreasen | 3/7/2017 2:13:00 PM |
|---|---|---|

Font color: Black